# *Artemisia* pollen dataset for exploring the potential ecological indicators in deep time

Li-Li Lu[a,d†], Bo-Han Jiao [a,d†], Feng Qin [b†], Gan Xie[a†], Kai-Qing Lu[a,d], Jin-Feng Li[a], Bin Sun[a], Min Li[a], David K. Ferguson[c], Tian-Gang Gao[a,d *], Yi-Feng Yao[a,d*], Yu-Fei Wang[a,d*]

[a]*State Key Laboratory of Systematic and Evolutionary Botany, Institute of Botany, Chinese Academy of Sciences, 20 Nanxincun Xiangshan, Beijing 100093, China*

[b]*Key Laboratory of Land Surface Pattern and Simulation, Institute of Geographic Sciences and Natural Resources Research, Chinese Academy of Sciences, Beijing 100101, China*

[c]*Department of Paleontology, University of Vienna, Althanstrasse 14, Vienna A-1090, Austria*

[d]*University of Chinese Academy of Sciences, Beijing 100049, China*

[†] These authors contributed equally to this work.

[*]Corresponding authors. Tel: +86 (10) 62836439.

Email addresses: gaotg@ibcas.ac.cn (T. G. GAO); yaoyf@ibcas.ac.cn (Y. F. YAO); wangyf@ibcas.ac.cn (Y. F. WANG).

**Abstract.** *Artemisia*, along with Chenopodiaceae is the dominant component growing in the desert and dry grassland of the Northern Hemisphere. *Artemisia* pollen with its high productivity, wide distribution, and easy identification, is usually regarded as an eco-indicator for assessing aridity and distinguishing grassland from desert vegetation in terms of the pollen relative abundance ratio of Chenopodiaceae/*Artemisia* (C/A). Nevertheless, divergent opinions on the degree of aridity evaluated by *Artemisia* pollen have been circulating in the palynological community for a long time. To solve the confusion, we first selected 36 species from 9 clades and 3 outgroups of *Artemisia* based on the phylogenetic framework, which attempts to cover the maximum range of pollen morphological variation. Then, sampling, experiments, photography, and measurements were taken using standard methods. Here, we present pollen datasets containing 4018 original pollen photographs, 9360 pollen morphological trait measurements, information on 30858 source plant occurrences, and corresponding environmental factors. Hierarchical cluster analysis on pollen morphological traits was carried out to subdivide *Artemisia* pollen into three types. When plotting the three pollen types of *Artemisia* onto the global terrestrial biomes, different pollen types of *Artemisia* were found to have different habitat ranges. These findings change the traditional concept of *Artemisia* being restricted to arid and semi-arid environments. The data framework that we designed is open and expandable for new pollen data of *Artemisia* worldwide. In the future, linking pollen morphology with habitat via these pollen datasets will create additional knowledge that will increase the resolution of the ecological environment in the geological past. The *Artemisia* pollen datasets are freely available at Zenodo (https://doi.org/10.5281/zenodo.6900308; Lu et al., 2022).

## 1 Introduction

The concept of global change can be considered as any consistent trend in the environment - past, present, or projected - that affects a substantial part of the globe. Consequently past climates shed light on our future (Tierney et al., 2020). When attempting to reconstruct past global change prior to meteorological records, we need some appropriate biological or abiotic proxies based on long-term, consistently collected data, e.g. leaf wax biomarkers (Bhattacharya et al., 2018), tree-ring data (Moberg et al., 2005), leaf form (Yang et al., 2015), pollen data (Mosbrugger et al., 2005; Guiot and Cramer, 2016; Marsicek et al., 2018), atmospheric carbon dioxide (Zachos et al., 2008; Beerling and Royer, 2011), and isotope records (Zachos et al., 2001; Sánchez-Murillo et al., 2019). Determining a suitable proxy to reconstruct palaeoclimate and palaeoenvironment is a great scientific challenge (Tierney et al., 2020; McClelland et al., 2021).

The pollen of *Artemisia* (A), together with that of Chenopodiaceae (C) in arid and semi-arid areas, in the form of the ratio of C/A pollen abundance, was applied to distinguish grassland and desert vegetation types and assess the degree of drought in the geological past (El-Moslimany, 1990; Sun et al., 1994; Davies and Fall, 2001; Herzschuh et al., 2004; Xu et al., 2007; Zhao et al., 2009; Zhang et al., 2010; Zhao et al., 2012; Li et al., 2017; Ma et al., 2017; Koutsodendris et al., 2019; Wang et al., 2020), because both Chenopodiaceae and *Artemisia* are dominant elements of desert vegetation (China Vegetation Editorial Committee, 1980; Vrba, 1980; Tarasov et al., 1998; Herzschuh et al., 2004; Li et al., 2010; Zhao et al., 2021), and the sum of their pollen relative abundances in the surface soil is usually more than 50% in arid and semi-arid areas (Sun et al., 1994; Lu et al., 2020).

Among them, the pollen of *Artemisia*, with its high productivity, wide spatial and temporal distribution, easy identification, and morphological uniformity under the light microscope (LM), is an essential component and useful bio-indicator in pollen-based past vegetation reconstructions and environmental assessments. Some researchers regarded *Artemisia* as an aridity indicator (El-Moslimany, 1990; Yi et al., 2003a; Yi et al., 2003b; Liu et al., 2006; Cai et al., 2019; Cui et al., 2019; Chen et al., 2020; Wu et al., 2020; Cao et al., 2021), while others suggested that the correlation between the relative abundance of *Artemisia* pollen and humidity was insignificant (Weng et al., 1993; Sun et al., 1996; Koutsodendris et al., 2019; Lu et al., 2020; Zhao et al., 2021). Consequently, there is an urgent need to evaluate whether different pollen types of *Artemisia* represent distinct habitats.

In the past, *Artemisia* pollen was regarded as very uniform under LM (Wodehouse, 1926; Sing and Joshi, 1969; Ling, 1982; Wang et al., 1995). For instance, following the description and statistics of pollen morphology of 27 species of *Artemisia* in Eurasia under LM, Sing and Joshi (1969) stated that the pollen grains of *Artemisia* are consistent and continuous in morphology. Later, some authors recognized a series of pollen types (Chen, 1987; Jiang et al., 2005; Ghahraman et al., 2007; Shan et al., 2007; Hayat et al., 2009; Hayat et al., 2010; Hussain et al., 2019), based on a detailed survey of the pollen micromorphology of different taxa under the scanning electron microscope (SEM).

For example, Chen (1987) described the pollen morphology of 77 *Artemisia* species from China under LM and SEM and divided these pollen grains into six types by using pollen characters, such as the shape and size of the spinules as well as the density of spinules and granules. Type I (sparse spinules with granules among them), type II (dense spinules, no or few granules), type III (sparse spinules, no granules), type IV (dense spinules, well-developed granules), type V (small and sparse spinules, smooth tectum) and type VI (dissimilar spinules with granules among them).

Shan et al. (2007) investigated the pollen morphology of 32 *Artemisia* species from the Loess Plateau of China under LM and SEM and divided these pollen grains into five types according to exine sculpture: type I (dense spinules with swollen bases, small granules), type II (dense spinules, swollen bases almost united), type III (dense spinules with swollen bases and smooth tectum), type IV (sparse small spinules and smooth tectum) and type V (sparse spinules, small granules).

Jiang et al. (2005) observed the pollen morphology of 57 representative plants in 7 groups of *Artemisia* under LM and SEM. This pollen can be divided into two types based on exine sculpture: type I (spinules multi-ruminated with flared bases, connecting the mostly densely arranged spinules) and type II (densely or loosely arranged spinules without flared bases, interspace glandular or smooth) with subtypes II-1, II-2, II-3, and II-4 based on the distribution of the spinules.

Ghahraman et al. (2007) studied the pollen morphology of 26 species of the 33 *Artemisia* species in Iran under LM and SEM. Based on exine ornamentation observed under SEM, two types of pollen grains were recognized: type I, exine surface covered with dense acute spinules, and type II, exine surface with few spinules.

Hayat et al. (2009, 2010) carried out a palynological study of 22 *Artemisia* species from Pakistan under LM and SEM. Earlier work demonstrated the phylogenetic associations within *Artemisia* based on a phylogenetic analysis of 9 characters (pollen type, pollen shape, spinule arrangement, exine sculpture, spinule

base, the length of polar axis, the length of equatorial axis, exine thickness, and colpus width) of pollen grains of *Artemisia*. In the latter work, eight micromorphological characters were identified and pooled by cluster analysis, leading to the recognition of 5 groups.

Hussain et al. (2019) studied the pollen morphology of 15 *Artemisia* species in the Gilgit-Baltistan region of Pakistan utilizing SEM and divided these species into four groups based on cluster analysis of seven micromorphological characters (pollen type, pollen shape, spinule arrangement, exine sculpture, spinule base, polar length, and equatorial width).

Almost all of the above-mentioned *Artemisia* pollen classifications were designed to solve taxonomic or phylogenetic problems, and only a few were concerned with linking diverse habitats to the different pollen types in *Artemisia*.

Here we attempt to 1) present abundant pollen photographs of 36 species from 9 branches and 3 outgroups of the genus (ca. 400 species worldwide, see Ling, 1982; Bremer and Humphries, 1993), constrained by the phylogenetic framework of *Artemisia* (Sanz et al., 2008; Malik et al., 2017); 2) describe and measure the morphological traits of these pollen grains; 3) provide a new classification of pollen types and their distribution worldwide, with a key to pollen types in *Artemisia*; 4) explore the diverse ecological niches of *Artemisia* represented by different pollen types in order to evaluate palaeovegetation and reconstruct palaeoenvironments.

## 2 Materials and methods

### 2.1 Sampling strategy

The 36 pollen samples studied were selected from voucher sheets in the PE herbarium at the Institute of Botany, Chinese Academy of Sciences (Fig. 1, Table B1), covering 9 main clades, i.e., Subg. *Tridentata*, Subg. *Artemisia* (contains Sect. *Artemisia*, Sect. *Abrotanum* I, Sect. *Abrotanum* II and Sect. *Abrotanum* III), Subg. *Pacifica*, Subg. *Seriphidium*, Subg. *Absinthium*, and Subg. *Dracunculus*, constrained by the phylogenetic framework of *Artemisia* (Malik et al., 2017) and 3 outgroups (Sanz et al., 2008), reflecting the maximum diversity or morphological variation under LM and SEM.

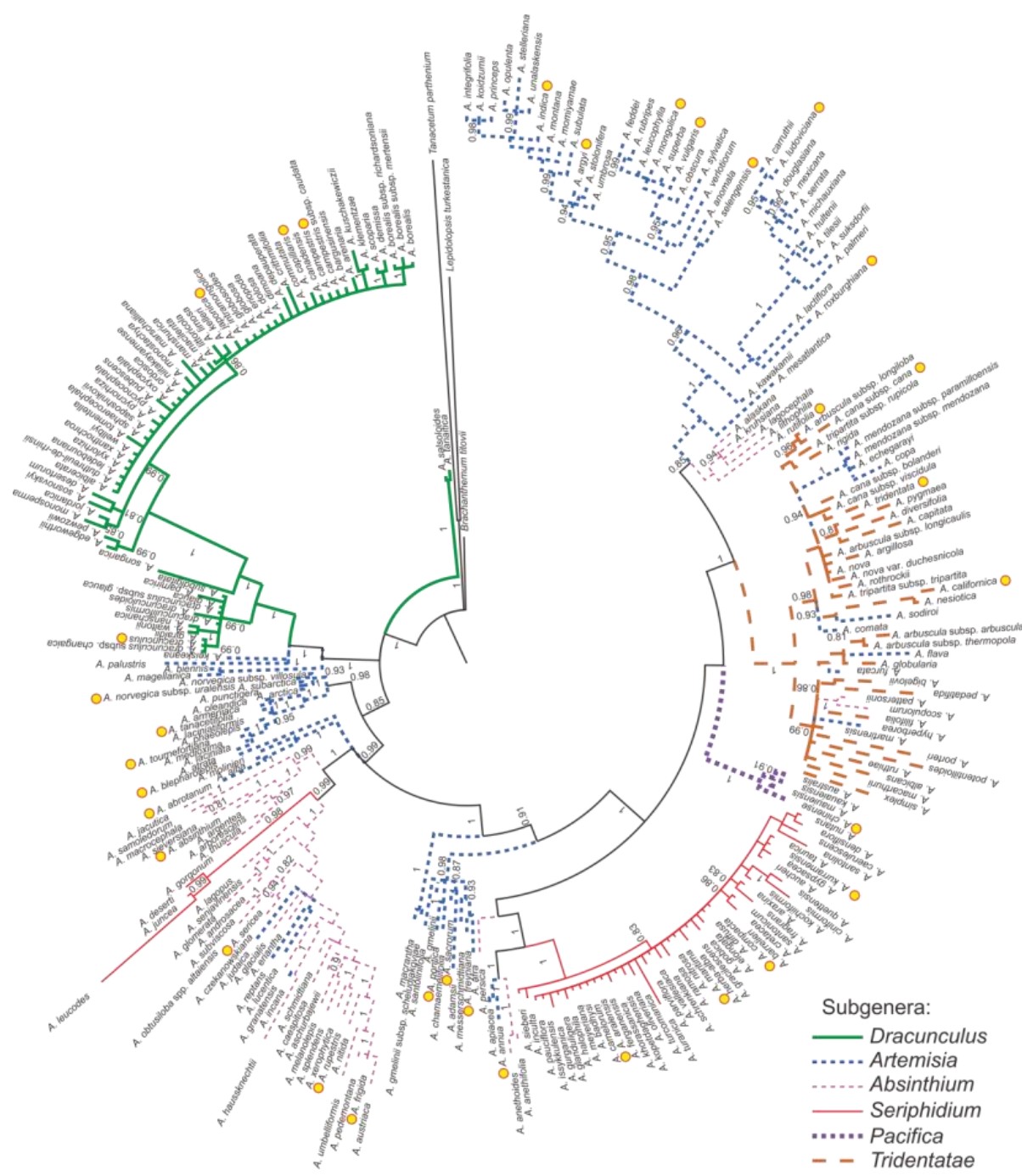

117

**Figure 1.** Phylogenetic tree of *Artemisia* (modified from Malik et al., 2017). The styles of the strokes that were used to draw the branches indicate the traditional subgeneric classification of *Artemisia*, and the yellow spots indicate sampled taxa.

## 2.2 Data acquisition

Pollen samples were acetolyzed by the standard method (Erdtman, 1960) and fixed in glycerine jelly. Standard procedures were followed for LM and SEM (Chen, 1987; Wang et al., 1995). The pollen grains were photographed under LM (Leica DM 4000) at a magnification of ×1000 and SEM (Hitachi S-4800) at an accelerating voltage of 30 kV. The pollen terminology followed the descriptions of Hesse et al. (2009) and

Halbritter et al. (2018). The statistical pollen morphological traits under LM (Figs. 2a-b, P: Polar length; E: Equatorial width; P/E; T: Exine thickness; L: Pollen length; T/L) of each species were measured using 20 pollen grains. We chose five pollen grains under SEM for each exine ornamentation trait in each species (Figs. 2c-f, D: Diameter of spinule base; H: Spinule height; D/H; Gs: Granule spacing; Ss: Spinule spacing; Gs/Ss; Ps: Perforation spacing), and on average, randomly selected four regions of each pollen grain for measuring, yielding a total of 20 measurements. The mean value (M) and standard deviation (SD) of the pollen grains of each species were measured and calculated in both polar and equatorial views (Appendix A, Table 1).

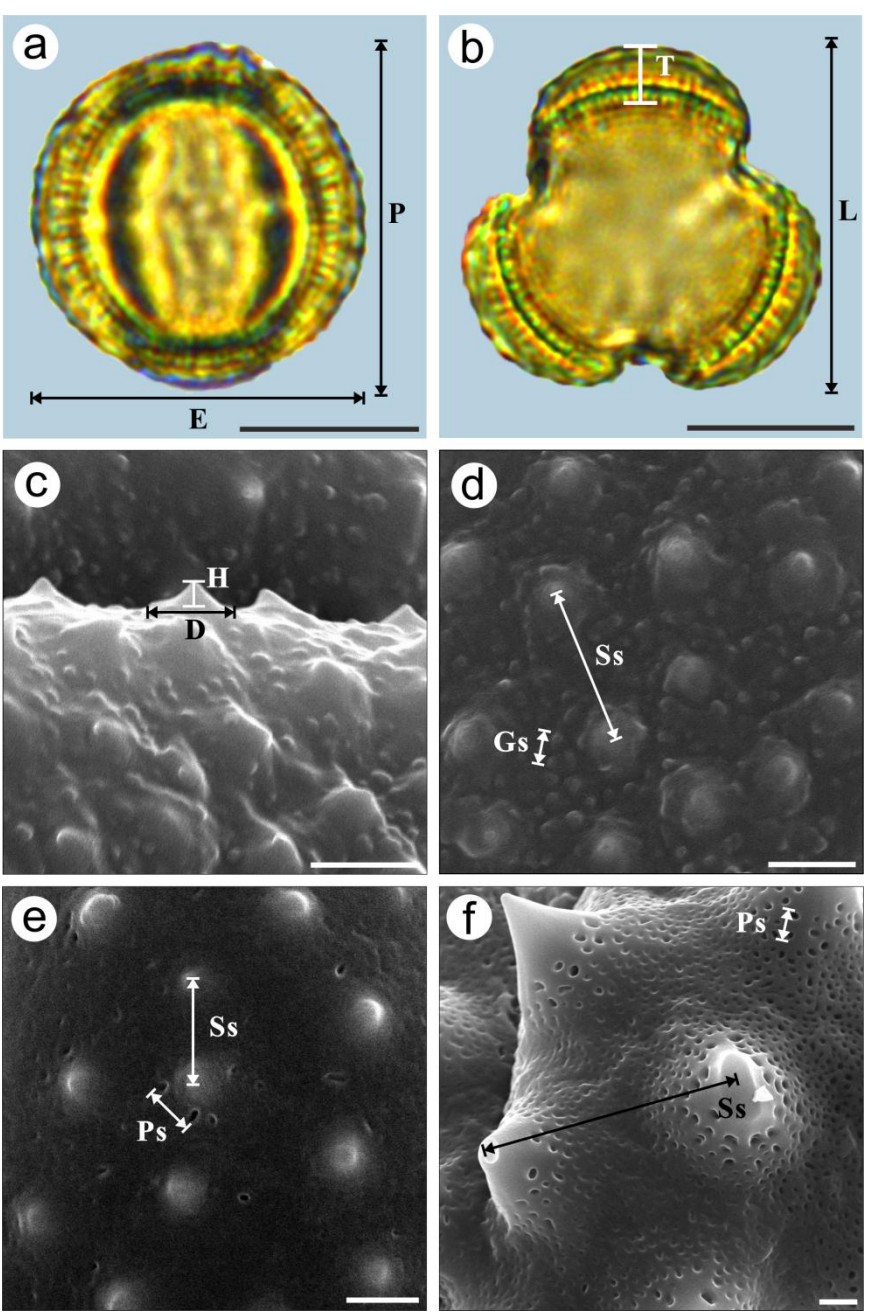

**Figure 2.** Graphical illustration of measured pollen morphological traits in *Artemisia (*a-b: *A. annua*; c-d: *A. vulgaris)* and outgroups (e: *Kaschagaria brachanthemoides;* f: *Ajania pallasiana*). Scale bar in LM and SEM overview 10 μm, in SEM close-up 1 μm.

The scientific names of selected taxa were standardized according to Plants of the World Online (https://powo.science.kew.org/). The specimen sampling coordinates of the corresponding taxa were obtained from the Global Biodiversity Information Facility (GBIF, https://www.gbif.org/). Only preserved specimens were filtered for GBIF data given their well-documented geographical information and the availability of specimens as definitive vouchers. The distribution data on observations and cultivated collections provided by GBIF were excluded because they may contain incorrect identification or incorrect geo-referencing (Brummitt et al., 2020). Next, the distribution data was standardized cleaned using R package "CoordinateCleaner" (Zizka et al., 2019); no outliers were found.

The corresponding environmental factors including altitude and 19 climate parameters of these coordinates were obtained from WorldClim ( https://www.worldclim.org/) with a spatial resolution of 30 seconds (~1 $km^2$) in 1970-2000 by Extract MultiValues To Points using ArcGIS 10.2 software in bilinear interpolation.

## 2.3 Data processing

OriginPro 2021 software was used for hierarchical cluster analysis on *Artemisia* and its outgroup pollen data. The Euclidean distance was calculated after the normalization of the original data, and the Ward method was used for clustering. Five groups were established, and the center point of each group was calculated according to the sum of distances. Pollen morphological traits for the principal component analysis (PCA) of *Artemisia* and its outgroups and grouped according to the five groups of the cluster analysis. OriginPro 2021 software was used to draw group violin diagrams and boxplots respectively, and run an ANOVA to test for an overall difference between the pollen characters of 3 pollen types and testing intraspecific variability in pollen exine ultrastructure characters among three representative species, followed by post hoc tests (Tukey). OriginPro 2021 software was also used to draw group violin diagrams and run a KWANOVA to test for overall differences between the environmental factors of the 3 pollen types. The images of habitats reproduced in the text are from the websites listed in Table B1.

The global distribution data of the 36 representative species and 3 pollen types were plotted on the map of terrestrial ecological regions (Olson et al., 2001) using ArcGIS 10.2 software (Figs. 16, 21).

## 3 Data description

### 3.1 *Artemisia* pollen grains and their source plant habitats

Here we provide detailed data on pollen morphological traits, covering 36 species from 9 main clades of *Artemisia* and 3 outgroups constrained by the phylogenetic framework (Fig. 1, Sanz et al., 2008; Malik et al., 2017) under LM and SEM, the habitats of their source plants (Figs. 3-14).

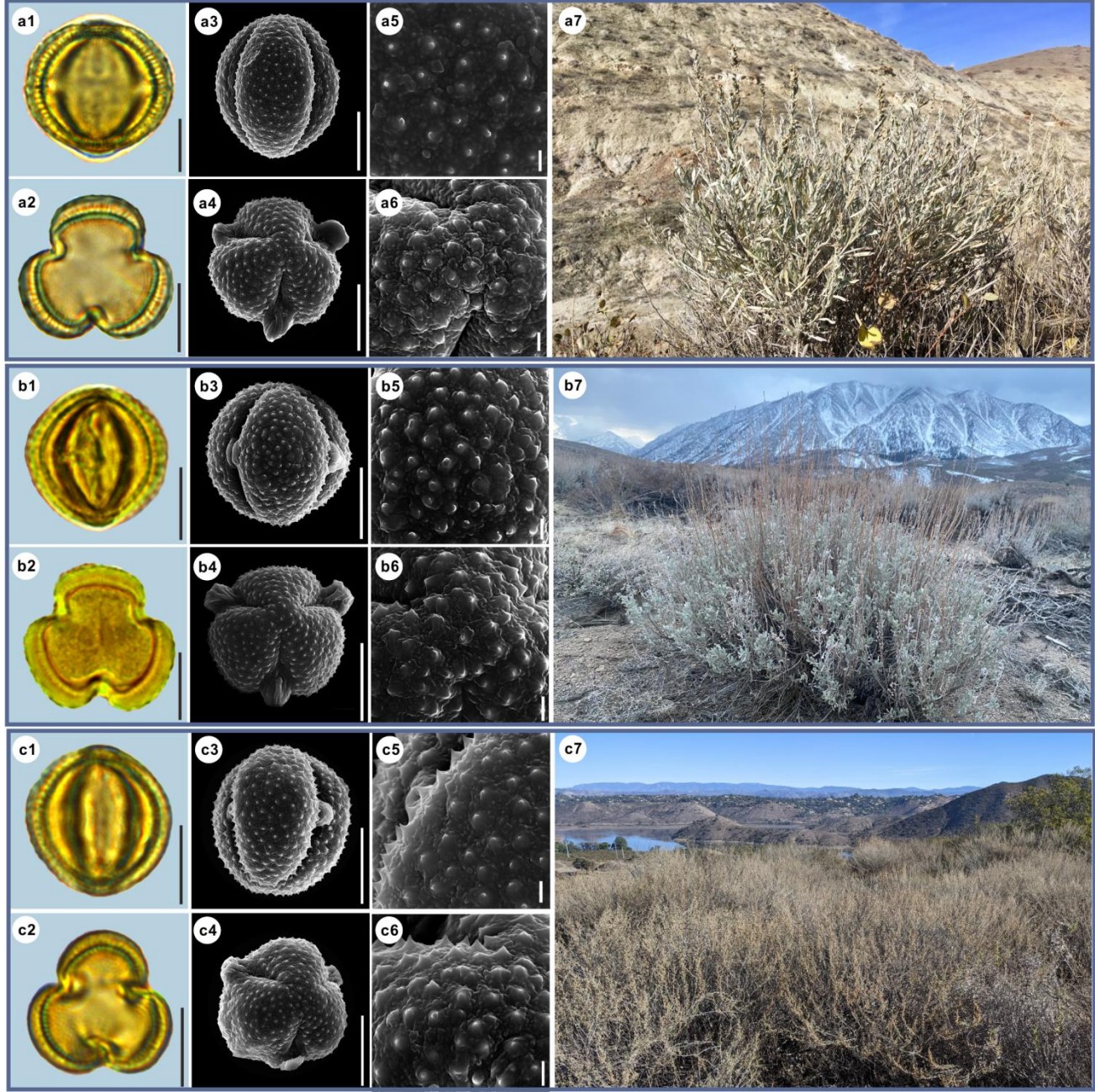

**Figure 3.** Pollen grains and the habitats of their source plants.
a. *Artemisia cana*; b. *Artemisia tridentata*; c. *Artemisia californica*.

Pollen grains in equatorial view under LM (a1, b1, c1) and SEM (a3, a5, b3, b5, c3, c5), in polar view under LM (a2, b2, c2) and SEM (a4, a6, b4, b6, c4, c6), along with the habitats of their source plants (a7 cited from https://www.inaturalist.org/photos/54492753 by © Jason Headley, b7 cited from https://www.inaturalist.org/photos/117436654 by © Matt Berger, c7 cited from https://www.inaturalist.org/photos/108921528 by © Don Rideout).

Scale bar in LM and SEM overview 10 μm, in SEM close-up 1 μm.

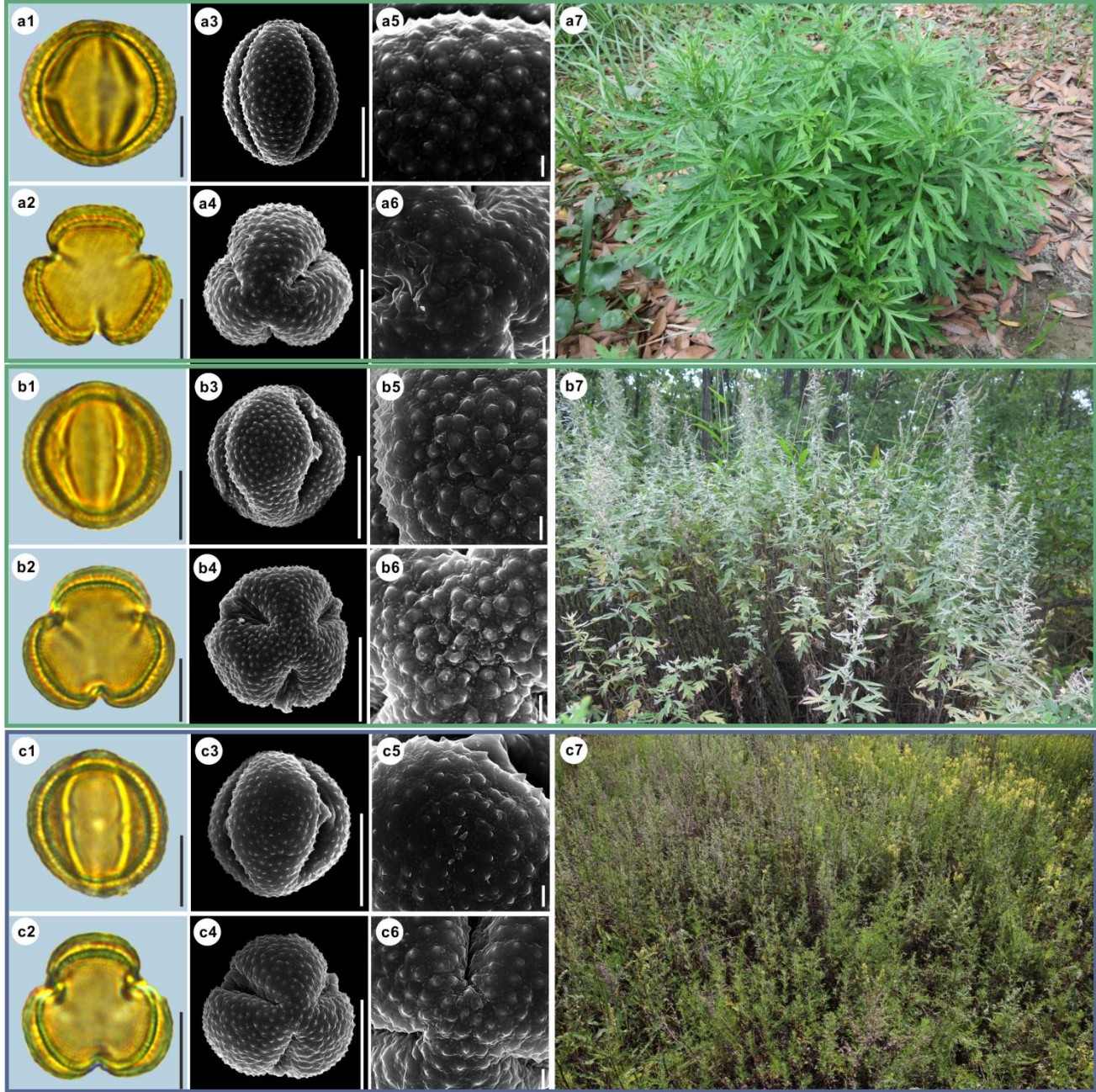

**Figure 4.** Pollen grains and the habitats of their source plants.

a. *Artemisia indica*; b. *Artemisia argyi*; c. *Artemisia mongolica*.

Pollen grains in equatorial view under LM (a1, b1, c1) and SEM (a3, a5, b3, b5, c3, c5), in polar view under LM (a2, b2, c2) and SEM (a4, a6, b4, b6, c4, c6), along with the habitats of their source plants (a7 cited from https://www.inaturalist.org/photos/66336449 by © yangting, b7 cited from

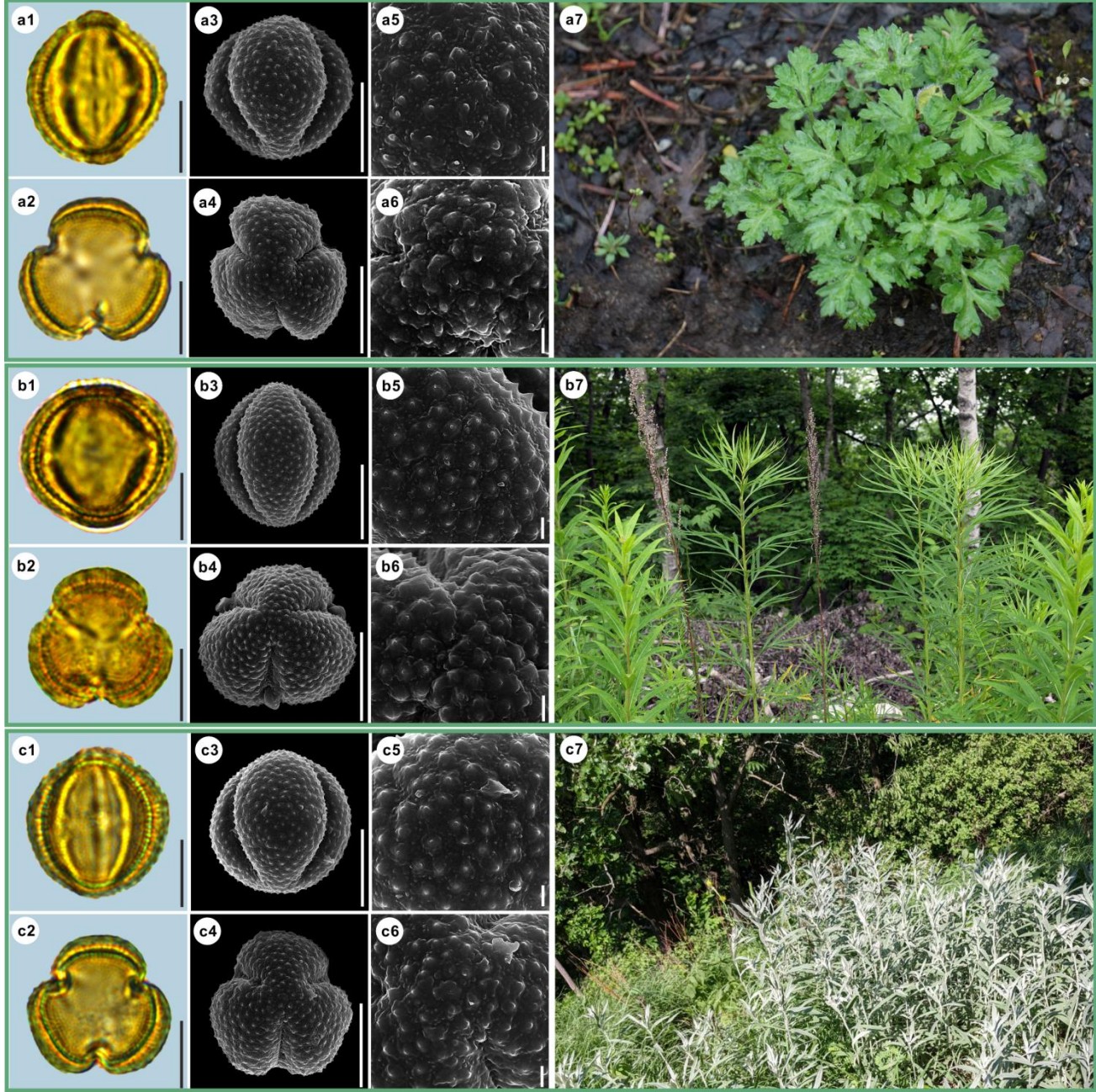

186

**Figure 5.** Pollen grains and the habitats of their source plants.

a. *Artemisia vulgaris*; b. *Artemisia selengensis*; c. *Artemisia ludoviciana*.

Pollen grains in equatorial view under LM (a1, b1, c1) and SEM (a3, a5, b3, b5, c3, c5), in polar view under LM (a2, b2, c2) and SEM (a4, a6, b4, b6, c4, c6), along with the habitats of their source plants (a7 cited from https://www.inaturalist.org/photos/120600448 by © Sara Rall, b7 cited from https://www.inaturalist.org/photos/46352423 by © Gularjanz Grigoryi Mihajlovich, c7 cited from https://www.inaturalist.org/photos/77690333 by © Ethan Rose).

Scale bar in LM and SEM overview 10 μm, in SEM close-up 1 μm.

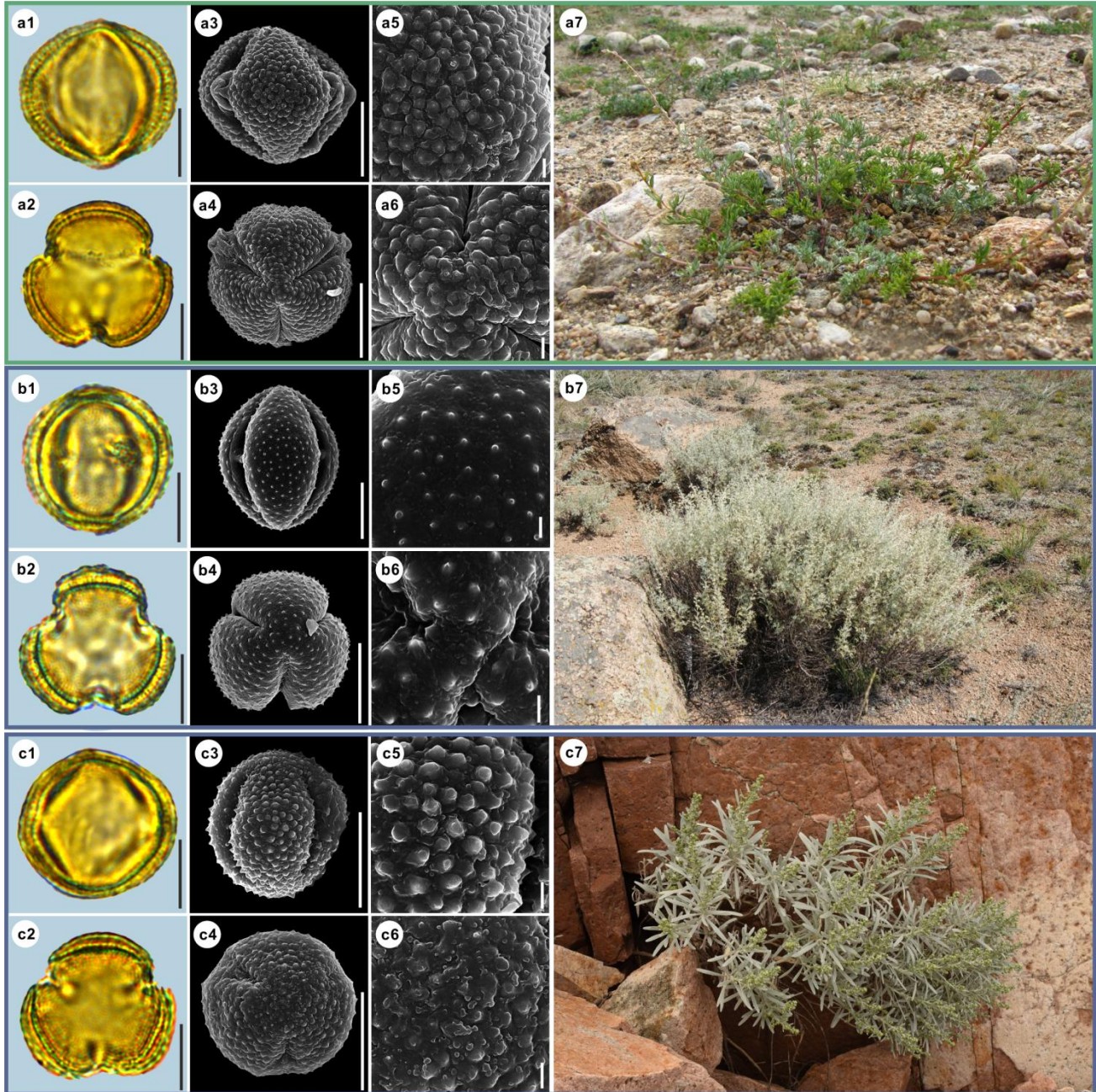

195

**Figure 6.** Pollen grains and the habitats of their source plants.

a. *Artemisia roxburghiana*; b. *Artemisia rutifolia*; c. *Artemisia chinensis*.

Pollen grains in equatorial view under LM (a1, b1, c1) and SEM (a3, a5, b3, b5, c3, c5), in polar view under LM (a2, b2, c2) and SEM (a4, a6, b4, b6, c4, c6), along with the habitats of their source plants (a7 provided by © Bo-Han Jiao, b7 cited from https://www.inaturalist.org/photos/62207191 by © Daba, c7 provided by © Jia-Hao Shen).

Scale bar in LM and SEM overview 10 μm, in SEM close-up 1 μm.

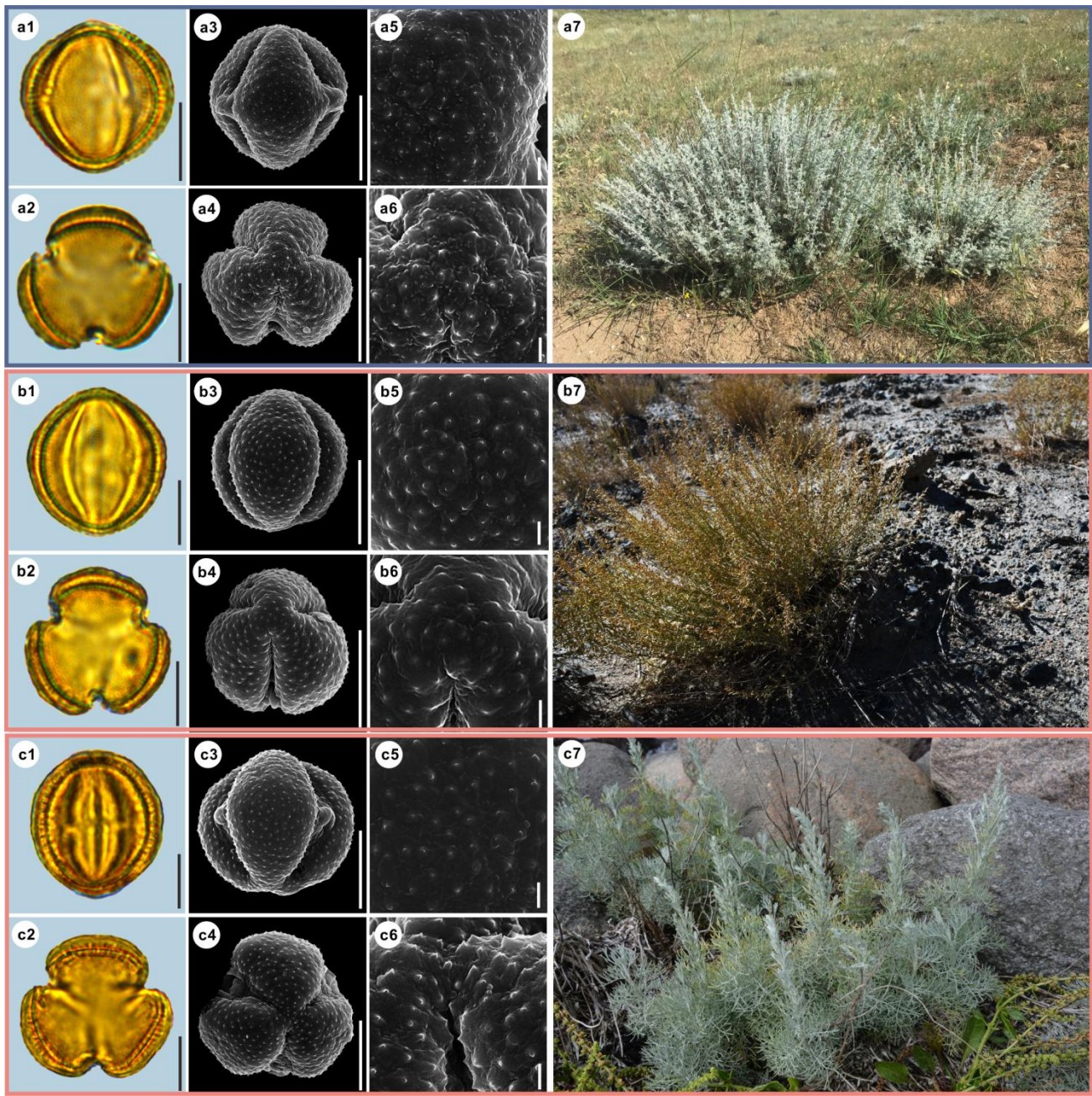

203

**Figure 7.** Pollen grains and the habitats of their source plants.

a. *Artemisia kurramensis*; b. *Artemisia compactum*; c. *Artemisia maritima*.

Pollen grains in equatorial view under LM (a1, b1, c1) and SEM (a3, a5, b3, b5, c3, c5), in polar view under LM (a2, b2, c2) and SEM (a4, a6, b4, b6, c4, c6), along with the habitats of their source plants (a7 cited from https://www.inaturalist.org/photos/133758174 by © Andrey Vlasenko, b7 provided by © Chen Chen, c7 cited from https://www.inaturalist.org/photos/86515371 by © torkild).

Scale bar in LM and SEM overview 10 μm, in SEM close-up 1 μm.

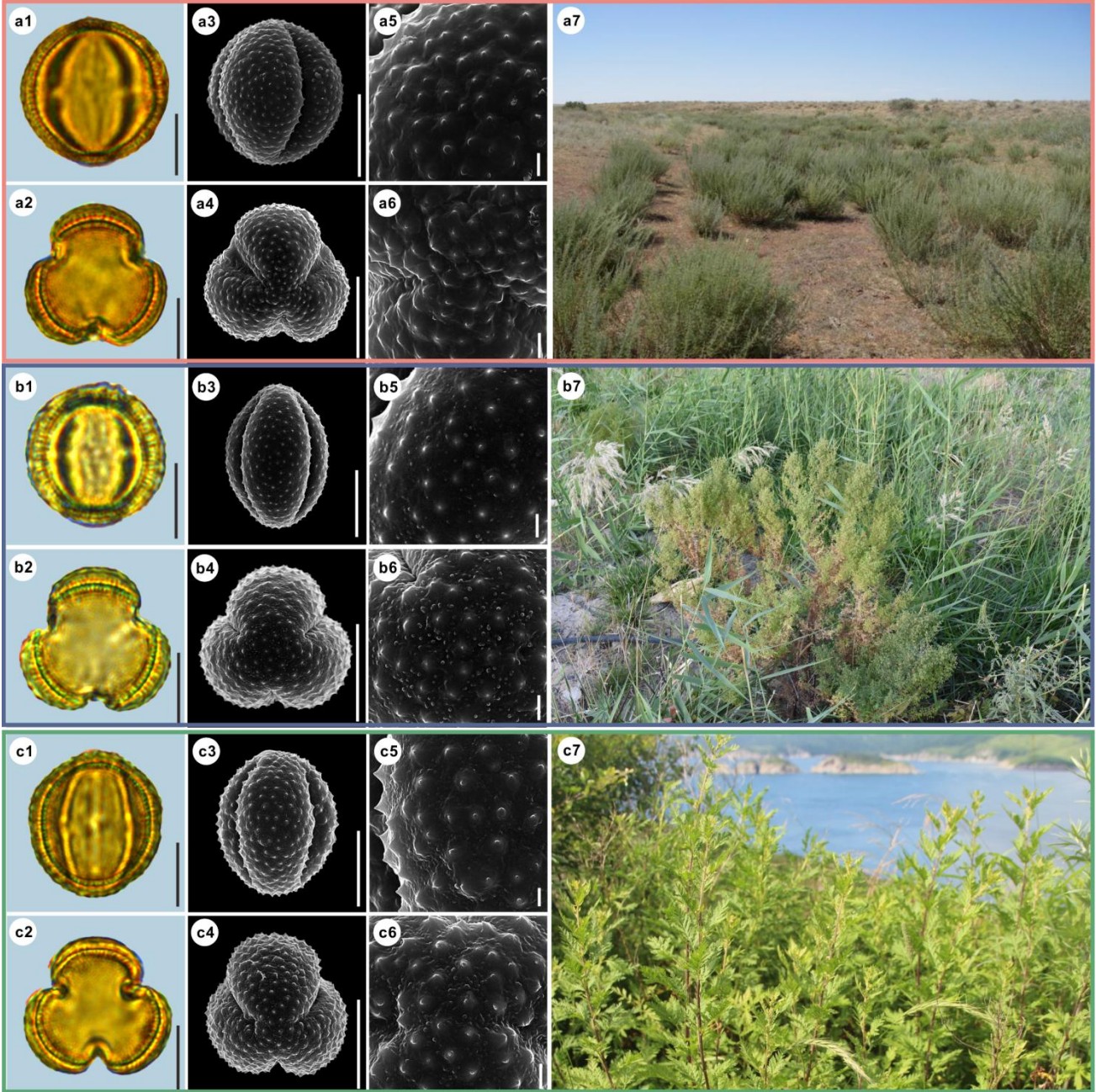

Figure 8. Pollen grains and the habitats of their source plants.

a. *Artemisia aralensis*; b. *Artemisia annua*; c. *Artemisia freyniana*.

Pollen grains in equatorial view under LM (a1, b1, c1) and SEM (a3, a5, b3, b5, c3, c5), in polar view under LM (a2, b2, c2) and SEM (a4, a6, b4, b6, c4, c6), along with the habitats of their source plants (a7 cited from https://www.plantarium.ru/lang/en/page/image/id/73063.html by © Полынь аральская, b7 provided by © Chen Chen, c7 cited from https://www.inaturalist.org/photos/154390279 by © Шильников Дмитрий Сергеевич).

Scale bar in LM and SEM overview 10 μm, in SEM close-up 1 μm.

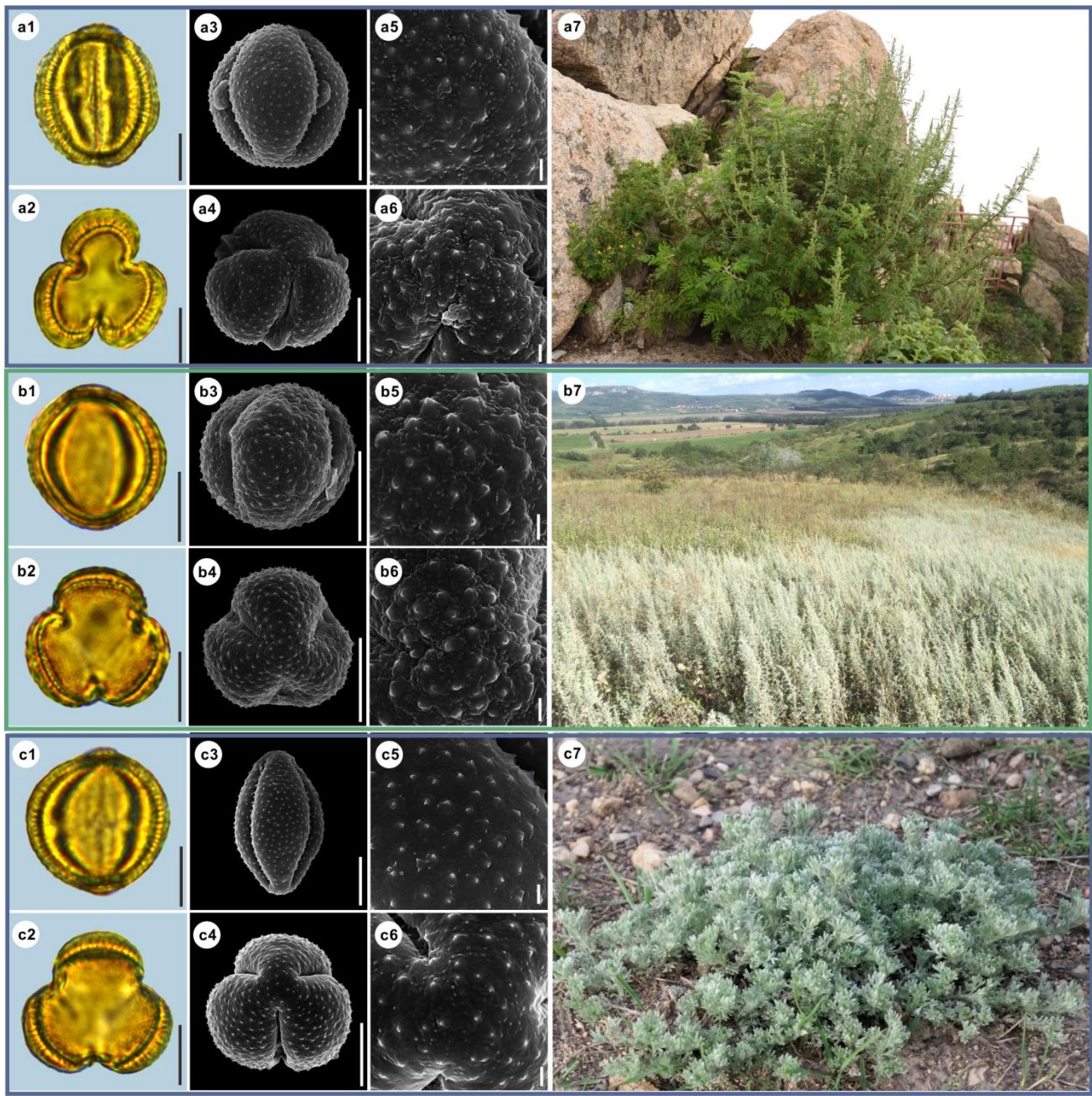

**Figure 9.** Pollen grains and the habitats of their source plants.

a. *Artemisia stechmanniana*; b. *Artemisia pontica*; c. *Artemisia frigida*.

Pollen grains in equatorial view under LM (a1, b1, c1) and SEM (a3, a5, b3, b5, c3, c5), in polar view under LM (a2, b2, c2) and SEM (a4, a6, b4, b6, c4, c6), along with the habitats of their source plants (a7 provided by © Bo-Han Jiao, b7 cited from https://www.inaturalist.org/photos/93438780 by © Martin Pražák, c7 cited from https://www.inaturalist.org/photos/125022240 by © Suzanne Dingwell).

Scale bar in LM and SEM overview 10 μm, in SEM close-up 1 μm.

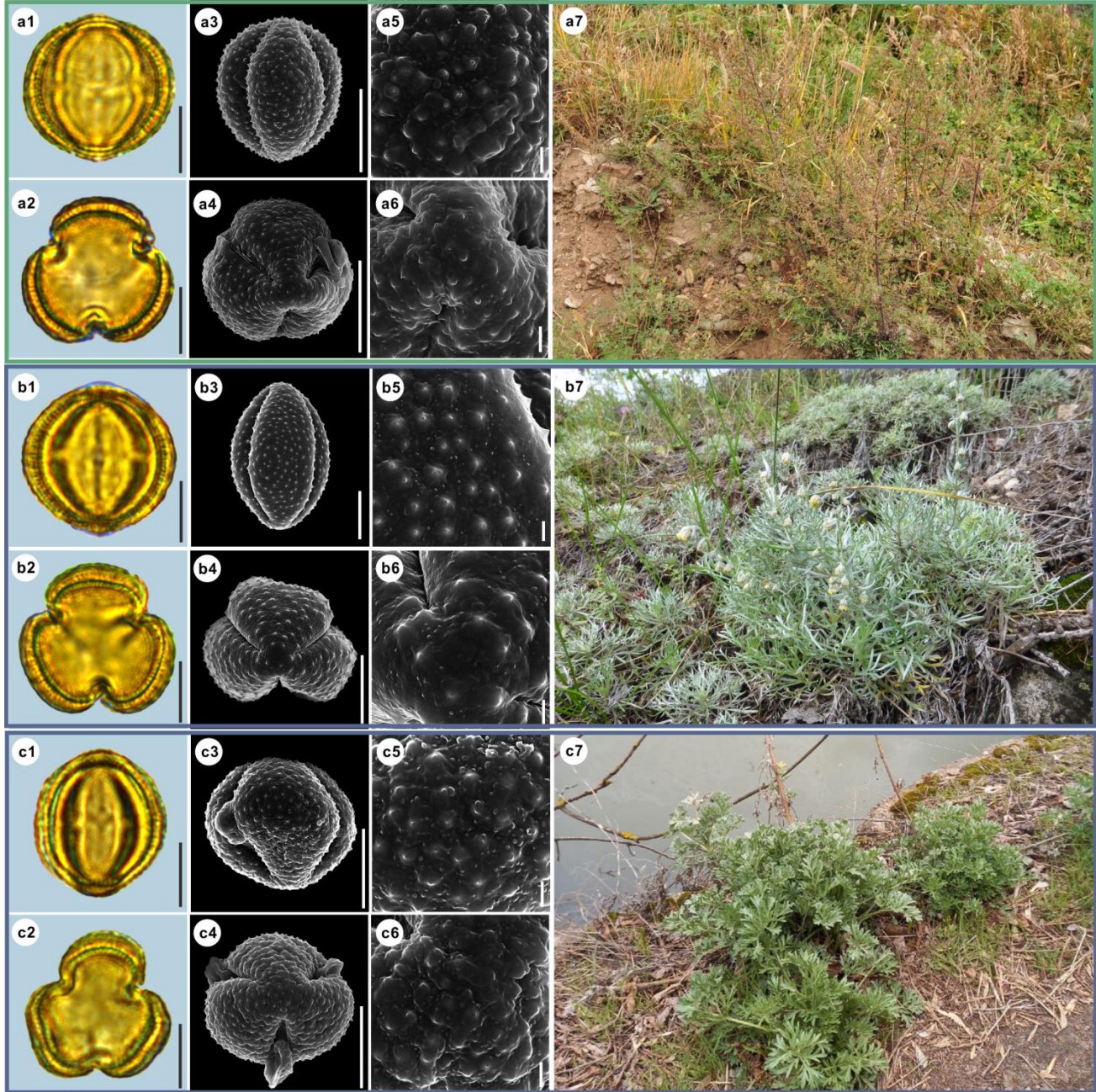

**Figure 10.** Pollen grains and the habitats of their source plants.

a. *Artemisia rupestris*; b. *Artemisia sericea*; c. *Artemisia absinthium*.

Pollen grains in equatorial view under LM (a1, b1, c1) and SEM (a3, a5, b3, b5, c3, c5), in polar view under LM (a2, b2, c2) and SEM (a4, a6, b4, b6, c4, c6), along with the habitats of their source plants (a7 provided by © Bo-Han Jiao, b7 cited from https://www.inaturalist.org/photos/48033353 by © svetlana_katana, c7 cited from https://www.inaturalist.org/photos/123569286 by © Станислав Лебедев).

Scale bar in LM and SEM overview 10 μm, in SEM close-up 1 μm.

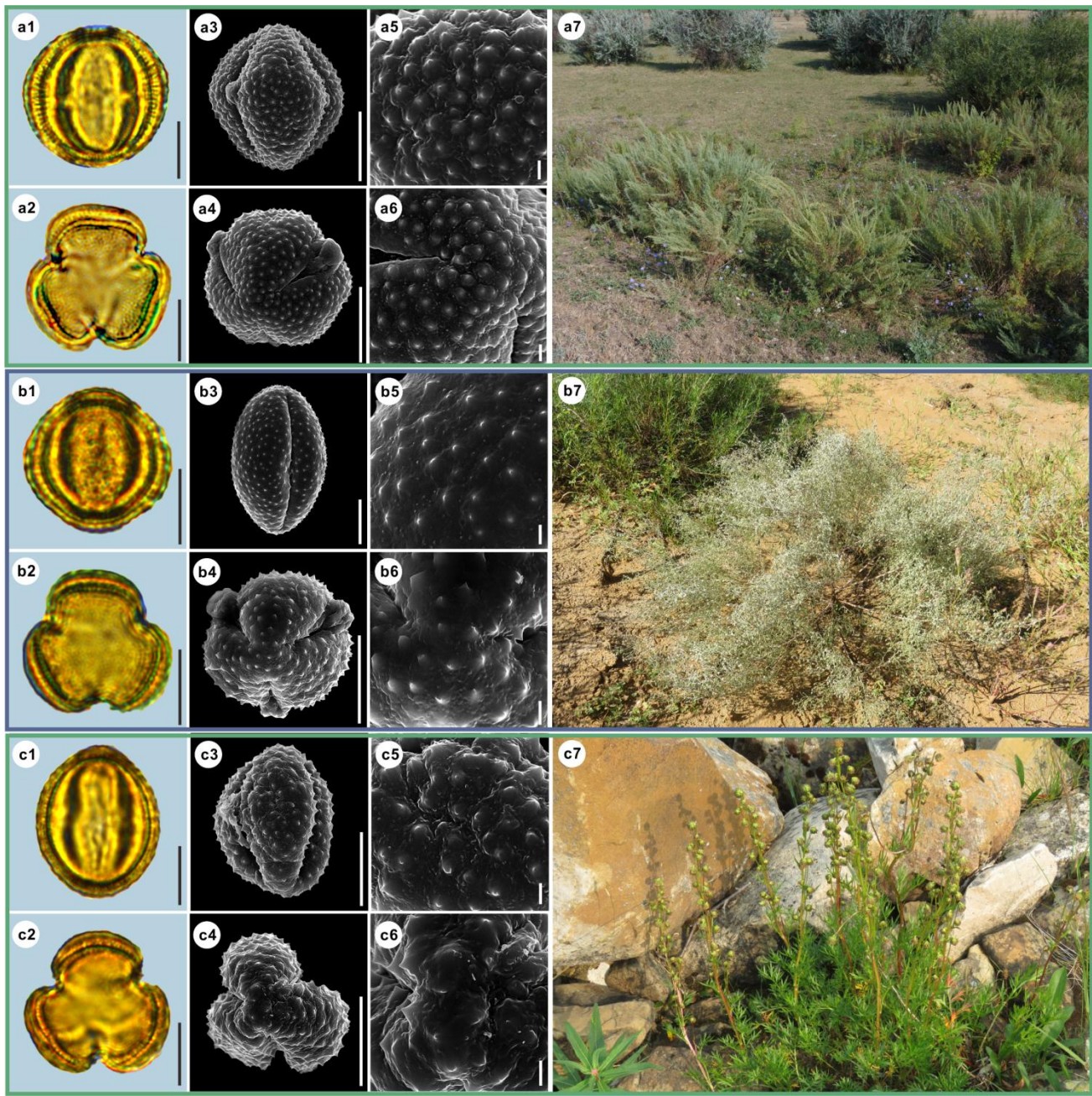

236

**Figure 11.** Pollen grains and the habitats of their source plants.

a. *Artemisia abrotanum*; b. *Artemisia blepharolepis*; c. *Artemisia norvegica*.

Pollen grains in equatorial view under LM (a1, b1, c1) and SEM (a3, a5, b3, b5, c3, c5), in polar view under LM (a2, b2, c2) and SEM (a4, a6, b4, b6, c4, c6), along with the habitats of their source plants (a7 cited from https://www.inaturalist.org/photos/116106722 by © Андрей Москвичев, b7 provided by © Ji-Ye Zheng, c7 cited from https://www.inaturalist.org/photos/161393521 by © Erin Springinotic).

Scale bar in LM and SEM overview 10 μm, in SEM close-up 1 μm.

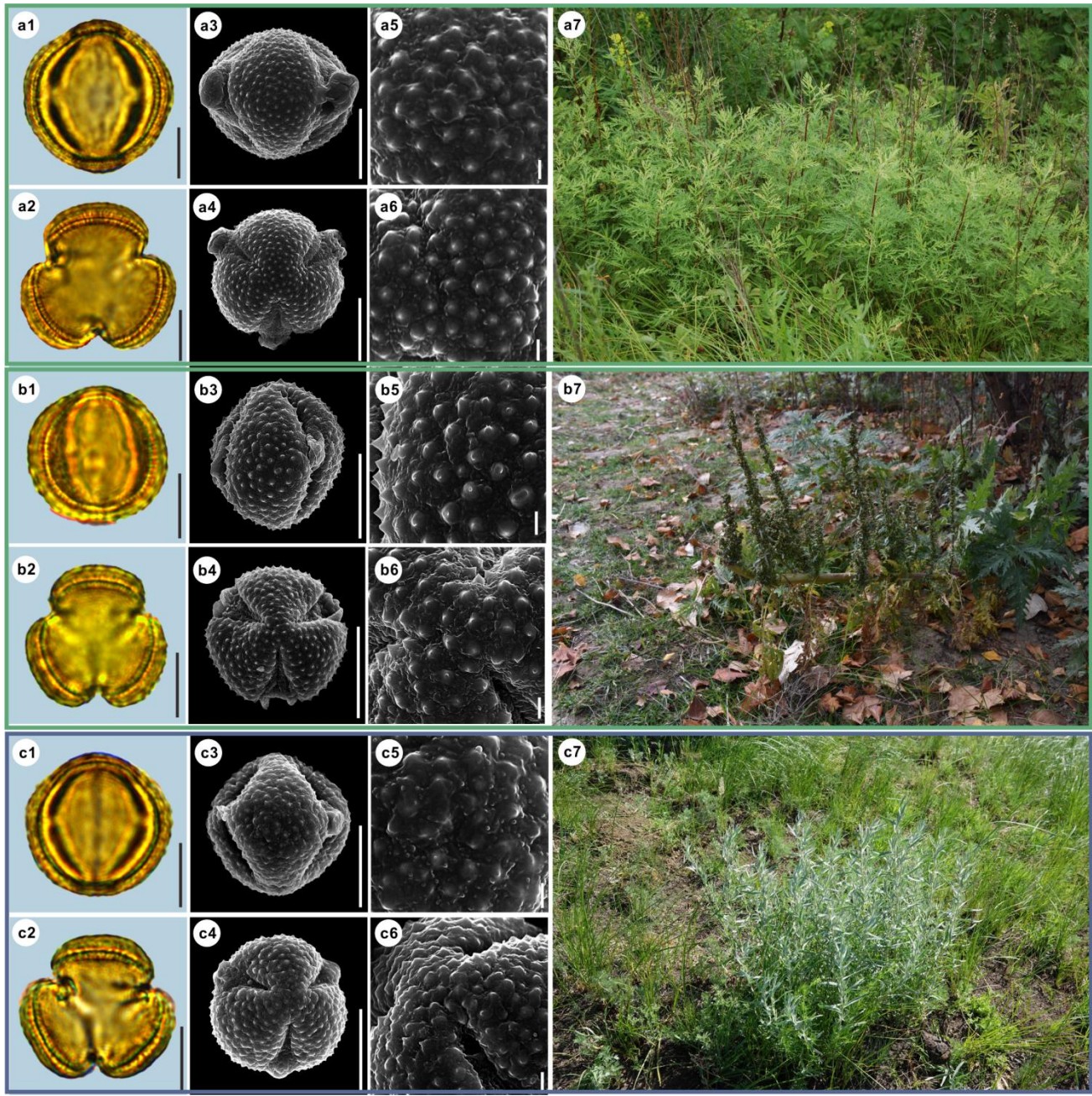

244

**Figure 12.** Pollen grains and the habitats of their source plants.

a. *Artemisia tanacetifolia*; b. *Artemisia tournefortiana*; c. *Artemisia dracunculus*.

Pollen grains in equatorial view under LM (a1, b1, c1) and SEM (a3, a5, b3, b5, c3, c5), in polar view under LM (a2, b2, c2) and SEM (a4, a6, b4, b6, c4, c6), along with the habitats of their source plants (a7 cited from https://www.inaturalist.org/photos/78902853 by © Alexander Dubynin, b7 provided by © Chen Chen, c7 cited from https://www.inaturalist.org/photos/76312868 by © anatolymikhaltsov).

Scale bar in LM and SEM overview 10 μm, in SEM close-up 1 μm.

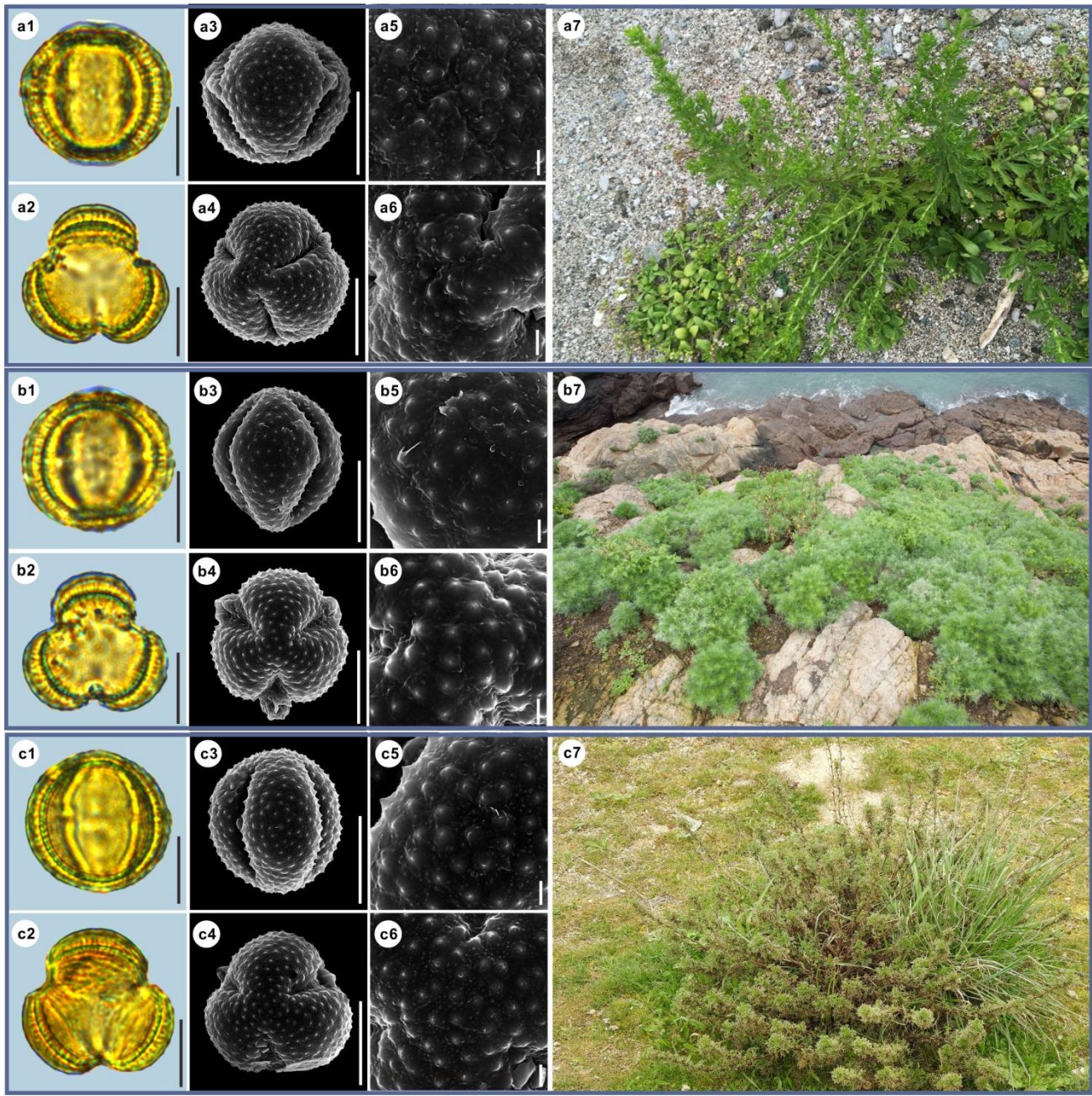

**Figure 13.** Pollen grains and the habitats of their source plants.

a. *Artemisia japonica*; b. *Artemisia capillaris*; c. *Artemisia campestris*.

Pollen grains in equatorial view under LM (a1, b1, c1) and SEM (a3, a5, b3, b5, c3, c5), in polar view under LM (a2, b2, c2) and SEM (a4, a6, b4, b6, c4, c6), along with the habitats of their source plants (a7 cited from https://www.inaturalist.org/photos/44507659 by © 陳達智 , b7 cited from https://www.inaturalist.org/photos/60639286 by © Cheng-Tao Lin, c7 cited from https://www.inaturalist.org/photos/113822257 by © pedrosanz-anapri).

Scale bar in LM and SEM overview 10 μm, in SEM close-up 1 μm.

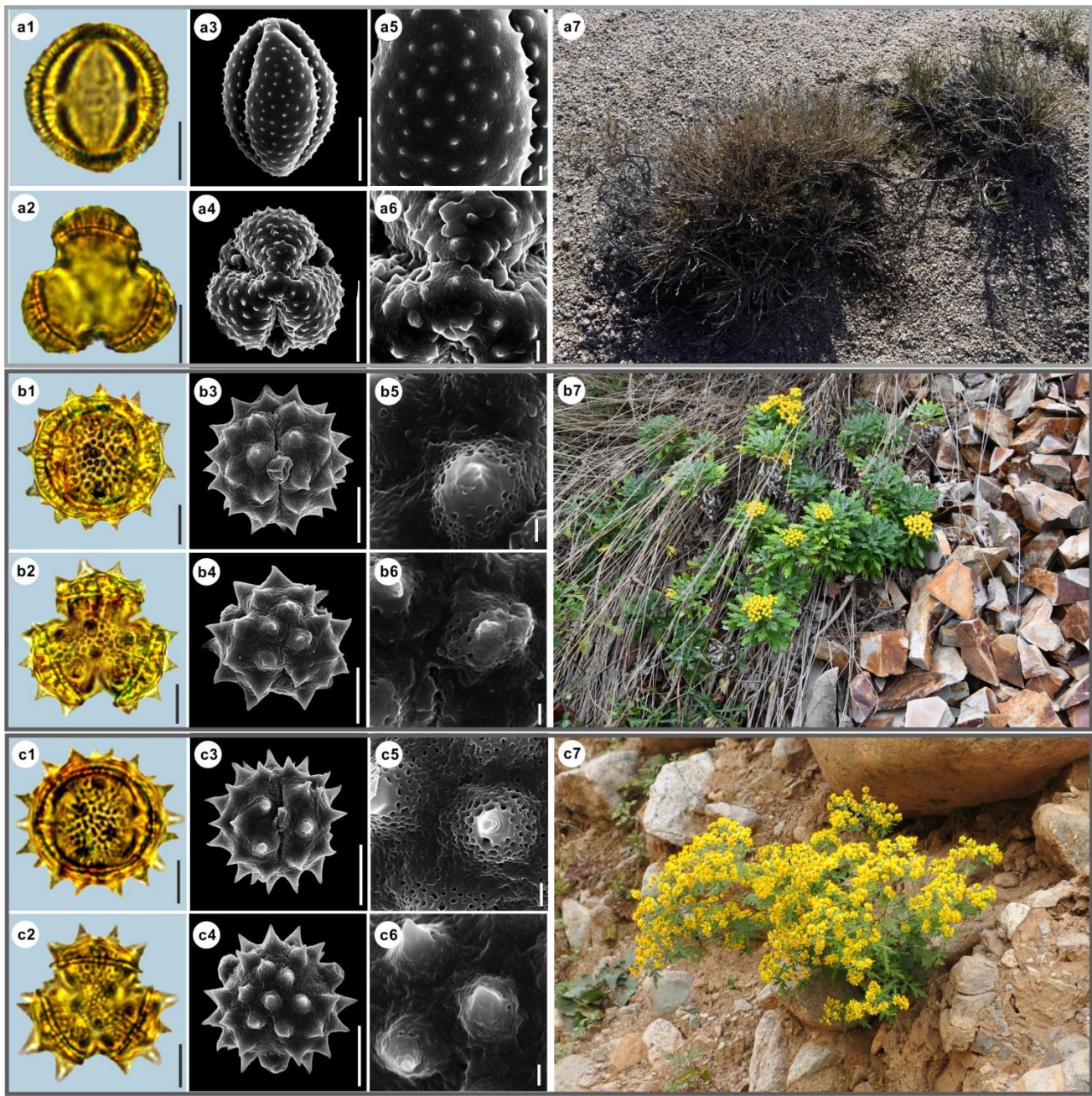

**Figure 14.** Pollen grains and the habitats of their source plants.

a. *Kaschgaria brachanthemoides*; b. *Ajania pallasiana*; c. *Chrysanthemum indicum*.

Pollen grains in equatorial view under LM (a1, b1, c1) and SEM (a3, a5, b3, b5, c3, c5), in polar view under LM (a2, b2, c2) and SEM (a4, a6, b4, b6, c4, c6), along with the habitats of their source plants (a7 provided by © Chen Chen, b7 cited from https://www.inaturalist.org/photos/162408714 by © Игорь Поспелов, c7 provided by © Bo-Han Jiao).

Scale bar in LM and SEM overview 10 μm, in SEM close-up 1 μm.

## 3.2 Statistical pollen morphological trait data of 36 sampled taxa

The mean values of 10 pollen morphological traits of 36 sampled species are listed in Table 1, and these data distribution patterns are shown in boxplots (Fig. 15) in the form of variation (25%-75%), and further described in the form of mean value ± standard deviation (M ± SD, Appendix A).

**Table 1.** Pollen morphological traits of 36 selected species (P: Polar length; E: Equatorial width; T: Exine thickness; L: Pollen length; D: Diameter of spinule base; H: Spinule height; Gs: Granule spacing; Ss: Spinule spacing; Ps: Perforation spacing).

| No. | Species | P (μm) | E (μm) | P/E | T (μm) | L (μm) | T/L | D (μm) | H (μm) | D/H | Gs (μm) | Ss (μm) | Gs/Ss | Ps (μm) |
|---|---|---|---|---|---|---|---|---|---|---|---|---|---|---|
| 1 | *Artemisia cana* | 23.46 | 24.5 | 0.96 | 3.91 | 24.58 | 0.16 | 0.58 | 0.46 | 1.28 | 0.33 | 1.60 | 0.21 | 0 |
| 2 | *Artemisia tridentata* | 21.36 | 20.69 | 1.04 | 3.55 | 22.35 | 0.16 | 0.76 | 0.60 | 1.30 | 0.24 | 1.12 | 0.22 | 0 |
| 3 | *Artemisia californica* | 18.94 | 19.13 | 0.99 | 2.70 | 18.85 | 0.14 | 0.75 | 0.71 | 1.08 | 0.24 | 1.45 | 0.17 | 0 |
| 4 | *Artemisia indica* | 23.47 | 23.81 | 0.99 | 3.50 | 23.31 | 0.15 | 0.76 | 0.39 | 2.04 | 0.28 | 1.21 | 0.24 | 0 |
| 5 | *Artemisia argyi* | 21.8 | 21.67 | 1.01 | 3.55 | 22.24 | 0.16 | 0.64 | 0.38 | 1.71 | 0.22 | 0.90 | 0.26 | 0 |
| 6 | *Artemisia mongolica* | 21.05 | 20.42 | 1.03 | 3.29 | 19.78 | 0.17 | 0.62 | 0.41 | 1.54 | 0.19 | 0.91 | 0.22 | 0 |
| 7 | *Artemisia vulgaris* | 19.72 | 19.29 | 1.03 | 2.92 | 18.94 | 0.16 | 0.69 | 0.34 | 2.13 | 0.29 | 1.55 | 0.20 | 0 |
| 8 | *Artemisia selengensis* | 20.67 | 19.68 | 1.06 | 3.72 | 20.8 | 0.18 | 0.67 | 0.38 | 1.76 | 0.22 | 1.05 | 0.22 | 0 |
| 9 | *Artemisia ludoviciana* | 21.65 | 20.82 | 1.04 | 3.71 | 20.94 | 0.18 | 0.70 | 0.37 | 1.94 | 0.2 | 1.23 | 0.16 | 0 |
| 10 | *Artemisia roxburghiana* | 23.88 | 23.69 | 1.01 | 3.78 | 21.81 | 0.17 | 0.76 | 0.39 | 1.96 | 0.28 | 0.79 | 0.36 | 0 |
| 11 | *Artemisia rutifolia* | 22.22 | 22.7 | 0.98 | 3.53 | 24.93 | 0.14 | 0.31 | 0.26 | 1.2 | 0.21 | 1.27 | 0.17 | 0 |
| 12 | *Artemisia chinensis* | 21.53 | 22.75 | 0.95 | 2.97 | 23.71 | 0.13 | 0.70 | 0.55 | 1.29 | 0.27 | 0.91 | 0.31 | 0 |
| 13 | *Artemisia kurramensis* | 19.71 | 19.35 | 1.02 | 3.30 | 19.44 | 0.17 | 0.38 | 0.27 | 1.41 | 0.23 | 1.25 | 0.19 | 0 |
| 14 | *Artemisia compactum* | 22.33 | 21.97 | 1.02 | 2.97 | 21.67 | 0.14 | 0.41 | 0.28 | 1.50 | 0.51 | 0.92 | 0.56 | 0 |
| 15 | *Artemisia maritima* | 26.24 | 23.09 | 1.14 | 3.54 | 24.42 | 0.14 | 0.28 | 0.23 | 1.30 | 0.53 | 1.08 | 0.50 | 0 |
| 16 | *Artemisia aralensis* | 22.32 | 21.91 | 1.02 | 3.16 | 22.76 | 0.14 | 0.25 | 0.22 | 1.16 | 0.50 | 1.09 | 0.46 | 0 |

| 17 | *Artemisia annua* | 19.71 | 19.45 | 1.02 | 3.45 | 19.2 | 0.18 | 0.45 | 0.39 | 1.18 | 0.27 | 1.29 | 0.21 | 0 |
|----|----|----|----|----|----|----|----|----|----|----|----|----|----|----|
| 18 | *Artemisia freyniana* | 23.39 | 21.3 | 1.10 | 3.17 | 21.29 | 0.15 | 0.56 | 0.40 | 1.40 | 0.2 | 1.15 | 0.18 | 0 |
| 19 | *Artemisia stechmanniana* | 26.31 | 25.16 | 1.05 | 3.97 | 23.45 | 0.17 | 0.37 | 0.35 | 1.07 | 0.19 | 1.40 | 0.14 | 0 |
| 20 | *Artemisia pontica* | 20.64 | 19.62 | 1.05 | 3.01 | 19.75 | 0.15 | 0.6 | 0.37 | 1.63 | 0.17 | 1.32 | 0.13 | 0 |
| 21 | *Artemisia frigida* | 25.11 | 24.9 | 1.01 | 4.61 | 24.83 | 0.19 | 0.46 | 0.32 | 1.44 | 0.31 | 1.3 | 0.24 | 0 |
| 22 | *Artemisia rupestris* | 24.45 | 22.92 | 1.07 | 3.18 | 21.96 | 0.14 | 0.55 | 0.33 | 1.68 | 0.25 | 0.91 | 0.28 | 0 |
| 23 | *Artemisia sericea* | 26.31 | 27.9 | 0.94 | 3.75 | 26.89 | 0.14 | 0.89 | 0.54 | 1.71 | 0.28 | 1.74 | 0.16 | 0 |
| 24 | *Artemisia absinthium* | 22.79 | 20.84 | 1.09 | 3.39 | 19.92 | 0.17 | 0.59 | 0.40 | 1.52 | 0.18 | 1.11 | 0.16 | 0 |
| 25 | *Artemisia abrotanum* | 24.47 | 23.73 | 1.03 | 3.15 | 18.82 | 0.17 | 0.72 | 0.51 | 1.44 | 0.22 | 1.41 | 0.16 | 0 |
| 26 | *Artemisia blepharolepis* | 18.96 | 19.26 | 0.99 | 3.15 | 18.82 | 0.17 | 0.69 | 0.44 | 1.64 | 0.37 | 1.68 | 0.23 | 0 |
| 27 | *Artemisia norvegica* | 24.51 | 22.11 | 1.11 | 3.48 | 22.61 | 0.15 | 0.67 | 0.43 | 1.66 | 0.19 | 1.56 | 0.12 | 0 |
| 28 | *Artemisia tanacetifolia* | 28.38 | 27.75 | 1.03 | 3.46 | 27.63 | 0.13 | 0.71 | 0.32 | 2.23 | 0.30 | 1.08 | 0.29 | 0 |
| 29 | *Artemisia tournefortiana* | 20.76 | 20.43 | 1.02 | 3.33 | 20.03 | 0.17 | 0.73 | 0.42 | 1.81 | 0.26 | 1.25 | 0.22 | 0 |
| 30 | *Artemisia dracunculus* | 22.89 | 22.87 | 1.00 | 2.82 | 21.91 | 0.13 | 0.68 | 0.45 | 1.56 | 0.31 | 0.92 | 0.34 | 0 |
| 31 | *Artemisia japonica* | 20.18 | 21.23 | 0.95 | 4.24 | 21.02 | 0.2 | 0.57 | 0.32 | 1.8 | 0.26 | 1.26 | 0.21 | 0 |
| 32 | *Artemisia capillaris* | 19.53 | 19.64 | 1.00 | 3.54 | 19.18 | 0.18 | 0.51 | 0.36 | 1.44 | 0.26 | 1.27 | 0.21 | 0 |
| 33 | *Artemisia campestris* | 21.69 | 21.26 | 1.02 | 3.68 | 21.21 | 0.17 | 0.57 | 0.38 | 1.53 | 0.41 | 1.23 | 0.34 | 0 |
| 34 | *Kaschagaria brachanthemoides* | 23.26 | 22.09 | 1.06 | 3.93 | 21.01 | 0.19 | 0.55 | 0.44 | 1.25 | 0 | 1.75 | 0 | 0.47 |
| 35 | *Ajania pallasiana* | 35.16 | 35.92 | 0.98 | 10.23 | 38.31 | 0.27 | 4.41 | 3.47 | 1.29 | 0 | 7.84 | 0 | 0.39 |
| 36 | *Chrysanthemum indicum* | 33.54 | 34.42 | 0.98 | 8.65 | 34.82 | 0.25 | 2.94 | 3.59 | 0.82 | 0 | 7.11 | 0 | 0.37 |

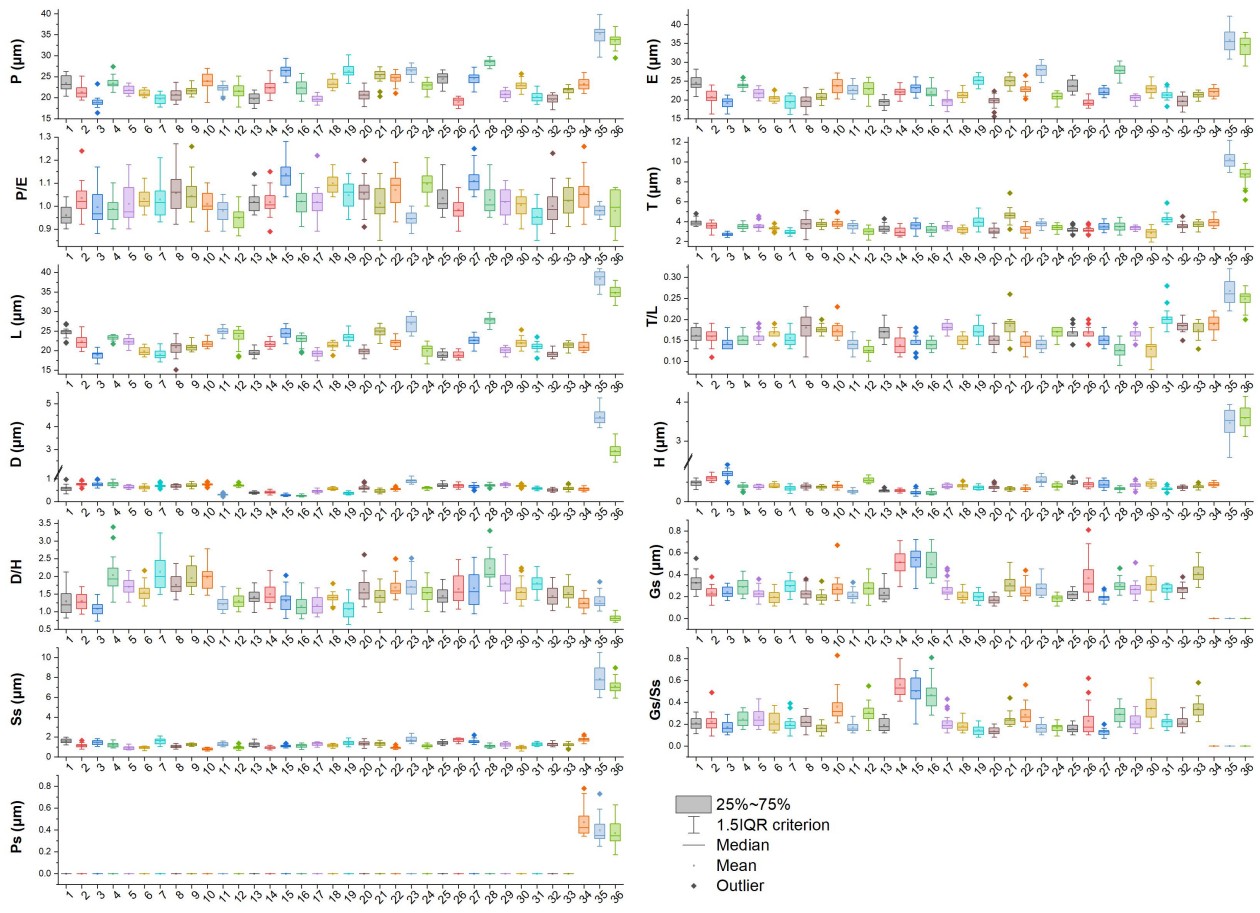

276

**Figure 15.** Boxplots of 36 sampled taxa, showing the variations in pollen morphological traits.
1. *Artemisia cana*; 2. *Artemisia tridentata*; 3. *Artemisia californica*; 4. *Artemisia indica*; 5. *Artemisia argyi*; 6. *Artemisia mongolica*; 7. *Artemisia vulgaris*; 8. *Artemisia selengensis*; 9. *Artemisia ludoviciana*; 10. *Artemisia roxburghiana*; 11. *Artemisia rutifolia*; 12. *Artemisia chinensis*; 13. *Artemisia kurramensis*; 14. *Artemisia compactum*; 15. *Artemisia maritima;* 16. *Artemisia aralensis*; 17. *Artemisia annua*; 18. *Artemisia freyniana*; 19. *Artemisia stechmanniana*; 20. *Artemisia pontica*; 21. *Artemisia frigida*; 22. *Artemisia rupestris*; 23. *Artemisia sericea*; 24. *Artemisia absinthium*; 25. *Artemisia abrotanum*; 26. *Artemisia blepharolepis*; 27. *Artemisia norvegica*; 28. *Artemisia tanacetifolia*; 29. *Artemisia tournefortiana*; 30. *Artemisia dracunculus*; 31. *Artemisia japonica*; 32. *Artemisia capillaris*; 33. *Artemisia campestris*; 34. *Kaschagaria brachanthemoides*; 35. *Ajania pallasiana*; 36. *Chrysanthemum indicum*.

## 3.3 The source plant occurrences

The source plant distributions in global terrestrial biomes of 36 sampled species are shown in Fig. 16. In *Artemisia*, some species have worldwide distributions, such as *A. vulgaris* (Fig. 16-7), *A. absinthium* (Fig. 16-24), and *A. campestris* (Fig. 16-33); a few taxa are limited to East Asia, such as *A. roxburghiana* (Fig. 16-10) and *A. blepharolepis* (Fig. 16-26), while others have narrow and isolated distributions in deserts and xeric shrublands of Central Asia, e.g. *A. kurramensis* (Fig. 16-13) and *A. aralensis* (Fig. 16-16). In outgroups of *Artemisia*, *Kaschagaria brachanthemoides* is also confined to deserts and xeric shrublands of Central Asia (Fig. 16-34), while *Ajania pallasiana* lives in forests of East Asia (Fig. 16-35).

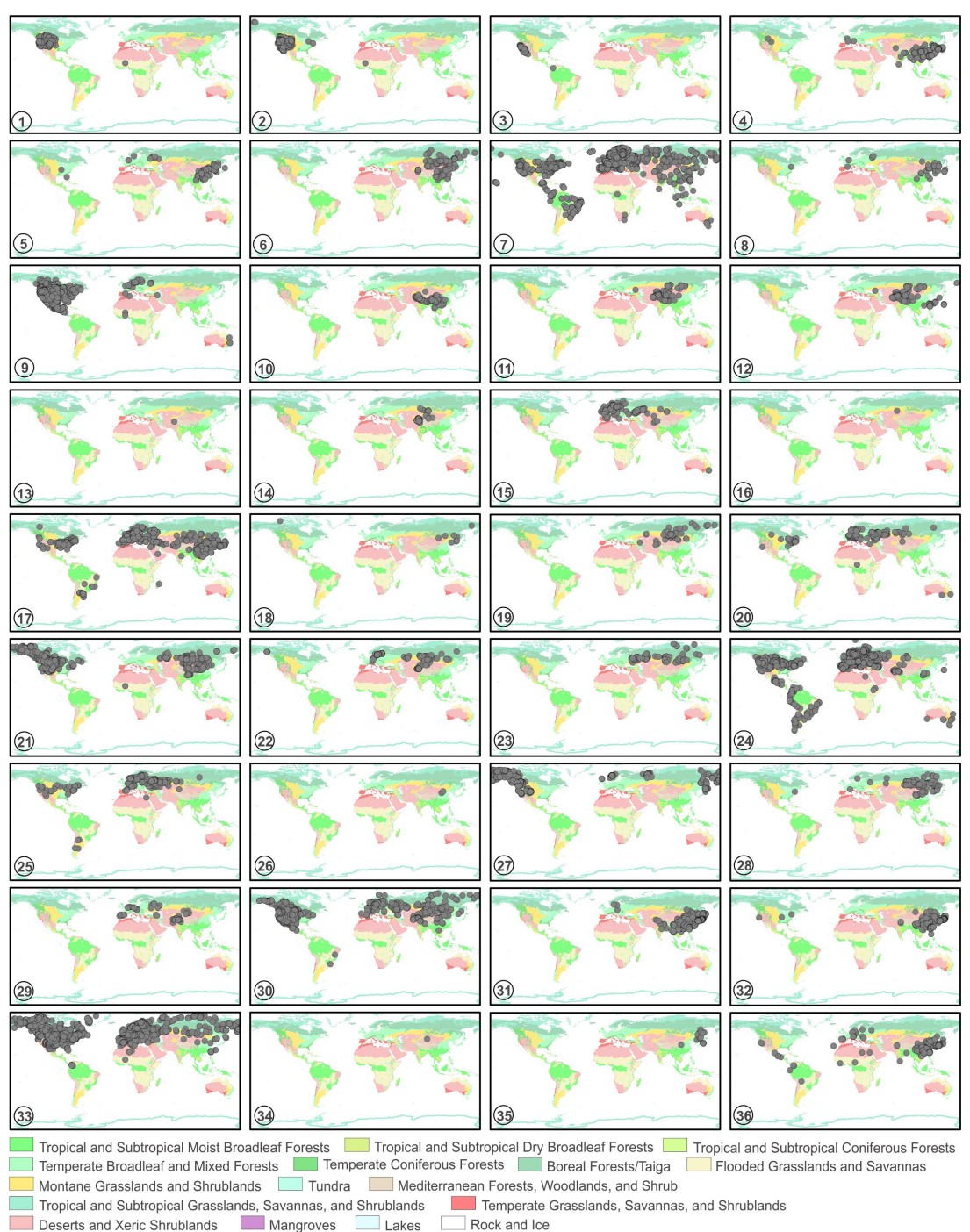

295

**Figure 16.** The global distribution maps of 36 sampled taxa in terrestrial biomes (modified from Olson et al., 2001).

1. *Artemisia cana*; 2. *Artemisia tridentata*; 3. *Artemisia californica*; 4. *Artemisia indica*; 5. *Artemisia argyi*; 6. *Artemisia mongolica*; 7. *Artemisia vulgaris*; 8. *Artemisia selengensis*; 9. *Artemisia ludoviciana*; 10. *Artemisia roxburghiana*; 11. *Artemisia rutifolia*; 12. *Artemisia chinensis*; 13. *Artemisia kurramensis*; 14. *Artemisia compactum*; 15. *Artemisia maritima;* 16. *Artemisia aralensis*; 17. *Artemisia annua*; 18. *Artemisia freyniana*; 19. *Artemisia stechmanniana*; 20. *Artemisia pontica*; 21. *Artemisia frigida*; 22. *Artemisia rupestris*; 23. *Artemisia sericea*; 24. *Artemisia absinthium*; 25. *Artemisia abrotanum*; 26. *Artemisia blepharolepis*; 27. *Artemisia norvegica*; 28. *Artemisia tanacetifolia*; 29. *Artemisia tournefortiana*; 30. *Artemisia dracunculus*; 31. *Artemisia japonica*; 32. *Artemisia capillaris*; 33. *Artemisia campestris*; 34. *Kaschagaria brachanthemoides*; 35. *Ajania pallasiana*; 36. *Chrysanthemum indicum*.

## 4 Potential use of the *Artemisia* pollen datasets

### 4.1 The pollen classification of *Artemisia*

The pollen grains of Anthemideae and Asteraceae under LM could be simply divided into *Artemisia* pollen type (Figs. 3-13, 14a, Appendix A) with indistinct and short spinules and *Anthemis* pollen type such as *Chrysanthemum indicum* and *Ajania pallasiana* (Figs. 14b-c, Appendix A) with distinct and long spines on pollen exine ornamentation (Wodehouse, 1926; Stix, 1960; Chen, 1987; Chen and Zhang, 1991; Martín et al., 2001; Martín et al., 2003; Sanz et al., 2008; Blackmore et al., 2009; Vallès et al., 2011). *Artemisia* pollen grains are difficult to separate from those of other related genera with *Artemisia* pollen type such as *Kaschgaria brachanthemoides* (Figs. 14a1-2, Appendix A), *Elachanthemum*, *Ajaniopsis*, *Filifolium*, and *Neopallasia* (Chen and Zhang, 1991) under LM due to their great similarity in pollen exine ornamentation and colporate patterns (Chen, 1987; Martín et al., 2001; Martín et al., 2003; Vallès et al., 2011). Furthermore, Sing and Joshi (1969) questioned the feasibility of recognizing pollen types under LM in the highly uniform pollen of *Artemisia*. Later, SEM made it possible to subdivide the pollen of *Artemisia* and those of other related genera within the *Artemisia* pollen type using pollen exine ultrastructure characters (Chen, 1987; Chen and Zhang, 1991; Sun and Xu, 1997; Jiang et al., 2005; Ghahraman et al., 2007; Shan et al., 2007; Hayat et al., 2009; Hayat et al., 2010; Hussain et al., 2019).

Hierarchical cluster analysis (Fig. 17a) revealed that the pollen morphological traits (P/E, H, D, D/H, Ss, Gs, Gs/Ss, and Ps) of *Artemisia* and its outgroups were divided into Clade A with perforations and without granules (Figs. 13a5-6, b5-6, c5-6) and Clade B with granules and without perforations (Figs. 3-13a5-6, b5-6, c5-6) on the pollen exine under SEM.

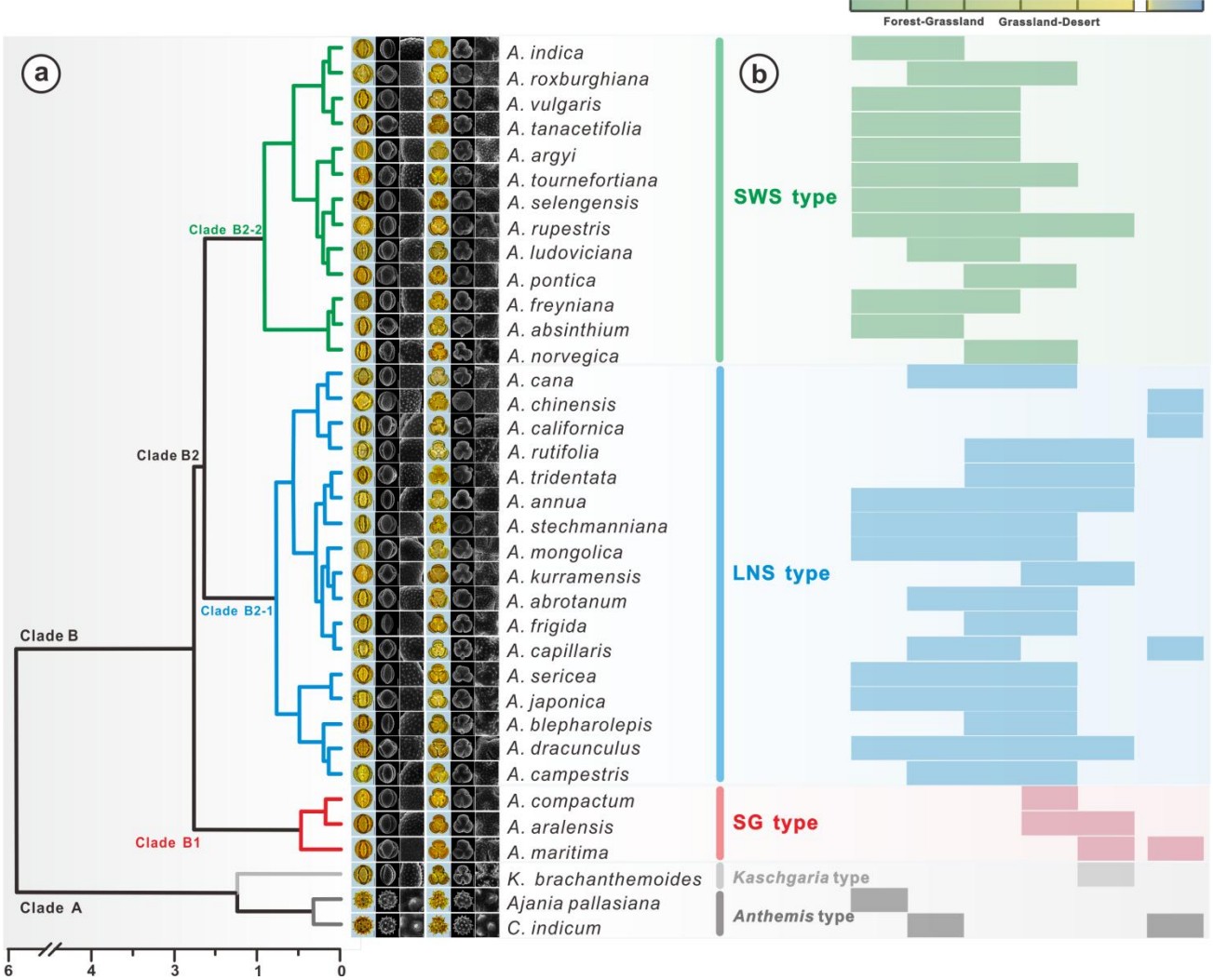

**Figure 17.** Hierarchical cluster analysis, showing the dendrogram for pollen types from *Artemisia* and outgroups (a) and the habitat ranges of 36 representative species (b, Tutin et al., 1976; Zhang, 2007; Ling et al., 2011).

In addition, Clade A, as the outgroup of *Artemisia*, includes *Anthemis* type (*Chrysanthemum indicum* and *Ajania pallasiana*) with prominent spines on pollen exine under LM, and *Kaschgaria* type (*Kaschgaria brachanthemoides*) with spinules on pollen exine (Figs. 14a, 17a). Clade B comprises three pollen types from three branches of *Artemisia* (Fig. 17a), i.e., SG type (short and wide spinule pollen type, Clade B1), LNS type (long and narrow spinule pollen type, Clade B2-1), and SG type (sparse granule pollen type, Clade B2-2).

Eight pollen morphological traits (P/E, H, D, D/H, Ss, Gs, Gs/Ss, and Ps) were selected for the principal component analysis (PCA) of 36 taxa of *Artemisia* and its outgroups (Fig. 18) and grouped according to the five clades of the cluster analysis, i.e. the five pollen types (Fig. 17a). The results reveal that *Artemisia* pollen morphology differs significantly from that of the outgroups, and that three *Artemisia* pollen types could be distinguished.

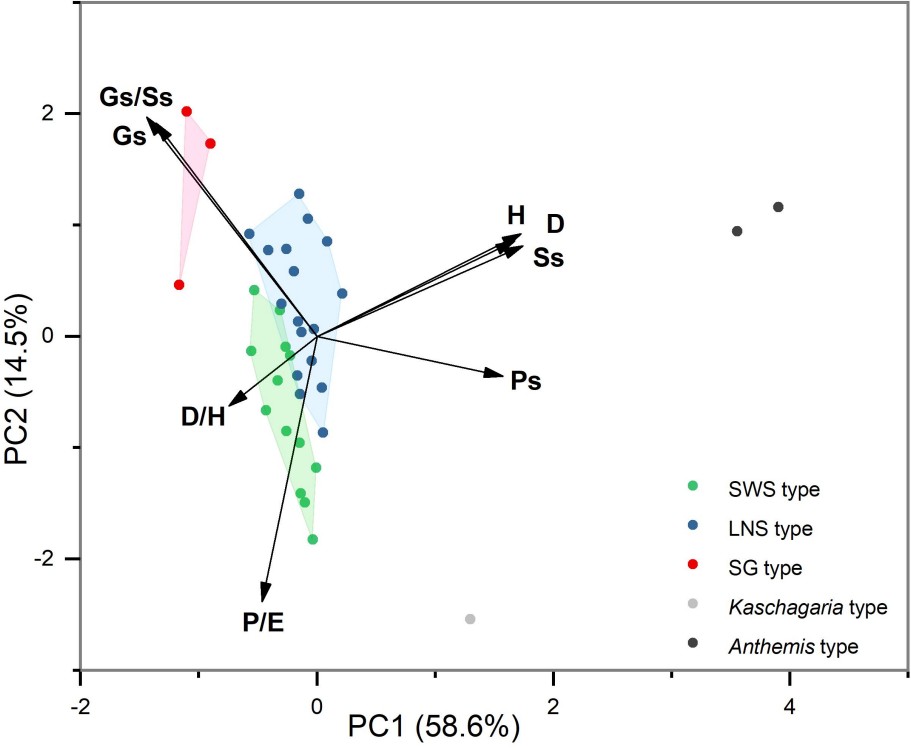

**Figure 18.** Principal component analysis of 36 taxa of *Artemisia* and its outgroups.

Nine characteristics of *Artemisia* pollen could partially explain the differences between these 3 pollen types (Fig. 19). P/E (the length of polar axis/the length of equatorial axis) in LNS types (0.93-1.06) are significantly different (ANOVA P < 0.001) from both SWS (0.97-1.12) and SG (0.98-1.14), so could be used to identify the LNS type. D/H (diameter of spinule base/spinule height) in the SWS type differ significantly (ANOVA P < 0.001) from both LNS and SG types. The variation range of D/H is 1.38-2.23 in the SWS type, 1.07-1.75 in the LNS type, and 0.98-1.66 in the SG type, indicating that the SWS pollen type is distinguished by short and wide spinules. Gs/Ss (granule spacing/spinule spacing) in the SG type was higher than those of the SWS and LNS types (ANOVA P < 0.001), which distinguished the SG type from the other two types. Moreover, the SG type is characterized by sparse granules with the variation range of Gs/Ss spanning 0.37-0.64, while the SWS and LNS types show much denser granules whose Gs/Ss are mainly below 0.35.

Within the new *Artemisia* pollen classification (Fig. 17a, Key), the SWS type represents a type of pollen with short and wide spinules (D/H > 1.81) and dense granules (Figs. 17a, 19). The LNS type represents a type of pollen with long and narrow spinules (D/H < 1.38) and dense granules (Figs. 17a, 19). The SG type is characterized by sparse granules (Gs/Ss > 0.37) and small, long, and narrow spinules (Figs. 17a, 19).

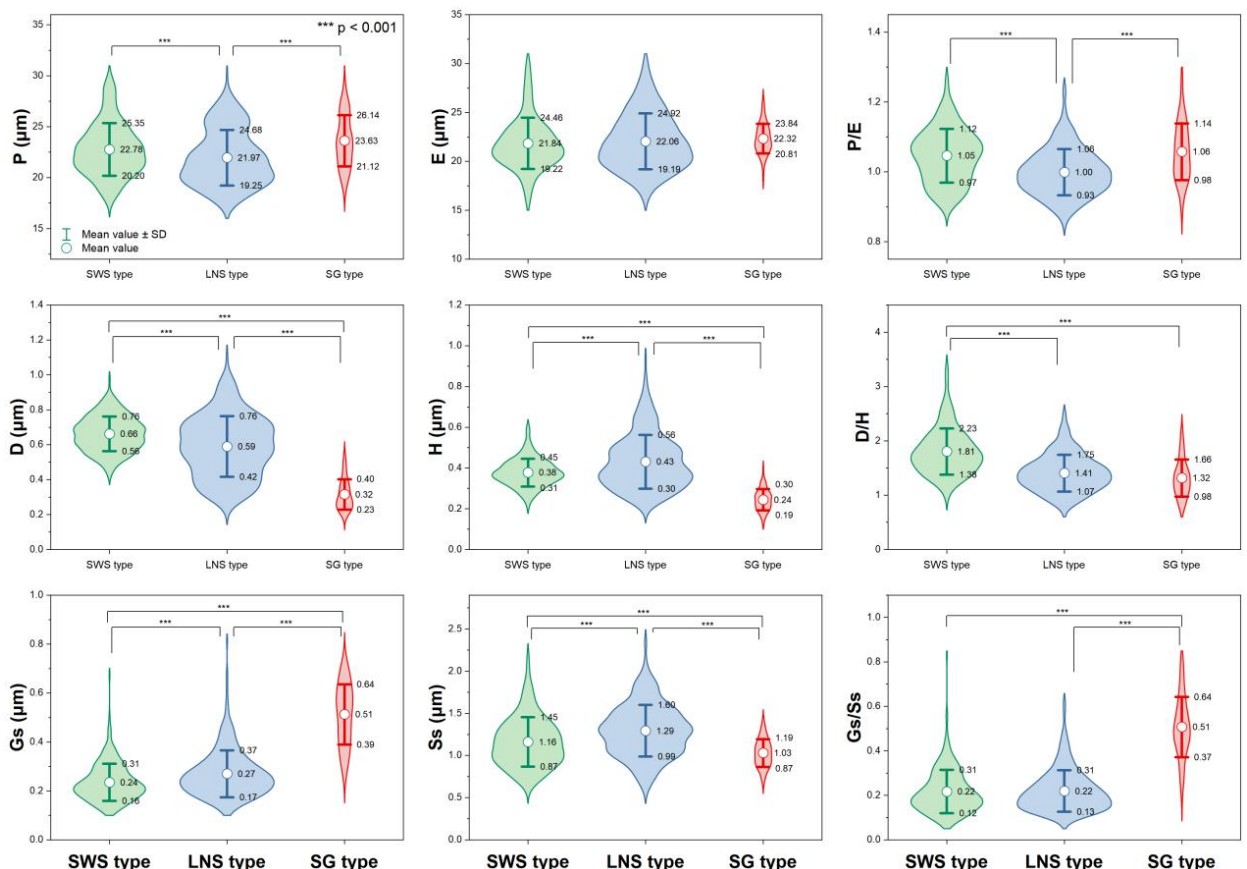

**Figure 19.** Violin diagrams of three pollen types from *Artemisia*, showing the variations (M ± SD) in nine pollen characters (P: length of polar axis; E: length of equatorial axis; D: diameter of spinule base; H: spinule height; Gs: granule spacing; Ss: spinule spacing; Ps: perforation spacing). Asterisks indicate statistically significant differences (p < 0.001).

## 4.2 Testing the pollen intraspecific variability within *Artemisia*

Evidence shows that the pollen morphology in *Artemisia* is highly uniform under LM without discrimination (Wodehouse, 1926; Sing and Joshi, 1969; Ling, 1982; Chen, 1987; Wang et al., 1995), which might suggest that statistical analyses of the intraspecific morphological variation of pollen under the LM are limited or meaningless. Right now, the SEM technique has made it possible to subdivide *Artemisia* pollen into different types using pollen exine ultrastructure characters (Chen, 1987; Chen and Zhang, 1991; Sun and Xu, 1997; Jiang et al., 2005; Ghahraman et al., 2007; Shan et al., 2007; Hayat et al., 2009; Hayat et al., 2010; Hussain et al., 2019).

In order to test the intraspecific variability of pollen exine ultrastructure traits, we selected one species respectively from the three pollen types corresponding to the three morphological clades of *Artemisia* pollen, i.e. *Artemisia vulgaris* (SWS type), *Artemisia annua* (LNS type), and *Artemisia maritima* (SG type), and

sampled five specimens of each species (Table B2). Six pollen traits, i.e. D, H, D/H, Gs, Ss, and Gs/Ss, were
counted and analysed under SEM to test for intraspecific variability of pollen exine ultrastructure traits.
The test showed that it was feasible to use stable D/H and Gs/Ss for pollen type classification of
*Artemisia* because 1) D/H and Gs/Ss were stable within species (Figure 20, Table 2) for the pollen
classification; 2) D, H, Gs, and Ss were variable as size values, e.g. these four traits were significantly
different within species in both *A. vulgaris* and *A. annua*, while D, H, and Ss were significantly different
within species in *A. maritima* (Figure 20, Table 2). There was evidence showing that size values such as pollen
exine ultrastructure size were often variable within species due to their genetic divergence, various habitats,
and different experimental treatments (Mo et al., 1997; Zhao and Yao, 1999; Zhang and Qian, 2011).

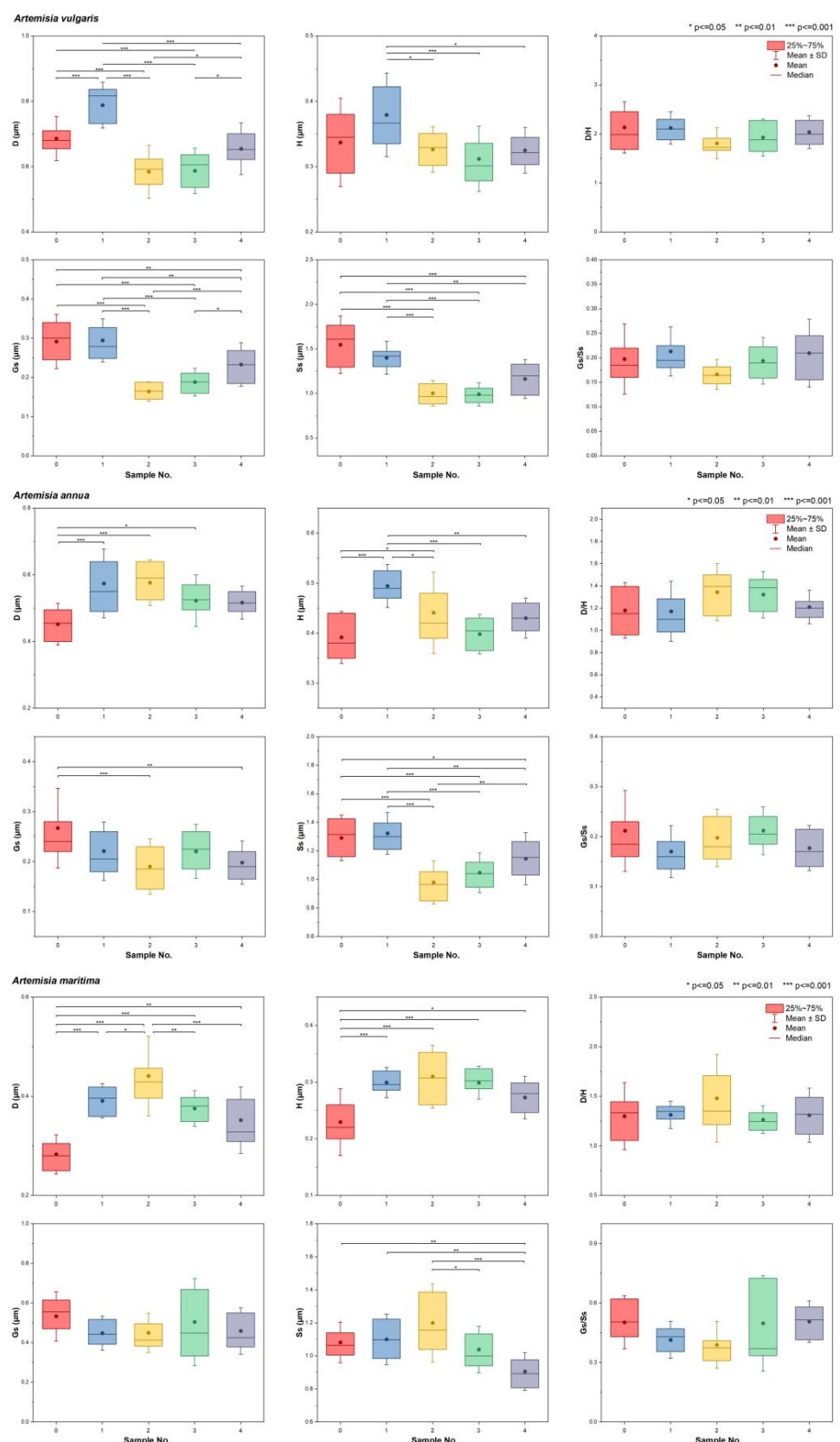


**Figure 20.** Boxplots of intraspecific pollen exine ultrastructure characters from three species of *Artemisia*, showing the variations (M ± SD) in six pollen characters (D: diameter of spinule base; H: spinule height; Gs: granule spacing; Ss: spinule spacing; Ps: perforation spacing). Asterisks indicate statistically significant differences (*p <= 0.05, **p <= 0.01, ***p < 0.001).

**Table 2.** The results of ANOVA for intraspecific variability in pollen exine ultrastructure characters among
three representative species.

| Pollen exine ultrastructure characters | SWS type | LNS type | SG type |
| --- | --- | --- | --- |
| | *Artemisia vulgaris* | *Artemisia annua* | *Artemisia maritima* |
| **D (μm)** | significant | significant | significant |
| **H (μm)** | significant | significant | significant |
| **D/H** | non-significant | non-significant | non-significant |
| **Gs (μm)** | significant | significant | significant |
| **Ss (μm)** | significant | significant | significant |
| **Gs/Ss** | non-significant | non-significant | non-significant |

**Key to 3 pollen types of *Artemisia* and 3 outgroups**
1. Pollen exine with perforations and without granules under SEM …………………………………………2
1. Pollen exine with granules and without perforations under SEM …………………………………………3
2. Distinct and long spines on pollen exine, with H > 3 μm……………………………………*Anthemis* type
2. Indistinct and short spinules on pollen exine, with H < 1μm……………………………… *Kaschgaria* type
3. Pollen exine with sparse granules and Gs/Ss ≥ 0.37 under SEM ……………………………………SG type
3. Pollen exine with dense granules and Gs/Ss ≤ 0.31 under SEM…………………………………………4
4. Pollen exine with D/H < 1.38 under SEM…………………………………………………………LNS type
4. Pollen exine with D/H ≥ 1.38 under SEM…………………………………………………………SWS type
**4.3 The ecological implications of *Artemisia* pollen types**
Plotting the distribution data of 33 species from 9 main branches of *Artemisia* constrained by the phylogenetic
framework (Fig. 1) onto the global terrestrial biomes (Fig. 21), we noticed that the genus is widely distributed
from forest to grassland, desert, and saline habitats (Figs. 16, 17b, 21). Furthermore, different species of
*Artemisia* with SWS pollen type (Fig. 21a) and LNS type (Fig. 21b) have a rather wide distribution with
severely overlapping ranges while those with SG type (Fig. 21c) have narrow and isolated distributions.

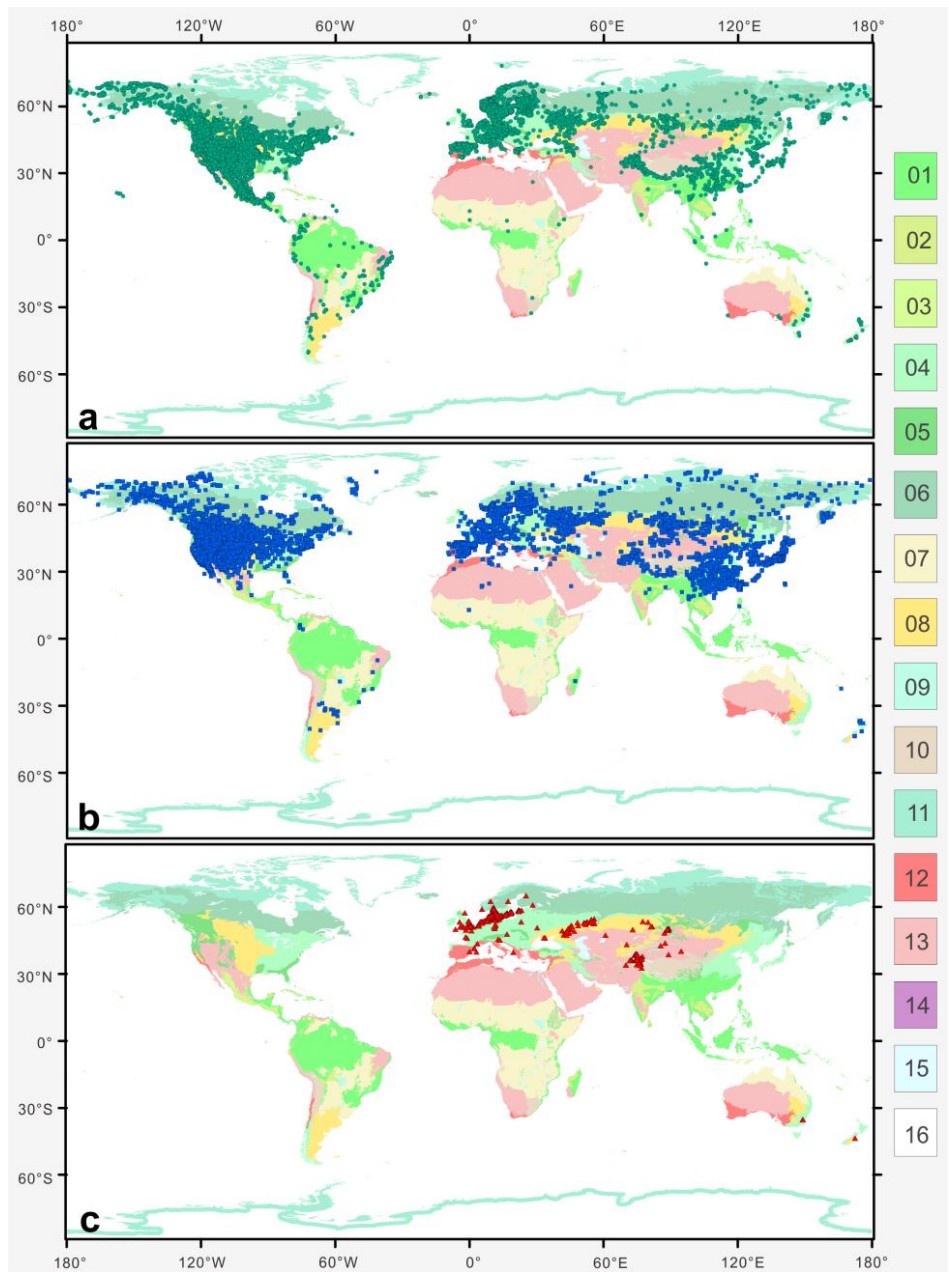

**Figure 21.** The global distribution pattern of 3 *Artemisia* pollen types in terrestrial biomes (modified from Olson et al., 2001). a. SG type; b. LNS type; c. SWS type.

14 terrestrial biomes: 01. Tropical and Subtropical Moist Broadleaf Forests; 02. Tropical and Subtropical Dry Broadleaf Forests; 03. Tropical and Subtropical Coniferous Forests; 04. Temperate Broadleaf and Mixed Forests; 05. Temperate Coniferous Forests; 06. Boreal Forests/Taiga; 07. Flooded Grasslands and Savannas; 08. Montane Grasslands and Shrublands; 09. Tundra; 10: Mediterranean Forests, Woodlands, and Shrub; 11. Tropical and Subtropical Grasslands, Savannas, and Shrublands; 12. Temperate Grasslands, Savannas, and Shrublands; 13. Deserts and Xeric Shrublands; 14. Mangroves; 15. Lakes; 16. Rock and Ice.

The ecological implications of *Artemisia* pollen types mentioned above fall into four categories. (i) *Artemisia* with the SG pollen type all belong to the subg. *Seriphidium*, which generally grows in dry habitats ranging from grassland desert to desert and coastal saline-alkaline environments, with their distribution largely limited to Eurasia and growing at low altitude (Figs. 17b, 21c, 22). (ii) The habitats of *Artemisia* with

LNS pollen type have a global distribution and occur in forest, grassland and desert, and even coastal areas
(Figs. 17b, 21b, 22), with the highest mean annual temperature (MAT). Hence, the LNS pollen type is a
generalist. (iii) *Artemisia* with SWS pollen type include Sect. *Artemisia* and its habitats range from forest to
desert, although most of the taxa are confined to humid environments from forest to grassland with a global
distribution and the highest mean annual precipitation (MAP, Figs. 17b, 21c, 22). (iv) If the SWS pollen type
and the SG pollen type appear together, the range of vegetation types could be reduced to grassland desert and
desert through niche coexistence (Fig. 17b).

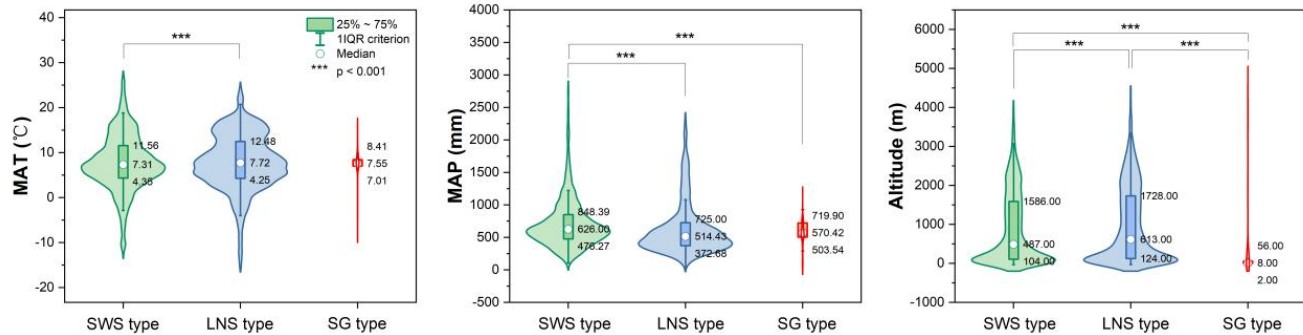


**Figure 22.** Violin diagrams of three pollen types from *Artemisia*, showing the variations (25%-75%) in MAT, MAP, and altitude. Asterisks indicate statistically significant differences (p < 0.001).

In addition, we noticed that *Kaschgaria brachanthemoides* as an outgroup of *Artemisia* lives in dry
mountain valleys or dry riverbeds of Northwest China (Toksun) and Kazakhstan, with highly characteristic
pollen (Fig. 14a), narrow habitats (Fig. 17b), and regional distribution (Fig. 16-34) and has the potential to
indicate some specific habitats.
**5 Data availability**
Pollen datasets (Table 3) including pollen photographs under LM and SEM, statistical data of pollen
morphological traits, and their source plant distribution for each species are available at Zenodo
(https://doi.org/10.5281/zenodo.6900308; Lu et al., 2022).
**Table 3.** *Artemisia* pollen datasets in this study.

| Data type | Data format | Data acquisition | Data accessibility |
| --- | --- | --- | --- |

| | | | |
|---|---|---|---|
| The phylogenetic framework of *Artemisia* pollen sampling. | .png | Literature survey (modified from Malik et al., 2017). | This article |
| A voucher specimen list of 36 representative species. | .doc | Pollen samples were obtained from PE herbarium at the Institute of Botany, Chinese Academy of Sciences. | |
| 12 illustrations of pollen grains and the habitats of their source plants. | .png | Habitat photos from online sources (Appendix Table A). | |
| 4018 original pollen photographs (3205 under LM, 813 under SEM). | .jpg | Pollen samples were acetolyzed by the standard method and fixed in glycerine jelly. The pollen grains were photographed under LM and SEM using standard procedures. | |
| 9360 pollen morphological trait measurements of 36 representative species. | .xlsx | Statistical data of pollen morphological traits were measured by standard methods. | |
| 1800 pollen morphological trait measurements for testing the pollen intraspecific variability within *Artemisia*. | .xlsx | Statistical data of pollen morphological traits were measured by standard methods. | Zenodo (https://doi.org/10.5281/zenodo.6900308; Lu et al., 2022) |
| 30858 source plant occurrence information, and corresponding environmental factors including altitude and 19 climate parameters. | .xlsx | Their source plant distribution coordinates were obtained from GBIF (https://doi.org/10.15468/dl.596xd9). The corresponding environmental factors of these coordinates were obtained from WorldClim (https://www.worldclim.org/) with a spatial resolution of 30 seconds between 1970-2000. | |

**6 Summary**
To cover the maximum range of *Artemisia* pollen morphological variation, we provide a pollen dataset of 36
species from 9 clades and 3 outgroups of *Artemisia* constrained by the phylogenetic framework, containing
high-quality pollen photographs under LM and SEM, statistical data of pollen morphological traits, together

with their source plant distribution, and corresponding environmental factors. Here, we attempt to decipher the underlying causes of the long-standing disagreement in the palynological community on the correlation between *Artemisia* pollen and aridity by recognizing the different ecological implications of *Artemisia* pollen types.

This dataset should work well for identifying and classifying *Artemisia* pollen from Neogene and Quaternary sediments. While *Artemisia* pollen grains are uniform in morphology under LM, different types can be recognized under SEM. So, the single-grain technique for picking out fossil pollen grains and photographing the same grains under LM and SEM should provide valuable insights in the diversity of fossil *Artemisia* (Ferguson et al., 2007; Grímsson et al., 2011; Grímsson et al., 2012; Halbritter et al., 2018). Furthermore, those *Artemisia* pollen grains could then be compared with the rich photographs from this dataset, and together with the key provided here, possibly attributed to one of the three *Artemisia* pollen types, which in turn may provide a link to the different habitat ranges.

However, the application of this dataset probably may not work well for the Palaeogene, as 1) *Artemisia* might have originated in the Palaeocene, although there is no evidence for a specific location or time interval of its origin (e.g. Ling 1982; Wang 2004; Miao 2011); 2) both the lack of macrofossils of *Artemisia* and the strong pollen similarity between *Artemisia* and its closely related taxa under LM might lead to confusion and more uncertainty in tracing the origin of *Artemisia*. On the other hand, the present dataset provides a potential morphological tool to distinguish *Artemisia* pollen grains from those of its related taxa at the SEM level and may shed light on the origin of this genus in the Palaeogene.

Moreover, these pollen photographs also have potential and the possibility to be used for deep learning research. We are attempting to automatically identify pollen images using pollen assemblages from the eastern Central Asian desert as an example with deep convolutional neural network (DCNN) of artificial intelligence. Pollen images of the many species of *Artemisia* provided here, and the increasing number of intraspecific replications in the future, will all serve for projected image identification research.

Finally and most importantly, the *Artemisia* pollen dataset as designed is open and expandable for new pollen data from *Artemisia* worldwide in order to better serve the global environment assessment and refined reconstruction of vegetation in the geological past as a basis or blueprint for other overarching statistical analyses on pollen morphology.

**Appendix A**

**Text A1**

Pollen morphological descriptions of 36 representative species from 9 clades of *Artemisia* and 3 outgroups.

Pollen morphology of *Artemisia*: pollen grains oblate, spherical, or ellipsoidal; apertures tricolporate; almost circular in equatorial view and trilobate circular in polar view; the exine near the colpi gradually thinned; the exine has an obvious double structure of inner and outer layers where the outer is thicker than the inner under LM; the exine ornamentation is psilate (LM), spinulate and granule (SEM).

**1. *Artemisia cana* (Table 1, Figs. 3a, 15)**

Pollen grains spheroidal or oblate. Almost circular in equatorial view and trilobate circular in polar view. Apertures tricolporate. The exine near the colpi gradually thinned. Polar length (P) = 23.46 ± 1.76 μm (M ± SD), equatorial width (E) = 24.50 ± 2.13 μm (M ± SD), P/E = 0.96 ± 0.04 (M ± SD), Exine thickness (T) = 3.91 ± 0.36 μm (M ± SD), Pollen length (L) = 24.58 ± 1.24 μm (M ± SD), T/L = 0.16 ± 0.02. The exine ornamentation is psilate (LM), spinulate (SEM). Under SEM, diameter of spinule base (D) = 0.58 ± 0.13 μm (M ± SD), spinule height (H) = 0.46 ± 0.08 μm (M ± SD), D/H = 1.28 ± 0.38 (M ± SD), granule spacing (Gs) = 0.33 ± 0.08 μm (M ± SD), spinule spacing (Ss) = 1.60 ± 0.22 μm (M ± SD), Gs/Ss = 0.21 ± 0.06 (M ± SD).

Habitat: grasslands, gravel soils, mountain meadows, stream banks; Wet mountain meadows, stream banks, rocky areas with late-lying snows.

**2. *Artemisia tridentata* (Table 1, Figs. 3b, 15)**

Pollen grains prolate or spheroidal. Almost circular in equatorial view and trilobate circular in polar view. Apertures tricolporate. The exine near the colpi gradually thinned. P = 21.36 ± 1.54 μm, E = 20.69 ± 1.85 μm, P/E = 1.04 ± 0.07, T = 3.55 ± 0.41 μm, L = 22.35 ± 1.90 μm, T/L = 0.16 ± 0.02. The exine ornamentation is psilate (LM), spinulate (SEM). Under SEM, D = 0.76 ± 0.08 μm, H = 0.60 ± 0.08 μm, D/H = 1.30 ± 0.23, Gs = 0.24 ± 0.06 μm, Ss = 1.12 ± 0.22 μm, Gs/Ss = 0.22 ± 0.08.

Habitat: mountains, grasslands, and meadows of western North America. Arid and semi-arid, desert, or semi-desert areas of the growing shrub or semi-shrub environment.

**3. *Artemisia californica* (Table 1, Figs. 3c, 15)**

Pollen grains prolate or spheroidal or oblate. Almost circular in equatorial view and trilobate circular in polar view. Apertures tricolporate. The exine near the colpi gradually thinned. P = 18.94 ± 1.30 μm, E = 19.13 ± 1.43 μm, P/E = 0.99 ± 0.08, T = 2.70 ± 0.16 μm, L = 18.85 ± 1.12 μm, T/L = 0.14 ± 0.01. The exine ornamentation is psilate (LM), spinulate (SEM). Under SEM, D = 0.75 ± 0.11 μm, H = 0.71 ± 0.10 μm, D/H = 1.08 ± 0.20, Gs = 0.24 ± 0.05 μm, Ss = 1.45 ± 0.23 μm, Gs/Ss = 0.17 ± 0.05.

Habitat: coastal scrub, dry foothills.
**4. *Artemisia indica* (Table 1, Figs. 4a, 15)**
Pollen grains spheroidal or oblate. Almost circular in equatorial view and trilobate circular in polar view.
Apertures tricolporate. The exine near the colpi gradually thinned. $P = 23.47 \pm 1.39$ μm, $E = 23.81 \pm 0.86$ μm,
$P/E = 0.99 \pm 0.06$, $T = 3.50 \pm 0.27$ μm, $L = 23.31 \pm 0.61$ μm, $T/L = 0.15 \pm 0.01$. The exine ornamentation is
psilate (LM), spinulate (SEM). Under SEM, $D = 0.76 \pm 0.10$ μm, $H = 0.39 \pm 0.06$ μm, $D/H = 2.04 \pm 0.53$, Gs
$= 0.28 \pm 0.07$ μm, $Ss = 1.21 \pm 0.24$ μm, $Gs/Ss = 0.24 \pm 0.07$.
Habitat: roadsides, forest margins, slopes, shrublands; low elevations to 2000 m.
**5. *Artemisia argyi* (Table 1, Figs. 4b, 15)**
Pollen grains prolate or spheroidal. Almost circular in equatorial view and trilobate circular in polar view.
Apertures tricolporate. The exine near the colpi gradually thinned. $P = 21.80 \pm 1.00$ μm, $E = 21.67 \pm 1.27$ μm,
$P/E = 1.01 \pm 0.08$, $T = 3.55 \pm 0.40$ μm, $L = 22.24 \pm 1.13$ μm, $T/L = 0.16 \pm 0.01$. The exine ornamentation is
psilate (LM), spinulate (SEM). Under SEM, $D = 0.64 \pm 0.07$ μm, $H = 0.38 \pm 0.04$ μm, $D/H = 1.71 \pm 0.23$, Gs
$= 0.22 \pm 0.06$ μm, $Ss = 0.90 \pm 0.17$ μm, $Gs/Ss = 0.26 \pm 0.09$.
Habitat: waste places, roadsides, slopes, hills, steppes, forest steppes; low elevations to 1500 m.
**6. *Artemisia mongolica* (Table 1, Figs. 4c, 15)**
Pollen grains prolate or spheroidal. Almost circular in equatorial view and trilobate circular in polar view.
Apertures tricolporate. The exine near the colpi gradually thinned. $P = 21.05 \pm 0.82$ μm, $E = 20.42 \pm 1.01$ μm,
$P/E = 1.03 \pm 0.05$, $T = 3.29 \pm 0.19$ μm, $L = 19.78 \pm 0.99$ μm, $T/L = 0.17 \pm 0.01$. The exine ornamentation is
psilate (LM), spinulate (SEM). Under SEM, $D = 0.62 \pm 0.08$ μm, $H = 0.41 \pm 0.05$ μm, $D/H = 1.54 \pm 0.25$, Gs
$= 0.19 \pm 0.06$ μm, $Ss = 0.91 \pm 0.14$ μm, $Gs/Ss = 0.22 \pm 0.08$.
Habitat: slopes, shrublands, riverbanks, lakeshores, roadsides, steppes, forest steppes, dry valleys; low
elevations to 2000 m.
**7. *Artemisia vulgaris* (Table 1, Figs. 5a, 15)**
Pollen grains prolate or spheroidal. Almost circular in equatorial view and trilobate circular in polar view.
Apertures tricolporate. The exine near the colpi gradually thinned. $P = 19.72 \pm 1.25$ μm, $E = 19.29 \pm 1.82$ μm,
$P/E = 1.03 \pm 0.08$, $T = 2.92 \pm 0.23$ μm, $L = 18.94 \pm 1.09$ μm, $T/L = 0.16 \pm 0.02$. The exine ornamentation is
psilate (LM), spinulate (SEM). Under SEM, $D = 0.69 \pm 0.07$ μm, $H = 0.34 \pm 0.07$ μm, $D/H = 2.13 \pm 0.52$, Gs
$= 0.29 \pm 0.07$ μm, $Ss = 1.55 \pm 0.32$ μm, $Gs/Ss = 0.20 \pm 0.07$.
Habitat: roadsides, slopes, canyons, forest margins, forest steppes, subalpine steppes; 1500-3800 m.
**8. *Artemisia selengensis* (Table 1, Figs. 5b, 15)**
Pollen grains prolate or spheroidal. Almost circular in equatorial view and trilobate circular in polar view.
Apertures tricolporate. The exine near the colpi gradually thinned. $P = 20.67 \pm 1.57$ μm, $E = 19.68 \pm 1.94$ μm,
$P/E = 1.06 \pm 0.09$, $T = 3.72 \pm 0.72$ μm, $L = 20.80 \pm 2.21$ μm, $T/L = 0.18 \pm 0.03$. The exine ornamentation is
psilate (LM), spinulate (SEM). Under SEM, $D = 0.67 \pm 0.08$ μm, $H = 0.38 \pm 0.05$ μm, $D/H = 1.76 \pm 0.27$, Gs
$= 0.22 \pm 0.06$ μm, $Ss = 1.05 \pm 0.15$ μm, $Gs/Ss = 0.22 \pm 0.07$.
Habitat: riverbanks, lakeshores, humid areas, meadows, slopes, roadsides.
**9. *Artemisia ludoviciana* (Table 1, Figs. 5c, 15)**
Pollen grains prolate or spheroidal. Almost circular in equatorial view and trilobate circular in polar view.
Apertures tricolporate. The exine near the colpi gradually thinned. $P = 21.65 \pm 1.02$ μm, $E = 20.82 \pm 1.10$ μm,
$P/E = 1.04 \pm 0.08$, $T = 3.71 \pm 0.28$ μm, $L = 20.94 \pm 1.13$ μm, $T/L = 0.18 \pm 0.01$. The exine ornamentation is
psilate (LM), spinulate (SEM). Under SEM, $D = 0.70 \pm 0.08$ μm, $H = 0.37 \pm 0.04$ μm, $D/H = 1.94 \pm 0.31$, Gs
$= 0.20 \pm 0.05$ μm, $Ss = 1.23 \pm 0.13$ μm, $Gs/Ss = 0.16 \pm 0.04$.
Habitat: disturbed roadsides, open meadows, rocky slopes.
**10. *Artemisia roxburghiana* (Table 1, Figs. 6a, 15)**
Pollen grains prolate or spheroidal. Almost circular in equatorial view and trilobate circular in polar view.
Apertures tricolporate. The exine near the colpi gradually thinned. $P = 23.88 \pm 2.04$ μm, $E = 23.69 \pm 2.00$ μm,
$P/E = 1.01 \pm 0.06$, $T = 3.78 \pm 0.39$ μm, $L = 21.81 \pm 1.05$ μm, $T/L = 0.17 \pm 0.02$. The exine ornamentation is
psilate (LM), spinulate (SEM). Under SEM, $D = 0.76 \pm 0.07$ μm, $H = 0.39 \pm 0.06$ μm, $D/H = 1.96 \pm 0.37$, Gs
$= 0.28 \pm 0.11$ μm, $Ss = 0.79 \pm 0.11$ μm, $Gs/Ss = 0.36 \pm 0.14$.
Habitat: roadsides, slopes, dry canyons, grasslands, waste areas, terraces; 700-3900 m.
**11. *Artemisia rutifolia* (Table 1, Figs. 6b, 15)**
Pollen grains spheroidal or oblate. Almost circular in equatorial view and trilobate circular in polar view.
Apertures tricolporate. The exine near the colpi gradually thinned. $P = 22.22 \pm 1.10$ μm, $E = 22.70 \pm 1.37$ μm,
$P/E = 0.98 \pm 0.05$, $T = 3.53 \pm 0.37$ μm, $L = 24.93 \pm 1.05$ μm, $T/L = 0.14 \pm 0.01$. The exine ornamentation is
psilate (LM), spinulate (SEM). Under SEM, $D = 0.31 \pm 0.04$ μm, $H = 0.26 \pm 0.04$ μm, $D/H = 1.20 \pm 0.18$, Gs
$= 0.21 \pm 0.05$ μm, $Ss = 1.27 \pm 0.19$ μm, $Gs/Ss = 0.17 \pm 0.04$.
Habitat: hills, dry river valleys, basins, steppes, semideserts, stony desert; 1300-5000 m.
**12. *Artemisia chinensis* (Table 1, Figs. 6c, 15)**
Pollen grains spheroidal or oblate. Almost circular in equatorial view and trilobate circular in polar view.
Apertures tricolporate. The exine near the colpi gradually thinned. $P = 21.53 \pm 1.95$ μm, $E = 22.75 \pm 2.00$ μm,
$P/E = 0.95 \pm 0.05$, $T = 2.97 \pm 0.40$ μm, $L = 23.71 \pm 2.30$ μm, $T/L = 0.13 \pm 0.01$. The exine ornamentation is
psilate (LM), spinulate (SEM). Under SEM, D = 0.70 ± 0.05 μm, H = 0.55 ± 0.07 μm, D/H = 1.29 ± 0.19, Gs
= 0.27 ± 0.07 μm, Ss = 0.91 ± 0.17 μm, Gs/Ss = 0.31 ± 0.09.
Habitat: littoral plants found on raised coral outcrops.
**13. *Artemisia kurramensis* (Table 1, Figs. 7a, 15)**
Pollen grains spheroidal. Almost circular in equatorial view and trilobate circular in polar view. Apertures
tricolporate. The exine near the colpi gradually thinned. P = 19.71 ± 1.28 μm, E = 19.35 ± 1.02 μm, P/E = 1.02
± 0.05, T = 3.30 ± 0.38 μm, L = 19.44 ± 0.92 μm, T/L = 0.17 ± 0.02. The exine ornamentation is psilate (LM),
spinulate (SEM). Under SEM, D = 0.38 ± 0.04 μm, H = 0.27 ± 0.03 μm, D/H = 1.41 ± 0.21, Gs = 0.23 ± 0.07
μm, Ss = 1.25 ± 0.21 μm, Gs/Ss = 0.19 ± 0.06.
Habitat: foothills, mountain slopes, dry graveyards, field borders with sparse vegetation on gravelly, fine to
coarse sandy-clay soils.
**14. *Artemisia compactum* (Table 1, Figs. 7b, 15)**
Pollen grains spheroidal. Almost circular in equatorial view and trilobate circular in polar view. Apertures
tricolporate. The exine near the colpi gradually thinned. P = 22.33 ± 1.81 μm, E = 21.97 ± 1.23 μm, P/E = 1.02
± 0.06, T = 2.97 ± 0.43 μm, L = 21.67 ± 0.87 μm, T/L = 0.14 ± 0.02. The exine ornamentation is psilate (LM),
spinulate (SEM). Under SEM, D = 0.41 ± 0.07 μm, H = 0.28 ± 0.03 μm, D/H = 1.50 ± 0.33, Gs = 0.51 ± 0.12
μm, Ss = 0.92 ± 0.12 μm, Gs/Ss = 0.56 ± 0.12.
Habitat: rocky slopes, semi-deserts, from low elevations to sub-alpine areas.
**15. *Artemisia maritima* (Table 1, Figs. 7c, 15)**
Pollen grains prolate. Almost circular in equatorial view and trilobate circular in polar view. Apertures
tricolporate. The exine near the colpi gradually thinned. P = 26.24 ± 1.61 μm, E = 23.09 ± 1.43 μm, P/E = 1.14
± 0.06, T = 3.54 ± 0.44 μm, L = 24.42 ± 1.51 μm, T/L = 0.14 ± 0.02. The exine ornamentation is psilate (LM),
spinulate (SEM). Under SEM, D = 0.28 ± 0.04 μm, H = 0.23 ± 0.06 μm, D/H = 1.30 ± 0.34, Gs = 0.53 ± 0.12
μm, Ss = 1.08 ± 0.12 μm, Gs/Ss = 0.50 ± 0.13.
Habitat: saltmarsh, dry and calcareous hillsides, seashores, and dry saline or alkaline soils.
**16. *Artemisia aralensis* (Table 1, Figs. 8a, 15)**
Pollen grains prolate or spheroidal. Almost circular in equatorial view and trilobate circular in polar view.
Apertures tricolporate. The exine near the colpi gradually thinned. P = 22.32 ± 1.72 μm, E = 21.91 ± 1.63 μm,
P/E = 1.02 ± 0.06, T = 3.16 ± 0.36 μm, L = 22.76 ± 1.45 μm, T/L = 0.14 ± 0.01. The exine ornamentation is
psilate (LM), spinulate (SEM). Under SEM, D = 0.25 ± 0.04 μm, H = 0.22 ± 0.04 μm, D/H = 1.16 ± 0.28, Gs
= 0.50 ± 0.13 μm, Ss = 1.09 ± 0.18 μm, Gs/Ss = 0.46 ± 0.14.
Habitat: clayey, sandy loam, solonetzic soils.
**17.** *Artemisia annua* **(Table 1, Figs. 8b, 15)**
Pollen grains prolate or spheroidal. Almost circular in equatorial view and trilobate circular in polar view.
Apertures tricolporate. The exine near the colpi gradually thinned. $P = 19.71 \pm 0.84$ μm, $E = 19.45 \pm 1.32$ μm,
$P/E = 1.02 \pm 0.07$, $T = 3.45 \pm 0.25$ μm, $L = 19.20 \pm 0.92$ μm, $T/L = 0.18 \pm 0.01$. The exine ornamentation is
psilate (LM), spinulate (SEM). Under SEM, $D = 0.45 \pm 0.06$ μm, $H = 0.39 \pm 0.05$ μm, $D/H = 1.18 \pm 0.25$, $Gs$
$= 0.27 \pm 0.08$ μm, $Ss = 1.29 \pm 0.16$ μm, $Gs/Ss = 0.21 \pm 0.08$.
Habitat: hills, waysides, wastelands, outer forest margins, steppes, forest steppes, dry flood lands, terraces,
semidesert steppes, rocky slopes, roadsides, saline soils; 2000-3700 m.
**18.** *Artemisia freyniana* **(Table 1, Figs. 8c, 15)**
Pollen grains prolate. Almost circular in equatorial view and trilobate circular in polar view. Apertures
tricolporate. The exine near the colpi gradually thinned. $P = 23.39 \pm 1.21$ μm, $E = 21.30 \pm 1.07$ μm, $P/E = 1.10$
$\pm 0.04$, $T = 3.17 \pm 0.26$ μm, $L = 21.29 \pm 0.95$ μm, $T/L = 0.15 \pm 0.01$. The exine ornamentation is psilate (LM),
spinulate (SEM). Under SEM, $D = 0.56 \pm 0.05$ μm, $H = 0.40 \pm 0.06$ μm, $D/H = 1.40 \pm 0.15$, $Gs = 0.20 \pm 0.05$
μm, $Ss = 1.15 \pm 0.15$ μm, $Gs/Ss = 0.18 \pm 0.05$.
Habitat: steppes, slopes, dry river valleys, riverbanks, outer forest margins.
**19.** *Artemisia stechmanniana* **(Table 1, Figs. 9a, 15)**
Pollen grains prolate or spheroidal. Almost circular in equatorial view and trilobate circular in polar view.
Apertures tricolporate. The exine near the colpi gradually thinned. $P = 26.31 \pm 1.48$ μm, $E = 25.16 \pm 1.22$ μm,
$P/E = 1.05 \pm 0.07$, $T = 3.97 \pm 0.60$ μm, $L = 23.45 \pm 1.38$ μm, $T/L = 0.17 \pm 0.02$. The exine ornamentation is
psilate (LM), spinulate (SEM). Under SEM, $D = 0.37 \pm 0.05$ μm, $H = 0.35 \pm 0.05$ μm, $D/H = 1.07 \pm 0.25$, $Gs$
$= 0.19 \pm 0.04$ μm, $Ss = 1.40 \pm 0.24$ μm, $Gs/Ss = 0.14 \pm 0.04$.
Habitat: hillsides, roadsides, shrubland, and forest-steppe areas, and often becoming the dominant species or
main associated species of plant communities in some areas of mountainous sunny slopes.
**20.** *Artemisia pontica* **(Table 1, Figs. 9b, 15)**
Pollen grains prolate or spheroidal. Almost circular in equatorial view and trilobate circular in polar view.
Apertures tricolporate. The exine near the colpi gradually thinned. $P = 20.64 \pm 1.54$ μm, $E = 19.62 \pm 1.59$ μm,
$P/E = 1.05 \pm 0.07$, $T = 3.01 \pm 0.39$ μm, $L = 19.75 \pm 0.84$ μm, $T/L = 0.15 \pm 0.02$. The exine ornamentation is
psilate (LM), spinulate (SEM). Under SEM, $D = 0.60 \pm 0.11$ μm, $H = 0.37 \pm 0.06$ μm, $D/H = 1.63 \pm 0.37$, $Gs$
$= 0.17 \pm 0.04$ μm, $Ss = 1.32 \pm 0.27$ μm, $Gs/Ss = 0.13 \pm 0.04$.
Habitat: rocky slopes, dry valleys, steppes, hills; low to middle elevations.

**21. *Artemisia frigida* (Table 1, Figs. 9c, 15)**

Pollen grains prolate or spheroidal. Almost circular in equatorial view and trilobate circular in polar view. Apertures tricolporate. The exine near the colpi gradually thinned. $P = 25.11 \pm 1.75$ μm, $E = 24.90 \pm 1.48$ μm, $P/E = 1.01 \pm 0.07$, $T = 4.61 \pm 0.74$ μm, $L = 24.83 \pm 1.27$ μm, $T/L = 0.19 \pm 0.02$. The exine ornamentation is psilate (LM), spinulate (SEM). Under SEM, $D = 0.46 \pm 0.08$ μm, $H = 0.32 \pm 0.04$ μm, $D/H = 1.44 \pm 0.26$, $Gs = 0.31 \pm 0.08$ μm, $Ss = 1.30 \pm 0.18$ μm, $Gs/Ss = 0.24 \pm 0.06$.

Habitat: steppes, sub-alpine meadows, dry hillsides, stable dunes, dry waste areas; 1000-4000 m.

**22. *Artemisia rupestris* (Table 1, Figs. 10a, 15)**

Pollen grains prolate or spheroidal. Almost circular in equatorial view and trilobate circular in polar view. Apertures tricolporate. The exine near the colpi gradually thinned. $P = 24.45 \pm 1.41$ μm, $E = 22.92 \pm 1.40$ μm, $P/E = 1.07 \pm 0.08$, $T = 3.18 \pm 0.40$ μm, $L = 21.96 \pm 1.15$ μm, $T/L = 0.14 \pm 0.02$. The exine ornamentation is psilate (LM), spinulate (SEM). Under SEM, $D = 0.55 \pm 0.05$ μm, $H = 0.33 \pm 0.04$ μm, $D/H = 1.68 \pm 0.28$, $Gs = 0.25 \pm 0.07$ μm, $Ss = 0.91 \pm 0.11$ μm, $Gs/Ss = 0.28 \pm 0.09$.

Habitat: dry hills, desert or semidesert steppes, grassy marshlands, dry river valleys, riverbeds, scrub, forest margins.

**23. *Artemisia sericea* (Table 1, Figs. 10b, 15)**

Pollen grains spheroidal or oblate. Almost circular in equatorial view and trilobate circular in polar view. Apertures tricolporate. The exine near the colpi gradually thinned. $P = 26.31 \pm 1.31$ μm, $E = 27.90 \pm 1.67$ μm, $P/E = 0.94 \pm 0.03$, $T = 3.75 \pm 0.32$ μm, $L = 26.89 \pm 2.12$ μm, $T/L = 0.14 \pm 0.01$. The exine ornamentation is psilate (LM), spinulate (SEM). Under SEM, $D = 0.89 \pm 0.09$ μm, $H = 0.54 \pm 0.10$ μm, $D/H = 1.71 \pm 0.36$, $Gs = 0.28 \pm 0.07$ μm, $Ss = 1.74 \pm 0.31$ μm, $Gs/Ss = 0.16 \pm 0.05$.

Habitat: Forest margins, hills, steppes, canyons, waste areas.

**24. *Artemisia absinthium* (Table 1, Figs. 10c, 15)**

Pollen grains prolate. Almost circular in equatorial view and trilobate circular in polar view. Apertures tricolporate. The exine near the colpi gradually thinned. $P = 22.79 \pm 1.22$ μm, $E = 20.84 \pm 1.11$ μm, $P/E = 1.09 \pm 0.05$, $T = 3.39 \pm 0.31$ μm, $L = 19.92 \pm 1.74$ μm, $T/L = 0.17 \pm 0.01$. The exine ornamentation is psilate (LM), spinulate (SEM). Under SEM, $D = 0.59 \pm 0.05$ μm, $H = 0.40 \pm 0.06$ μm, $D/H = 1.52 \pm 0.25$, $Gs = 0.18 \pm 0.04$ μm, $Ss = 1.11 \pm 0.15$ μm, $Gs/Ss = 0.16 \pm 0.04$.

Habitat: hillsides, steppes, scrub, forest margins, often in locally moist situations; 1100-1500 m.

**25. *Artemisia abrotanum* (Table 1, Figs. 11a, 15)**

Pollen grains prolate or spheroidal. Almost circular in equatorial view and trilobate circular in polar view. Apertures tricolporate. The exine near the colpi gradually thinned. $P = 24.47 \pm 1.56$ μm, $E = 23.73 \pm 1.65$ μm, $P/E = 1.03 \pm 0.07$, $T = 3.15 \pm 0.28$ μm, $L = 18.82 \pm 0.81$ μm, $T/L = 0.17 \pm 0.01$. The exine ornamentation is psilate (LM), spinulate (SEM). Under SEM, $D = 0.72 \pm 0.10$ μm, $H = 0.51 \pm 0.05$ μm, $D/H = 1.44 \pm 0.25$, $Gs = 0.22 \pm 0.04$ μm, $Ss = 1.41 \pm 0.19$ μm, $Gs/Ss = 0.16 \pm 0.04$.

Habitat: the wasteland of western, southern, central, and southern Europe.

**26.  *Artemisia blepharolepis* (Table 1, Figs. 11b, 15)**

Pollen grains spheroidal. Almost circular in equatorial view and trilobate circular in polar view. Apertures tricolporate. The exine near the colpi gradually thinned. $P = 18.96 \pm 0.98$ μm, $E = 19.26 \pm 0.99$ μm, $P/E = 0.99 \pm 0.05$, $T = 3.15 \pm 0.28$ μm, $L = 18.82 \pm 0.81$ μm, $T/L = 0.17 \pm 0.01$. The exine ornamentation is psilate (LM), spinulate (SEM). Under SEM, $D = 0.69 \pm 0.09$ μm, $H = 0.44 \pm 0.07$ μm, $D/H = 1.64 \pm 0.44$, $Gs = 0.37 \pm 0.18$ μm, $Ss = 1.68 \pm 0.20$ μm, $Gs/Ss = 0.23 \pm 0.14$.

Habitat: low-altitude areas of dry slopes, grasslands, steppes, waste areas, roadsides, dunes near riverbanks.

**27.  *Artemisia norvegica* (Table 1, Figs. 11c, 15)**

Pollen grains prolate. Almost circular in equatorial view and trilobate circular in polar view. Apertures tricolporate. The exine near the colpi gradually thinned. $P = 24.51 \pm 1.40$ μm, $E = 22.11 \pm 1.05$ μm, $P/E = 1.11 \pm 0.06$, $T = 3.48 \pm 0.39$ μm, $L = 22.61 \pm 1.31$ μm, $T/L = 0.15 \pm 0.01$. The exine ornamentation is psilate (LM), spinulate (SEM). Under SEM, $D = 0.67 \pm 0.08$ μm, $H = 0.43 \pm 0.11$ μm, $D/H = 1.66 \pm 0.51$, $Gs = 0.19 \pm 0.03$ μm, $Ss = 1.56 \pm 0.24$ μm, $Gs/Ss = 0.12 \pm 0.03$.

Habitat: bare stony ground, Racomitrium heath, bouldery crests of solifluction terraces, and sometimes hollows between rocks.

**28.  *Artemisia tanacetifolia* (Table 1, Figs. 12a, 15)**

Pollen grains prolate or spheroidal. Almost circular in equatorial view and trilobate circular in polar view. Apertures tricolporate. The exine near the colpi gradually thinned. $P = 28.38 \pm 0.90$ μm, $E = 27.75 \pm 1.70$ μm, $P/E = 1.03 \pm 0.06$, $T = 3.46 \pm 0.47$ μm, $L = 27.63 \pm 1.06$ μm, $T/L = 0.13 \pm 0.02$. The exine ornamentation is psilate (LM), spinulate (SEM). Under SEM, $D = 0.71 \pm 0.06$ μm, $H = 0.32 \pm 0.04$ μm, $D/H = 2.23 \pm 0.40$, $Gs = 0.30 \pm 0.07$ μm, $Ss = 1.08 \pm 0.16$ μm, $Gs/Ss = 0.29 \pm 0.07$.

Habitat: middle and low-altitude areas of forest grasslands, grasslands, meadows, forest edges, open forests, salty grasslands, grass slopes, and brushwood.

**29.  *Artemisia tournefortiana* (Table 1, Figs. 12b, 15)**

Pollen grains prolate or spheroidal. Almost circular in equatorial view and trilobate circular in polar view. Apertures tricolporate. The exine near the colpi gradually thinned. $P = 20.76 \pm 0.98$ μm, $E = 20.43 \pm 0.83$ μm,

P/E = 1.02 ± 0.06, T = 3.33 ± 0.19 μm, L = 20.03 ± 0.79 μm, T/L = 0.17 ± 0.01. The exine ornamentation is psilate (LM), spinulate (SEM). Under SEM, D = 0.73 ± 0.06 μm, H = 0.42 ± 0.07 μm, D/H = 1.81 ± 0.33, Gs = 0.26 ± 0.07 μm, Ss = 1.25 ± 0.20 μm, Gs/Ss = 0.22 ± 0.08.

Habitat: widely distributed on hills, terraces, dry flood lands, waste fields, steppes, open forests, semi-marshlands.

**30.  *Artemisia dracunculus* (Table 1, Figs. 12c, 15)**

Pollen grains spheroidal. Almost circular in equatorial view and trilobate circular in polar view. Apertures tricolporate. The exine near the colpi gradually thinned. P = 22.89 ± 1.24 μm, E = 22.87 ± 1.32 μm, P/E = 1.00 ± 0.05, T = 2.82 ± 0.52 μm, L = 21.91 ± 1.35 μm, T/L = 0.13 ± 0.03. The exine ornamentation is psilate (LM), spinulate (SEM). Under SEM, D = 0.68 ± 0.05 μm, H = 0.45 ± 0.07 μm, D/H = 1.56 ± 0.31, Gs = 0.31 ± 0.10 μm, Ss = 0.92 ± 0.15 μm, Gs/Ss = 0.34 ± 0.11.

Habitat: dry slopes, steppes, semidesert steppes, forest steppes, forest margins, waste areas, roadsides, terraces, subalpine meadows, meadow steppes, dry river valleys, rocky slopes, saline-alkaline soils; 500-3800 m.

**31.  *Artemisia japonica* (Table 1, Figs. 13a, 15)**

Pollen grains spheroidal or oblate. Almost circular in equatorial view and trilobate circular in polar view. Apertures tricolporate. The exine near the colpi gradually thinned. P = 20.18 ± 1.28 μm, E = 21.23 ± 1.26 μm, P/E = 0.95 ± 0.05, T = 4.24 ± 0.49 μm, L = 21.02 ± 1.14 μm, T/L = 0.20 ± 0.02. The exine ornamentation is psilate (LM), spinulate (SEM). Under SEM, D = 0.57 ± 0.05 μm, H = 0.32 ± 0.05 μm, D/H = 1.80 ± 0.24, Gs = 0.26 ± 0.05 μm, Ss = 1.26 ± 0.16 μm, Gs/Ss = 0.21 ± 0.04.

Habitat: forest margins, waste areas, shrublands, hills, slopes, roadsides. Low elevations to 3300 m.

**32.  *Artemisia capillaris* (Table 1, Figs. 13b, 15)**

Pollen grains spheroidal or oblate. Almost circular in equatorial view and trilobate circular in polar view. Apertures tricolporate. The exine near the colpi gradually thinned. P = 19.53 ± 1.09 μm, E = 19.64 ± 1.62 μm, P/E = 1.00 ± 0.08, T = 3.54 ± 0.34 μm, L = 19.18 ± 0.97 μm, T/L = 0.18 ± 0.01. The exine ornamentation is psilate (LM), spinulate (SEM). Under SEM, D = 0.51 ± 0.06 μm, H = 0.36 ± 0.04 μm, D/H = 1.44 ± 0.30, Gs = 0.26 ± 0.04 μm, Ss = 1.27 ± 0.16 μm, Gs/Ss = 0.21 ± 0.05.

Habitat: humid slopes, hills, terraces, roadsides, riverbanks; 100-2700 m.

**33.  *Artemisia campestris* (Table 1, Figs. 13c, 15)**

Pollen grains prolate or spheroidal. Almost circular in equatorial view and trilobate circular in polar view. Apertures tricolporate. The exine near the colpi gradually thinned. P = 21.69 ± 0.85 μm, E = 21.26 ± 0.89 μm, P/E = 1.02 ± 0.07, T = 3.68 ± 0.33 μm, L = 21.21 ± 0.89 μm, T/L = 0.17 ± 0.02. The exine ornamentation is

psilate (LM), spinulate (SEM). Under SEM, D = 0.57 ± 0.09 μm, H = 0.38 ± 0.05 μm, D/H = 1.53 ± 0.23, Gs
= 0.41 ± 0.09 μm, Ss = 1.23 ± 0.19 μm, Gs/Ss = 0.34 ± 0.08.
Habitat: steppes, waste areas, rocky slopes, dune margins; 300-3100 m.
**34. *Kaschgaria brachanthemoides* (Table 1, Figs. 14a, 15)**
Pollen grains prolate or spheroidal. Almost circular in equatorial view and trilobate circular in polar view.
Apertures tricolporate. The exine near the colpi gradually thinned. P = 23.26 ± 1.44 μm, E = 22.09 ± 1.18 μm,
P/E = 1.06 ± 0.08, T = 3.93 ± 0.44 μm, L = 21.01 ± 1.28 μm, T/L = 0.19 ± 0.02. The exine ornamentation is
psilate (LM), spinulate (SEM). Under SEM, D = 0.55 ± 0.07 μm, H = 0.44 ± 0.05 μm, D/H = 1.25 ± 0.20, Gs
= 0 μm, Ss = 1.75 ± 0.20 μm, Gs/Ss = 0, Pertorations spacing (Ps) = 0.47 ± 0.14 μm.
Habitat: dry mountain valleys, old dry riverbeds; 1000-1500 m.
**35. *Ajania pallasiana* (Table 1, Figs. 14b, 15)**
Pollen grains spheroidal. Almost circular in equatorial view and trilobate circular in polar view. Apertures
tricolporate. The exine near the colpi gradually thinned. P = 35.16 ± 2.68 μm, E = 35.92 ± 3.31 μm, P/E = 0.98
± 0.03, T = 10.23 ± 0.85 μm, L = 38.31 ± 2.06 μm, T/L = 0.27 ± 0.03 μm. The exine ornamentation spinose.
Under SEM, D = 4.41 ± 0.35 μm, H = 3.47 ± 0.38 μm, D/H = 1.29 ± 0.21, Gs = 0 μm, Ss = 7.84 ± 1.25 μm,
Gs/Ss = 0, Ps = 0.39 ± 0.12 μm.
Habitat: thickets, mountain slopes, 200-2900 m.
**36. *Chrysanthemum indicum* (Table 1, Figs. 14c, 15)**
Pollen grains prolate or spheroidal or oblate. Almost circular in equatorial view and trilobate circular in polar
view. Apertures tricolporate. The exine near the colpi gradually thinned. P = 33.54 ± 1.71 μm, E = 34.42 ±
2.46 μm, P/E = 0.98 ± 0.08, T = 8.65 ± 0.89 μm, L = 34.82 ± 1.65 μm, T/L = 0.25 ± 0.02. The exine
ornamentation spinose. Under SEM, D = 2.94 ± 0.33 μm, H = 3.59 ± 0.29 μm, D/H = 0.82 ± 0.10, Gs = 0 μm,
Ss = 7.11 ± 0.76 μm, Gs/Ss = 0, Ps = 0.37 ± 0.13 μm.
Habitat: grasslands on mountain slopes, thickets, wet places by rivers, fields, roadsides, saline places by
seashores, under shrubs, 100-2900 m.
**Appendix B**
**Table B1.** List of the voucher specimen in PE Herbarium, Institute of Botany, Chinese Academy of Sciences

| Subgenus | Species | Specimen barcodes | Coll. No. | Habitat photograph sources |
|---|---|---|---|---|
| **Subg. *Tridentata*** | *Artemisia cana* | PE 01668975 | H.Mozingo 79-97 | © Jason Headley https://www.inaturalist.org/photos/54492753 |
| | *Artemisia tridentata* | PE 01917565 | Debreczy-Racz-Biro s.n. | © Matt Berger https://www.inaturalist.org/photos/117436654 |
| | *Artemisia californica* | PE 01668942 | Lewis S.Rose 69107 | © Don Rideout https://www.inaturalist.org/photos/108921528 |
| **Subg. *Artemisia*, Sect. *Artemisia*** | *Artemisia indica* | PE 00444597 | Tian-Lun Dai 104336 | © yangting https://www.inaturalist.org/photos/66336449 |
| | *Artemisia argyi* | PE 00420930 | K.M.Liou 9276 | © sergeyprokopenko https://www.inaturalist.org/photos/95820686 |
| | *Artemisia mongolica* | PE 00445665 | Cheng-Yuan Yang & Zu-Gui Li 36466a | © Nikolay V Dorofeev https://www.inaturalist.org/photos/163584035 |
| | *Artemisia vulgaris* | PE 01669703 | P.Frost-Olsen 1833 | © Sara Rall https://www.inaturalist.org/photos/120600448 |
| | *Artemisia selengensis* | PE 00479106 | Ming-Gang Li et al. 486 | © Gularjanz Grigoryi Mihajlovich https://www.inaturalist.org/photos/46352423 |
| | *Artemisia ludoviciana* | PE 01669278 | W.Hess 2405 | © Ethan Rose https://www.inaturalist.org/photos/77690333 |
| | *Artemisia roxburghiana* | PE 00478222 | Xingan collection team 70 | © Bo-Han Jiao |
| | *Artemisia rutifolia* | PE 00478427 | Ke Guo 12528 | © Daba https://www.inaturalist.org/photos/62207191 |
| **Subg. *Pacifica*** | *Artemisia chinensis* | PE 01565620 | Y.Tateishi J.Murata.Y.Endo et al. 15202 | © Jia-Hao Shen |

| | | | | |
|---|---|---|---|---|
| **Subg. *Seriphidium*** | *Artemisia kurramensis* | PE 01669178 | M.Togasi 1672 | © Andrey Vlasenko https://www.inaturalist.org/photos/133758174 |
| | *Artemisia compactum* | PE 00457459 | Hexi team 313 | © Chen Chen |
| | *Artemisia maritima* | No. 1338063 | s.n. | © torkild https://www.inaturalist.org/photos/86515371 |
| | *Artemisia aralensis* | No. 202006 | s.n. | © Полынь аральская https://www.plantarium.ru/lang/en/page/image/id/73063.html |
| | *Artemisia annua* | PE 01197344 | Wen-Hong Jin-Tian, Kai-Yong Lang, Ge Yang 328 | © Chen Chen |
| **Subg. *Artemisia*, Sect. *Abrotanum* I** | *Artemisia freyniana* | PE 01669030 | S.Kharkevich 753 | © Шильников Дмитрий Сергеевич https://www.inaturalist.org/photos/154390279 |
| | *Artemisia stechmanniana* | PE 00478480 | Shen-E Liu, Pei-Yun Fu et al. 4715 | © Bo-Han Jiao |
| | *Artemisia pontica* | PE 01589110 | Gy.Szollat & K.Dobolyi s.n. | © Martin Pražák https://www.inaturalist.org/photos/93438780 |
| **Subg. *Absinthium*** | *Artemisia frigida* | PE 00444197 | Ren-Chang Qin 0913 | © Suzanne Dingwell https://www.inaturalist.org/photos/125022240 |
| | *Artemisia rupestris* | PE 00478380 | Anonymous 948 | © Bo-Han Jiao |
| | *Artemisia sericea* | PE 01669585 | N.Maltzev 3175 | © svetlana_katana https://www.inaturalist.org/photos/48033353 |
| | *Artemisia absinthium* | PE 01668816 | G.Bujorean s.n. | © Станислав Лебедев https://www.inaturalist.org/photos/123569286 |
| **Subg. *Artemisia*, Sect. *Abrotanum* II** | *Artemisia abrotanum* | PE 01668792 | T.Leonova s.n. | © Андрей Москвичев https://www.inaturalist.org/photos/116106722 |
| | *Artemisia blepharolepis* | PE 00421006 | Kun-Jun Fu 7252 | © Ji-Ye Zheng |
| **Subg. *Artemisia*, Sect. *Abrotanum* III** | *Artemisia norvegica* | PE 01669339 | J.Haug s.n. | © Erin Springinotic https://www.inaturalist.org/photos/161393521 |

| | Species | Voucher | Collector | Photo credit |
|---|---|---|---|---|
| | *Artemisia tanacetifolia* | PE 00479744 | T.P.Wang W.3379 | © Alexander Dubynin https://www.inaturalist.org/photos/78902853 |
| | *Artemisia tournefortiana* | PE 00479786 | Ren-Chang Qin 2266 | © Chen Chen |
| **Subg. *Dracunculus*** | *Artemisia dracunculus* | PE 00421462 | Shen-E Liu et al. 8084 | © anatolymikhaltsov https://www.inaturalist.org/photos/76312868 |
| | *Artemisia japonica* | PE 00444874 | Qianbei team 2850 | © 陳達智 https://www.inaturalist.org/photos/44507659 |
| | *Artemisia capillaris* | PE 00421156 | Han-Chen Wang 4078 | © Cheng-Tao Lin https://www.inaturalist.org/photos/60639286 |
| | *Artemisia campestris* | PE 00421097 | T.N.Liou L.1008 | © pedrosanz-anapri https://www.inaturalist.org/photos/113822257 |
| **Outgroups** | *Kaschagaria brachanthemoides* | PE 01577564 | Yun-Wen Tian 22158 | © Chen Chen |
| | *Ajania pallasiana* | PE 00420032 | Guang-Zheng Wang 497 | © Игорь Поспелов https://www.inaturalist.org/photos/162408714 |
| | *Chrysanthemum indicum* | PE 01258852 | Anonymous 221 | © Bo-Han Jiao |

Note: In the absence of habitat photographs of two species, habitat photographs of species with which they have close
phylogenetic relationships and similar habitats were used in this study instead, i.e. the habitat photograph of *Kaschagaria*
*komarovii* was used instead of *Kaschagaria brachanthemoides,* the habitat photograph of *Artemisia taurica* for *Artemisia*
*kurramensis*.
**Table B2.** List of the voucher specimens in PE Herbarium, Institute of Botany, Chinese Academy of Sciences.
Sample No. 0 was the specimen for cluster analysis. Sample Nos.1-4 were used in testing intraspecific
variability in pollen exine ultrastructure characters among three representative species.

| Sample No. | SWS type | | LNS type | | SG type | |
| --- | --- | --- | --- | --- | --- | --- |
| | *Artemisia vulgaris* | | *Artemisia annua* | | *Artemisia maritima* | |
| | Specimen barcodes | Coll. No. | Specimen barcodes | Coll. No. | Specimen No. | Coll. No. |
| 0 | PE 01669703 | P.Frost-Olsen 1833 | PE 01197344 | Wen-Hong Jin-Tian, Kai-Yong Lang, Ge Yang 328 | No. 1338063 | s.n. |
| 1 | PE 00532417 | 75-1521 | PE 420433 | Xue-Zhong Liang 328 | No. 209452 | G. Belloteau 1912.9.25 |
| 2 | PE 00492025 | K.M.Liou L.6601 | PE 420647 | T.N.Liou & C.Wang 731 | No. 209446 | s.n. |
| 3 | PE 00492038 | P.C.Hoch, Jia-Rui Chen 86077 | PE 420660 | Da-Shun Wang 856 | s.n. | Hanelt Schultze-Motel 446 |
| 4 | PE 00492029 | Hong-Bin Cui, You-Run Lin, Zhen-Dai Xia 80-290 | PE 420664 | Lei,C.I. 858 | s.n. | O. Nordstedt 1901-10-9 |


**Author contributions.** YFW, YFY, TGG conceived the ideas, LLL, BHJ, KQL, and BS collected the
literature, LLL extracted and compiled the data, LLL, FQ, and BHJ made the statistical analysis, GX and ML
collected pictures, LLL, KQL, and BS drew the figures and tables, LLL, YFW, YFY, LJF, FQ, and GX wrote
the first draft of this manuscript, DKF corrected the various versions of the manuscript, while all authors
contributed substantially to revisions.
**Competing interests.** The authors declare that they have no conflict of interest.
**Acknowledgments.** We thank Dr. Jian Yang, Institute of Botany, Chinese Academy of Sciences, for his kind
help in drafting graphics. We appreciate Chen Chen from Institute of Botany, Chinese Academy of Sciences,
Ji-Ye Zheng from No. 1 Middle School of Jiyang Shandong, and Jia-Hao Shen from Institute of Botany,
Jiangsu Province and Chinese Academy of Sciences (Nanjing Botanical Garden Mem. Sun Yat-Sen), for their
enthusiastic assistance in providing habitat photographs.
**Financial support.** This research was supported by the Strategic Priority Research Program of the Chinese
Academy of Sciences (No. XDB26000000), National Natural Science Foundation of China (Nos. 31970223,
32070240, and 42077416), and the Chinese Academy of Sciences President's International Fellowship
Initiative (No. 2018VBA0016).

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
