# Peer review of "Artemisia* pollen dataset for exploring the potential ecological indicators in deep time"

_Earth System Science Data, 2022_

## Author Comment (AC1)

**Responses to referee #1**

The paper by Lu et al presents an *Artemisia* pollen morphometric dataset, with a view to using these data to differentiate *Artemisia* species and link these to different environments. The authors have clearly put some effort into sampling the herbarium specimens, taking photographs and generating the data, and the data are all fully accessible and approriately curated. However, I have a number of concerns about the extent of the data and how it has been presented and analysed, that to me compromise the results of the study and the potential for the dataset to be employed in further research. There are also issues with how the data generating process is documented in the paper, and given that this is the main point of a paper in ESSD (i.e. to support the publication of a dataset, including presenting how the data have been generated) I find this particularly problematic. I'll detail each of these points, one at a time:

We appreciate the time and effort you spent on reviewing. We have revised the manuscript according to your comments. Please see the following point-by-point responses.

**C1.1.** Extent of the data: the dataset comprises six morphological characters measured for the *Artemisia* pollen, plus three ratios based on these and a further character ('Pertoration spacing' in Table 1 - should this be Perforation spacing?) that is only relevant for the outgroup taxa. Two of the six measured characters - polar length and equatorial width - can be measured by light microscopy (LM, which most palynologists routinely use) and four - diameter of sp, a number of characters that could readily be measured using LM are left out, most obviously including exine thickness at the equator and poles, colpi length and pore dimensions. These would have been easy to include, and may well be useful for morphometric analysis and classification. I therefore think that these characters should be added in (on line 127 the authors state that colporate pattern was measured, so I assume this was considered at some point).

**R1.1.** Thanks for your suggestions! We would like to make some explanations on the selection of pollen traits as follows:

(1) Following your suggestions, we add some new traits under the LM. The statistical data for these new characters have been added to the revised manuscript (see revised Table 1 and Figure 15 listed below). Here, we measured and counted some new traits of the trilobate circular in this view of the pollen (Chen, 1987; Wang et al., 1995), i.e. exine thickness (T, Figure R1), pollen length (L, Figure R1) and T/L, considering that *Artemisia* pollen traits are more recognizable in the polar view under LM (Wang et al., 1995).

[Figure]

Figure R1. *Artemisia* pollen (e.g. *A. annua*) is trilobate circular in polar view. T stands for exine thickness, and L stands for pollen length.

(2) We focused on the selection of pollen traits under SEM because distinctive characters of certain taxa are only observed under SEM. Numerous studies on the morphology of *Artemisia* pollen clearly show that *Artemisia* pollen is highly uniform under LM (Wodehouse, 1926; Sing and Joshi, 1969; Ling, 1982; Chen, 1987; Wang et al., 1995). In addition, evidence shows that *Artemisia* pollen is difficult to distinguish from those of some related genera under LM due to their great similarity in pollen exine ornamentation and colporate patterns (Chen, 1987; Martín et al., 2001; Martín et al., 2003; Vallès et al., 2011). *Kaschgaria brachanthemoides* can only be separated from *Artemisia* by the feature "pollen exine with perforations and without granules" under SEM.

**Revised Table 1 and Figure 15 listed below:**

[revised manuscript text omitted]

**C1.2.** Also, while the authors include 36 species, with 20 pollen grains measured per species, there is no consideration of plant-to-plant (i.e. intraspecific) variability. Therefore, it is not clear how much differences between species are really just differences between plants, and if this dataset is to be truly valuable as a resource then 3 to 5 plants per species would need to be sampled.

**R1.2.** This is a good suggestion! Evidence shows that the pollen morphology in *Artemisia* is highly uniform under LM without discrimination (Wodehouse, 1926; Sing and Joshi, 1969; Ling, 1982; Chen, 1987; Wang et al., 1995), which might suggest that statistical analyses of the intraspecific morphological variation of pollen under the LM are limited or meaningless. Right now, the SEM technique has made it possible to subdivide *Artemisia* pollen into different types using pollen exine ultrastructure characters (Chen, 1987; Chen and Zhang, 1991; Sun and Xu, 1997; Jiang et al., 2005; Ghahraman et al., 2007; Shan et al., 2007; Hayat et al., 2009; Hayat et al., 2010; Hussain et al., 2019).

Following your suggestion, we selected one species respectively from the three pollen types corresponding to the three morphological clades of *Artemisia* pollen, i.e. *Artemisia vulgaris* (SWS type), *Artemisia annua* (LNS type), and *Artemisia maritima* (SG type), and sampled five specimens of each species (Table R1). Six pollen traits, i.e. diameter of spinule base (D), spinule height (H), D/H, granule spacing (Gs), spinule spacing (Ss), and Gs/Ss, were counted and analysed under SEM to test for intraspecific variation of pollen exine ultrastructure traits.

The test showed that it was feasible to use stable D/H and Gs/Ss for pollen type classification of *Artemisia* because 1) D/H and Gs/Ss were stable within species (Figure R2, Table R2) for the pollen classification; 2) D, H, Gs, and Ss were variable as size values, e.g. these four traits were significantly different within species in both *A. vulgaris* and *A. annua*, while D, H, and Ss were significantly different within species in *A. maritima* (Figure R2, Table R2). There was evidence showing that size values such as pollen exine ultrastructure size were often variable within species due to their genetic divergence, various habitats, and different experimental treatments (Mo et al., 1997; Zhao and Yao, 1999; Zhang and Qian, 2011).

**Table R1.** List of the voucher specimen in PE Herbarium, Institute of Botany, Chinese Academy of Sciences.

Sample No. 0 was the sample specimen in the original text, Sample Nos.1-4 were the newly added voucher specimen in this response.

| Sample No. | SWS type | | LNS type | | SG type | |
| | Artemisia vulgaris | | Artemisia annua | | Artemisia maritima | |
| | Specimen barcodes | Coll. No. | Specimen barcodes | Coll. No. | Specimen No. | Coll. No. |
|---|---|---|---|---|---|---|
| 0 | PE 01669703 | P.Frost-Olsen 1833 | PE 01197344 | Wen-Hong Jin-Tian, Kai-Yong Lang, Ge Yang 328 | No. 1338063 | s.n. |
| 1 | PE 00532417 | 75-1521 | PE 420433 | Xue-Zhong Liang 328 | No. 209452 | G. Belloteau 1912.9.25 |
| 2 | PE 00492025 | K.M.Liou L.6601 | PE 420647 | T.N.Liou & C.Wang 731 | No. 209446 | s.n. |
| 3 | PE 00492038 | P.C.Hoch, Jia-Rui Chen 86077 | PE 420660 | Da-Shun Wang 856 | - | Hanelt Schultze-Motel 446 |
| 4 | PE 00492029 | Hong-Bin Cui, You-Run Lin, Zhen-Dai Xia 80-290 | PE 420664 | Lei,C.I. 858 | - | O. Nordstedt 1901-10-9 |

[Figure]

**Figure R2.** Boxplots of intraspecific pollen exine ultrastructure characters from three species of *Artemisia*, showing the variations (M ± SD) in six pollen characters. Asterisks indicate statistically significant differences (*p <= 0.05, **p <= 0.01, ***p < 0.001).
(D: diameter of spinule base; H: spinule height; Gs: granule spacing; Ss: spinule spacing; Ps: perforation spacing)

**Table R2.** The results of ANOVA for intraspecific variation in pollen exine ultrastructure characters among three representative species

| Pollen exine ultrastructure characters | SWS type | LNS type | SG type |
| --- | --- | --- | --- |
| | *Artemisia vulgaris* | *Artemisia annua* | *Artemisia maritima* |
| D (μm) | significant | significant | significant |
| H (μm) | significant | significant | significant |
| D/H | non-significant | non-significant | non-significant |
| Gs (μm) | significant | significant | significant |
| Ss (μm) | significant | significant | significant |
| Gs/Ss | non-significant | non-significant | non-significant |

**C1.3.** I also note that only 33 species are from *Artemisia*, while the other three are 'outgroup' taxa. It's not clear to me what the point of the outgroup taxa is – this is not a phylogenetic analysis, there is no requirement to have outgroups when using cluster analysis or other exploratory multivariate methods, and given that the pollen from these other taxa looks quite different to *Artemisia*, with an additional character measured that is not relevant for the main taxa being studied, it is inevitable that the outgroup taxa will cluster separately to the 33 *Artemisia* species. This therefore seems like an odd addition that could just as easily be removed.

**R1.3.** Here we adopted a representative sampling strategy constrained by the phylogenetic framework to cover the maximum range of *Artemisia* pollen morphological variation. The aims for comparison with outgroups are to 1) delimit the *Artemisia* pollen more objectively, i.e. pollen grains oblate, spherical, or ellipsoidal; apertures tricolporate; almost circular in equatorial view and trilobate circular in polar view; the exine near the colpi gradually thinned; the exine has an obvious double structure of inner and outer layers where the outer is thicker than the inner under LM; the exine ornamentation is psilate (LM), spinulate and granules (SEM); 2) depict the stability of pollen morphological characters within *Artemisia* and morphological differences of intergeneric pollen more clearly. For instance, among the three outgroups we selected, *Chrysanthemum indicum* and *Ajania pallasiana* with distinct and long spines on the pollen exine ornamentation could be easily distinguished from the *Artemisia* pollen type with indistinct and short spinules under LM, while *Kaschgaria brachanthemoides* with perforated pollen exine could be distinguished from *Artemisia* with its non-perforated pollen exine only under SEM.

**C1.4.** The other data presented, for example distribution and climate data, are valuable but have simply been downloaded from GBIF and WorldClim, and are thus not original data generated by the authors but rather are publicly accessible data that anyone could already find and download.

**R1.4.** Indeed, these distribution and climate data are publicly accessible. However, we reprocessed these raw data by correlating it with *Artemisia* pollen traits and presenting it visually in graphical form. This makes it

easier for the reader to refer to and reuse and contributes to promoting new scientific knowledge.

In addition, we have given additional instructions on the quality check of GBIF data in the second paragraph of section 2.2 Data acquisition, as suggested by Referee #2.

Lines 136-143: "The scientific names of selected taxa were standardized according to Plants of the World Online (https://powo.science.kew.org/). The specimen sampling coordinates of the corresponding taxa were obtained from the Global Biodiversity Information Facility (GBIF, https://www.gbif.org/). Only preserved specimens were filtered for GBIF data given their well-documented geographical information and the availability of specimens as definitive vouchers. The distribution data on observations and cultivated collections provided by GBIF were excluded because they may contain incorrect identification or incorrect geo-referencing (Brummitt et al., 2020). Next, the distribution data was standardized cleaned using R package "CoordinateCleaner" (Zizka et al., 2019); no outliers were found."

**C2.1.** Documentation of the data generating process: on line 126 the authors state that for each species 20 grains were measured using LM, and 5 using SEM. However, there are 20 measurements for the characters relating to sculpture size and distribution, which if I understand it right were measured using SEM. So where did these come from? Were 20 grains measured using SEM? This really needs to be clear so that people know what they are dealing with.

**R2.1.** In section 2.2 Data acquisition, we added additional clarifications in the first paragraph.

Lines 127-130: "We chose five pollen grains under SEM for each exine ornamentation trait in each species (Figs. 2c-f, D: Diameter of spinule base; H: Spinule height; D/H; Gs: Granule spacing; Ss: Spinule spacing; Gs/Ss; Ps: Perforation spacing), and on average, randomly selected four regions of each pollen grain for measuring, yielding a total of 20 measurements."

**C2.2.** And in the cluster analysis part (lines 135 to 137) the authors state that five main clusters were distinguished. But why five, when three main types of *Artemisia* pollen are considered elsewhere in the text?

**R2.2.** Indeed, five morphological clades could be identified (Figure R3) in the cluster analysis part (lines 135 to 137), comprising two outgroup pollen clades (① and ②), and three morphological clades of *Artemisia* pollen (③, ④, and ⑤), corresponding to the three pollen types of *Artemisia*.

In Line 148, We revised "*Artemisia* pollen data" to "*Artemisia* and its outgroup pollen data" in order to be more precise.

[Figure]

**Figure R3.** Illustration of the number of clusters of pollen morphological clades of *Artemisia* and its outgroups.

**C3.1.** Potential for re-use: The lack of measured characters, and the lack of within-species replication, really limits how useful the data will be for future work. This is compounded by the fact that the authors have already carried out an analysis of the data and shown that there is a lot of morphological overlap among taxa, which more or less answers the question of whether these sorts of measurements could be used for classification of individual pollen grains. The reliance on measurements of small sculptural elements that require SEM images also limits how useful this dataset will be for routine use, because not only is LM still the main way palynologists study their samples, even if SEMs are available there is only so much picking of pollen from samples that palynologists are going to do to generate further data to analyse using the dataset that the authors are presenting.

**R3.1.** As mentioned above, *Artemisia* pollen morphology is remarkably uniform under LM (Wodehouse, 1926; Sing and Joshi, 1969; Ling, 1982; Chen, 1987; Wang et al., 1995). Although LM is still one of the most common methods of observation and statistics for today's palynologists, however, with the advancement of science and technology, SEM has become a standard tool in palynological research, with an increasing number of further palynological studies implementing SEM extensively to assist pollen identification (e.g. Grímsson et al., 2012; Zhang et al., 2018; Lu et al., 2019).

Our enhanced classification criteria for *Artemisia* pollen under SEM have definitely improved the resolution of pollen identification. Stable traits (D/H and Gs/Ss) under SEM were shown to be feasible for pollen classification of *Artemisia* (Table R2, Figure R2) since they won't require to take into account genetic divergence, source plant environments, or experimental treatments (Mo et al., 1997; Zhao and Yao, 1999;

Zhang and Qian, 2011). One-way ANOVA revealed significant differences between these two traits (Fig. 17). The single-grain technique could be used to pick out fossilized pollen grains and photograph them under LM and SEM, then further classify them based on the observation and measurement of exine ornamentation under SEM, thereby inferring the habitat range to which they relate (Figure R4).

[Figure]

**Figure R4.** Flowchart for the application of *Artemisia* pollen classification.

**C3.2.** It is possible that the images that the authors have made available along with the data will be useful for further analysis, i.e. with deep learning based classification, but the authors do not flag this up so the whole resource might be missed, and again the lack of intraspecific replication probably limits how useful these photos are for classification attempts.

**R3.2.** Thank you for your valuable suggestion! These pollen photographs really have great potential and the possibility to be used for deep learning research. We added a paragraph highlighting the potential application of these images for image identification in section 6 Summary.

Lines 428-432: "Moreover, these pollen photographs also have the potential and the possibility to be used for deep learning research. We are attempting to automatically identify pollen images using pollen assemblages from the eastern Central Asian desert as an example with deep convolutional neural network (DCNN) of artificial intelligence. Pollen images of the various *Artemisia* species supplied here, as well as the growing number of intraspecific replications in the future, will all be served for projected image identification research."

**C4.1.** Data analysis: a brief comment on this, but it would be useful to accompany the cluster analysis with an ordination technique such as PCA, because with this you can see how much your different groups overlap in the morphospace, and therefore how distinct they are morphologically (cluster analysis imposes a hierarchical structure whether one is there or not, while ordinations reveal both gradients and groupings in the data, and give a more visually intuitive understanding of how similar or different the various taxa are).

**R4.1.** Thank you for your helpful suggestion! In the second paragraph of section 4.1 The pollen classification of *Artemisia* now includes principal component analysis.

Lines 332-338: "Eight pollen morphological traits (P/E, H, D, D/H, Ss, Gs, Gs/Ss, and Ps) were selected for the principal component analysis (PCA) of 36 taxa of *Artemisia* and its outgroups (Fig. 18) and grouped according to the five clades of the cluster analysis , i.e. the five pollen types. The results reveal that *Artemisia* pollen morphology differs significantly from that of the outgroups, and the three *Artemisia* pollen types could be distinguished."

[Figure]

**Figure 18.** Principal component analysis of 36 taxa of *Artemisia* and its outgroups.

**C4.2.** So, with all of this said, what could the authors do to fix this? As it stands I think the paper would be a better fit for a regular journal, where the aim is to answer a scientific question rather than present a dataset. Something specialist like Palynology, Grana or Review of Palynology and Palaeobotany would be suitable here, although the authors would have to make clear that the lack of sampling might limit the strength of their conclusions. The data would be very appropriate as supplementary material to such a paper though, including the data downloaded from GBIF and WorldClim that would support the analyses in the paper very well.

If the authors do want this to be published in ESSD, then I think there is nothing for it but to generate more data, using more characters and more plant specimens. In the paper itself I suggest much more emphasis on careful documentation of how the data were generated (which at the moment only covers two paragraphs of the whole paper – lines 111 to 129), and much less on the analysis that comes with it. More detail on different contexts in which the data and images could be re-used would also be worthwhile, given that this is one of the main points of the publication.

**R4.2.** Indeed, long-term and comprehensive international collaborations are needed for attaining the ultimate aim of pollen morphological research spanning all species of *Artemisia*, a group with a wide distribution and a significant number of species in the Northern Hemisphere.

To cover the maximum range of *Artemisia* pollen morphological variation, our pollen dataset of 36 species from 9 clades and 3 outgroups of *Artemisia* was constrained by the phylogenetic framework. Although the dataset currently only contains 36 species with pollen photographs under LM and SEM, statistical data of pollen morphological traits, together with their source plant distribution, and corresponding environmental factors, the dataset as designed is open and expandable for new pollen data from *Artemisia* worldwide to better serve the global environment assessment and refined reconstruction of vegetation in the geological past. This is a good attempt to bridge the gap between biology and earth sciences and is a promising start toward correlating *Artemisia* pollen morphology and the habitats of their source plants.

Table 2 in section 5 Data availability summarized the *Artemisia* pollen datasets generated by this study, and we described in detail the process of generating each type of data in section 2.2 Data acquisition.

Lines 121-147: "Pollen samples were acetolyzed by the standard method (Erdtman, 1960) and fixed in glycerine jelly. Standard procedures were followed for LM and SEM (Chen, 1987; Wang et al., 1995). The pollen grains were photographed under LM (Leica DM 4000) at a magnification of ×1000 and SEM (Hitachi S-4800) at an accelerating voltage of 30 kV. The pollen terminology followed the descriptions of Hesse et al. (2009) and Halbritter et al. (2018). The statistical pollen morphological traits under LM (Figs. 2a-b, P: Polar length; E: Equatorial width; P/E; T: Exine thickness; L: Pollen length; T/L) of each species were measured from 20 pollen grains. For each exine ornamentation trait under SEM (Figs. 2c-f, D: Diameter of spinule base; H: Spinule height; D/H; Gs: Granule spacing; Ss: Spinule spacing; Gs/Ss; Ps: Perforation spacing) of each species, we selected five pollen grains and randomly picked four regions of each pollen grain on average for measuring, obtaining a total of 20 measurements. The mean value (M) and standard deviation (SD) of the

pollen grains of each species were measured and calculated in both polar and equatorial views (Appendix A, Table 1).

[Figure]

**Figure 2.** Graphical illustration of measured pollen morphological traits in *Artemisia* (a-b: *A. annua*; c-d: *A. vulgaris*) and outgroups (e: *Kaschagaria brachanthemoides*; f: *Ajania pallasiana*). Scale bar in LM and SEM overview 10 μm, in SEM close-up 1 μm.

The scientific names of selected taxa were standardized according to Plants of the World Online (https://powo.science.kew.org/). The specimen sampling coordinates of the corresponding taxa were obtained

from the Global Biodiversity Information Facility (GBIF, https://www.gbif.org/). Only preserved specimens were filtered for GBIF data given their well-documented geographical information and the availability of specimens as definitive vouchers. The distribution data on observations and cultivated collections provided by GBIF were excluded because they may contain incorrect identification or incorrect geo-referencing (Brummitt et al., 2020). Next, the distribution data was standardized cleaned using R package "CoordinateCleaner" (Zizka et al., 2019); no outliers were found.

The corresponding environmental factors including altitude and 19 climate parameters of these coordinates were obtained from WorldClim (https://www.worldclim.org/) with a spatial resolution of 30 seconds (~1 km$^2$) in 1970-2000 by Extract MultiValues To Points using ArcGIS 10.2 software in bilinear interpolation."

As for the reuse of this dataset, we provided an example of how these data could be used for study in section 4 Potential use of the *Artemisia* pollen datasets: we clustered the pollen morphological trait data of representative species to distinguish different pollen types in *Artemisia* and attempted to decipher the underlying causes of the long-standing disagreement in the palynological community on the correlation between *Artemisia* pollen and aridity by recognizing the different ecological implications of *Artemisia* pollen types.

Those potential and prospective researchers might use our pollen morphological traits and abundant photographs to identify *Artemisia* pollen and link those pollen grains to specific pollen types by applying our pollen classification of *Artemisia* to infer the environment. At the same time, researchers could measure and analyse pollen morphological traits of their choice using photographs of *Artemisia* pollen in this dataset taken under LM and SEM. By the way, the utilization of these data and images in deep learning is included in our next research objective.

**References for the above responses are listed below:**

[revised manuscript text omitted]

Zhang, Y. N. and Qian, C.: SEM observation on pollen morphology of lily species, Acta Prataculturae Sinica, 20, 111-118, 2011 (in Chinese).

Zhao, X. L. and Yao, C. H.: Pollen Morphlogy Differences among *Osmanthus fragrans* Cultivars, Journal of Hubei Minzu University (Nature Science Edition), 17, 16-20, 1999 (in Chinese).

Zizka, A., Silvestro, D., Andermann, T., Azevedo, J., Ritter, C. D., Edler, D., Farooq, H., Herdean, A., Ariza, M., Scharn, R., Svantesson, S., Wengstrom, N., Zizka, V., and Antonelli, A.: CoordinateCleaner: Standardized cleaning of occurrence records from biological collection databases, Methods in Ecology and Evolution, 10, 744-751, https://doi.org/10.1111/2041-210X.13152, 2019.

---

## Author Comment (AC2)

**Responses to referee #2**

This manuscript provides a wealth of data on the pollen morphology of *Artemisia* and its relation to the habitat preferences of their mother plants. Such data will serve in the future for better reconstructions of paleoenvironmental setting, which in turn will help to better understand the earth's climate past.

The manuscript is well written, well structured and with a very informative introduction. The methods used are all well established and robust. I could envision more sophisticated statistical analyses to be applied, but this may be done in the future with the database grown further.

Thanks for these constructive suggestions. We have revised the manuscript according to your comments. Please see the following point-by-point responses.

**C1.** One critique I may raise here, is the assessment of the distribution of taxa in different biomes (here wrongly called ecoregions). First of all, the results shown in Figure 18 do not support the claims made in the text with respect to habitat preferences of the species or morpho-groups. Second, biomes are too large a vegetation unit to be able to reflect habitat preferences. Here, probably the actual habitat of the sampled specimen needs to be considered, or a different approach to be developed.

**R1.** Following your suggestions, we have replaced "ecoregions" with "biomes", and removed the assessment of the distribution of species specimens from GBIF in different biomes from Fig. 18. Therefore, the problem that Fig. 18b does not support habitat preference well in the text no longer exists. Section 4.2 The Ecological Implications of *Artemisia* pollen Types are modified as follows.

Lines 368-399: "Plotting the distribution data of 33 species from 9 main branches of *Artemisia* constrained by the phylogenetic framework (Fig. 1) onto the global terrestrial biomes (Fig. 20), we noticed that the genus is widely distributed from forest to grassland, desert, and saline habitats (Figs. 16, 17a, 20). Furthermore, different species of *Artemisia* with SWS pollen type (Fig. 20a) and LNS type (Fig. 20b) have a rather wide distribution with severely overlapping ranges while those with SG type (Fig. 20c) have narrow and isolated distributions.

[Figure]

**Figure 20.** The global distribution pattern of 3 *Artemisia* pollen types in terrestrial biomes (modified from Olson et al., 2001). a. SG type; b. LNS type; c. SWS type.

14 terrestrial biomes: 01. Tropical and Subtropical Moist Broadleaf Forests; 02. Tropical and Subtropical Dry Broadleaf Forests; 03. Tropical and Subtropical Coniferous Forests; 04. Temperate Broadleaf and Mixed Forests; 05. Temperate Coniferous Forests; 06. Boreal Forests/Taiga; 07. Flooded Grasslands and Savannas; 08. Montane Grasslands and Shrublands; 09. Tundra; 10: Mediterranean Forests, Woodlands, and Shrub; 11. Tropical and Subtropical Grasslands, Savannas, and Shrublands; 12. Temperate Grasslands, Savannas, and Shrublands; 13. Deserts and Xeric Shrublands; 14. Mangroves; 15. Lakes; 16. Rock and Ice.

The ecological implications of *Artemisia* pollen types mentioned above fall into four categories. (i) *Artemisia* with the SG pollen type all belong to the subg. *Seriphidium*, which generally grows in dry habitats ranging from grassland desert to desert and coastal saline-alkaline environments, with their distribution largely limited to Eurasia and growing at low altitude (Figs. 17b, 20c, 21). (ii) The habitats of *Artemisia* with LNS pollen type have a global distribution and occur in forest, grassland and desert, and even coastal areas (Figs. 17b, 20b, 21), with the highest mean annual temperature (MAT). Hence, the LNS pollen type is a

generalist. (iii) *Artemisia* with SWS pollen type include Sect. *Artemisia* and its habitats range from forest to desert, although most of the taxa are confined to humid environments from forest to grassland with a global distribution and the highest mean annual precipitation (MAP, Figs. 17b, 20c, 21). (iv) If the SWS pollen type and the SG pollen type appear together, the range of vegetation types could be reduced to grassland desert and desert through niche coexistence (Fig. 17b).

[Figure]

**Figure 21.** Violin diagrams of three pollen types from *Artemisia*, showing the variations (25%-75%) in MAT, MAP, and altitude. Asterisks indicate statistically significant differences (p < 0.001).

In addition, we noticed that *Kaschgaria brachanthemoides* as an outgroup of *Artemisia* lives in dry mountain valleys or dry riverbeds of Northwest China (Toksun) and Kazakhstan, with highly characteristic pollen (Fig. 14a), narrow habitats (Fig. 17b), and regional distribution (Fig. 16-34) and has the potential to indicate some specific habitats."

Following your constructive suggestion, we randomly sampled the coordinates of the distribution of 33 species of *Artemisia* using the R language package "sampling" (Tillé and Matei, 2021). If the number of species distribution coordinates was more than ten, ten coordinates are randomly selected, while any number below 10 was used completely. The refined biomes of these sampling coordinates were obtained using Arcgis 10.2 software according to 867 terrestrial biomes distinct units by Olson et al. (2001).

The proportion of the three pollen types of *Artemisia* in the four vegetation types (forest, grassland, desert, and saline) was determined by a specimen sampling survey, with the SWS pollen type having the highest proportion in the forest, the LNS pollen type having a more balanced distribution, and the SG pollen type having the highest proportion in the desert and saline (Figure R1). These were generally consistent with the results of our literature review (Fig. 16b).

[Figure]

**Figure R1.** Distribution of randomly sampled specimens showing the proportion of the three pollen types of *Artemisia* in the four vegetation types.

**C2.** The data are well documented. The rich dataset of detailed high quality pollen fotos is crucial for palynological studies and a treasure in itself. The dataset on measurements of pollen grains is quite small, considering only one specimen per species. It should definitively, and hopefully will, be enlarged in the future, but for sure will serve as a basis or blueprint to other overarching statistical analyses on pollen morphology.

**R2.** Thank you for your valuable suggestion! We modify the section 6 Summary as follows.

Lines 406-436: "To cover the maximum range of *Artemisia* pollen morphological variation, we provide a pollen dataset of 36 species from 9 clades and 3 outgroups of *Artemisia* constrained by the phylogenetic framework, containing high-quality pollen photographs under LM and SEM, statistical data of pollen morphological traits, together with their source plant distribution, and corresponding environmental factors. Here, we attempt to decipher the underlying causes of the long-standing disagreement in the palynological community on the correlation between *Artemisia* pollen and aridity by recognizing the different ecological implications of *Artemisia* pollen types.

This dataset should work well for identifying and classifying *Artemisia* pollen from Neogene and Quaternary sediments. While *Artemisia* pollen grains are uniform in morphology under LM, different types

can be recognized under SEM. So, the single-grain technique for picking out fossil pollen grains and photographing the same grains under LM and SEM should provide valuable insights in the diversity of fossil *Artemisia* (Ferguson et al., 2007; Grímsson et al., 2011; Grímsson et al., 2012; Halbritter et al., 2018). Furthermore, those *Artemisia* pollen grains could then be compared with the rich photographs from this dataset, and together with the key provided here, possibly attributed to one of the three *Artemisia* pollen types, which in turn may provide a link to the different habitat ranges.

However, the application of this dataset probably may not work well for the Palaeogene, as 1) *Artemisia* might have originated in the Palaeocene, although there is no evidence for a specific location or time interval of its origin (e.g. Ling 1982; Wang 2004; Miao 2011); 2) both the lack of macrofossils of *Artemisia* and the strong pollen similarity between *Artemisia* and its closely related taxa under LM might lead to confusion and more uncertainty in tracing the origin of *Artemisia*. On the other hand, the present dataset provides a potential morphological tool to distinguish *Artemisia* pollen grains from those of its related taxa at the SEM level and may shed light on the origin of this genus in the Palaeogene.

Moreover, these pollen photographs also have potential and the possibility to be used for deep learning research. We are attempting to automatically identify pollen images using pollen assemblages from the eastern Central Asian desert as an example with deep convolutional neural network (DCNN) of artificial intelligence. Pollen images of the many species of *Artemisia* provided here, and the increasing number of intraspecific replications in the future, will all serve for projected image identification research.

Finally and most importantly, the *Artemisia* pollen dataset as designed is open and expandable for new pollen data from *Artemisia* worldwide in order to better serve the global environment assessment and refined reconstruction of vegetation in the geological past as a basis or blueprint for other overarching statistical analyses on pollen morphology."

**C3.** With respect to the dataset which was downloaded from GBIF and WorldClim, I do not see the necessary to provide them here (except to justify the figures in the manuscript). GBIF is a dynamic database growing constantly and new retrievals from GBIF will give more detailed information with every day. Moreover, GBIF data have to be carefully quality checked before further analyses, but there is no awareness of this mentioned here in the text.

**R3.** This is a helpful suggestion! The GBIF species distribution data we have used were updated until 09 November 2021 and were included in the dataset to 1) visualize the global distribution of 36 species and three *Artemisia* pollen types in terrestrial biomes, and 2) attempt to determine the preferences of different pollen

types for different climatic indicators by correlating the environmental factors of the distribution coordinates of these specimens.

Although the climatic parameters may not be well-developed at present, the preferences of different pollen types for distinct climatic indicators in these data were consistent with the above literature survey, e.g. *Artemisia* with SWS pollen type is primarily found in humid environments from forest to grassland with the highest MAP (Figs. 17b, 20c, 21), whereas *Artemisia* with the SG pollen generally grows in dry habitats ranging from grassland desert to desert and coastal saline-alkaline environments with the lowest altitude (Figs. 17b, 20c, 21).

We have added some instructions on the quality check of GBIF data in the second paragraph of section 2.2 Data acquisition.

Lines 136-143: "The scientific names of selected taxa were standardized according to Plants of the World Online (https://powo.science.kew.org/). The specimen sampling coordinates of the corresponding taxa were obtained from the Global Biodiversity Information Facility (GBIF, https://www.gbif.org/). Only preserved specimens were filtered for GBIF data given their well-documented geographical information and the availability of specimens as definitive vouchers. The distribution data on observations and cultivated collections provided by GBIF were excluded because they may contain incorrect identification or incorrect geo-referencing (Brummitt et al., 2020). Next, the distribution data was standardized cleaned using R package "CoordinateCleaner" (Zizka et al., 2019); no outliers were found."

**C4.** In general, I consider this manuscript ready for publications after minor revisions. I have a few remarks annotated in the pdf.

**R4.** Thank you very much for your kind correction! The manuscript has been revised following your remarks.

**The references for the above responses are listed as below:**

Brummitt, N., Araujo, A. C., and Harris, T.: Areas of plant diversity-What do we know?, Plants People Planet, 3, 33-44, https://doi.org/10.1002/ppp3.10110, 2021.

Olson, D. M., Dinerstein, E., Wikramanayake, E. D., Burgess, N. D., Powell, G. V. N., Underwood, E. C., D'Amico, J. A., Itoua, I., Strand, H. E., Morrison, J. C., Loucks, C. J., Allnutt, T. F., Ricketts, T. H., Kura, Y., Lamoreux, J. F., Wettengel, W. W., Hedao, P., and Kassem, K. R.: Terrestrial ecoregions of the worlds: A new map of life on Earth, Bioscience, 51, 933-938, https://doi.org/10.1641/0006-3568(2001)051[0933:teotwa]2.0.co;2, 2001.

Tillé, Y. and Matei, A.: sampling: Survey Sampling. R package version 2.9., https://CRAN.R-project.org/package=sampling, 2021.

Zizka, A., Silvestro, D., Andermann, T., Azevedo, J., Duarte Ritter, C., Edler, D., Farooq, H., Herdean, A., Ariza, M., Scharn, R., Svantesson, S., Wengström, N., Zizka, V., and Antonelli, A.: CoordinateCleaner: Standardized cleaning of occurrence records from biological collection databases, Methods in Ecology

and Evolution, 10, 744-751 %U https://onlinelibrary.wiley.com/doi/abs/710.1111/2041-1210X.13152, 10.1111/2041-210X.13152, 2019.

---

## Author Response (AR1)

**Response to Reviewers**

Dear Prof. Dr. Alessio Rovere,

Thank you very much for handling our manuscript essd-2022-23 entitled "*Artemisia* pollen dataset for exploring the potential ecological indicators in deep time". Right now, we have completed our revisions following the suggestions and comments of the two reviewers. We hope the new version is suitable for publication in your journal.

Attached below are the items we respond to the questions posed by two reviewers and structure the response to the reviewers, followed by the sequence: (1) reviewer comment, (2) author's response, (3) changes to manuscript. Please let us know if further changes are required.

Yours Sincerely,

Yu-Fei Wang and co-authors

**Responses to referee #1**

**Reviewer comment 1:** The paper by Lu et al presents an *Artemisia* pollen morphometric dataset, with a view to using these data to differentiate *Artemisia* species and link these to different environments. The authors have clearly put some effort into sampling the herbarium specimens, taking photographs and generating the data, and the data are all fully accessible and approriately curated. However, I have a number of concerns about the extent of the data and how it has been presented and analysed, that to me compromise the results of the study and the potential for the dataset to be employed in further research. There are also issues with how the data generating process is documented in the paper, and given that this is the main point of a paper in ESSD (i.e. to support the publication of a dataset, including presenting how the data have been generated) I find this particularly problematic. I'll detail each of these points, one at a time:

Author's response: We appreciate the time and effort you spent on reviewing. We have revised the manuscript according to your comments. Please see the following point-by-point responses.

**Reviewer comment 2:** Extent of the data: the dataset comprises six morphological characters measured for the *Artemisia* pollen, plus three ratios based on these and a further character ('Pertoration spacing' in Table 1 - should this be Perforation spacing?) that is only relevant for the outgroup taxa. Two of the six measured characters - polar length and equatorial width - can be measured by light microscopy (LM, which most palynologists routinely use) and four - diameter of sp, a number of characters that could readily be measured using LM are left out, most obviously including exine thickness at the equator and poles, colpi length and pore dimensions. These would have been easy to include, and may well be useful for morphometric analysis and classification. I therefore think that these characters should be added in (on line 127 the authors state that colporate pattern was measured, so I assume this was considered at some point).

**Author's response: Thanks for your suggestions! We would like to make some explanations on the selection of pollen traits as follows:**

(1) Following your suggestions, we add some new traits under the LM. The statistical data for these new characters have been added to the revised manuscript (see revised Table 1 and Figure 15 listed below) and table of "Statistical pollen morphological traits" in datasets available at Zenodo (https://doi.org/10.5281/zenodo.6791891). Here, we measured and counted some new traits of the trilobate circular in this view of the pollen (Chen, 1987; Wang et al., 1995), i.e. exine thickness (T, Figure R1), pollen length (L, Figure R1) and T/L, considering that Artemisia pollen traits are more recognizable in the polar view under LM (Wang et al., 1995).

Figure R1. *Artemisia* pollen (e.g. *A. annua*) is trilobate circular in polar view. T stands for exine thickness, and L stands for pollen length.

(2) We focused on the selection of pollen traits under SEM because distinctive characters of certain taxa are only observed under SEM. Numerous studies on the morphology of *Artemisia* pollen clearly show that *Artemisia* pollen is highly uniform under LM (Wodehouse, 1926; Sing and Joshi, 1969; Ling, 1982; Chen, 1987; Wang et al., 1995). In addition, evidence shows that *Artemisia* pollen is difficult to distinguish from those of some related genera under LM due to their great similarity in pollen exine ornamentation and colporate patterns (Chen, 1987; Martín et al., 2001; Martín et al., 2003; Vallès et al., 2011). *Kaschgaria brachanthemoides* can only be separated from *Artemisia* by the feature "pollen exine with perforations and without granules" under SEM.

**Changes to manuscript (revised Table 1 and Figure 15 listed below):**

**Table 1.** Pollen morphological traits of 36 selected species (P: Polar length; E: Equatorial width; T: Exine thickness; L: Pollen length; D: Diameter of spinule base; H: Spinule height; Gs: Granule spacing; Ss: Spinule spacing; Ps: Perforation spacing).

| No. | Species                  | P
(µm) | Ε
(μm) | P/E  | Τ
(μm) | L
(µm) | T/L  | D
(µm) | Η
(μm) | D/H  | Gs
(µm) | Ss
(µm) | Gs/Ss | Ps
(µm) |
|-----|--------------------------|-----------|-----------|------|-----------|-----------|------|-----------|-----------|------|------------|------------|-------|------------|
| 1   | Artemisia cana           | 23.46     | 24.5      | 0.96 | 3.91      | 24.58     | 0.16 | 0.58      | 0.46      | 1.28 | 0.33       | 1.60       | 0.21  | 0          |
| 2   | Artemisia
tridentata  | 21.36     | 20.69     | 1.04 | 3.55      | 22.35     | 0.16 | 0.76      | 0.60      | 1.30 | 0.24       | 1.12       | 0.22  | 0          |
| 3   | Artemisia
californica | 18.94     | 19.13     | 0.99 | 2.70      | 18.85     | 0.14 | 0.75      | 0.71      | 1.08 | 0.24       | 1.45       | 0.17  | 0          |
| 4   | Artemisia indica         | 23.47     | 23.81     | 0.99 | 3.50      | 23.31     | 0.15 | 0.76      | 0.39      | 2.04 | 0.28       | 1.21       | 0.24  | 0          |

[revised manuscript text omitted]

---

## Author Response (AR2)

**Response to Comments of Anonymous Referee #1**

Dear Prof. Dr. Alessio Rovere,

Thank you very much for handling our manuscript essd-2022-23 entitled "*Artemisia* pollen dataset for exploring the potential ecological indicators in deep time" and moving it forward. Right now, we have completed our revisions following the comments of anonymous referee #1. We hope this new version is suitable for publication in your journal.

Attached below are the items we respond to the comments posted by referee #1 and structure the response followed by the sequence: (1) reviewer comment, (2) author's response, and (3) changes to manuscript. Please let us know if further changes are required.

Yours Sincerely,

Yu-Fei Wang and co-authors

**Responses to referee #1**

**Reviewer comment 1:** I thank the authors for addressing my comments in detail, and generating additional data to add to their dataset. I just have a two minor suggestions for revision:

**Author's response:** We appreciate the time and effort you spent on reviewing. We have revised the manuscript according to your comments. Please see the following point-by-point responses.

**Reviewer comment 2:** Line 24: suggest changing to '9360 pollen morphological trait measurements' - the traits are usually considered to be the characters that you are measuring, rather than the individual measurements.

**Author's response:** Done.

**Changes to manuscript:** We changed "9360 statistical pollen morphological traits" to "9360 pollen morphological trait measurements" in Abstract, Table 3 and the description of dataset.

**Reviewer comment 3:** Intraspecific variability (reviewer comment 3 in the authors' response to my review): there are additional measurements and analyses here but they are only in the review response, not the paper. It would be much more useful to readers to work these into the manuscript, including Figure R2, and to include the additional measurements in the supplementary material (this could be a separate sheet in the 'Statistical pollen morphological traits' Excel file).

**Author's response:** Thank you for your helpful suggestions and improvements on the manuscript!

**Changes to manuscript:** 1. We added this part into the revised manuscript as an independent section 4.2 Testing the pollen intraspecific variability within *Artemisia* (lines 361-387).

**"4.2 Testing the pollen intraspecific variability within *Artemisia**

[revised manuscript text omitted]

3. We revised Table 3 in section 5 Data availability including the additional measurements of the intraspecific variability test (line 434).

**Table 3.** *Artemisia* pollen datasets in this study.

| Data type | Data format | Data acquisition | Data accessibility |
|---|---|---|---|
| The phylogenetic framework of *Artemisia* pollen sampling. | .png | Literature survey (modified from Malik et al., 2017). | This article |
| A voucher specimen list of 36 representative species. | .doc | Pollen samples were obtained from PE herbarium at the Institute of Botany, Chinese Academy of Sciences. | |
| 12 illustrations of pollen grains and the habitats of their source plants. | .png | Habitat photos from online sources (Appendix Table A). | |
| 4018 original pollen photographs (3205 under LM, 813 under SEM). | .jpg | Pollen samples were acetolyzed by the standard method and fixed in glycerine jelly. The pollen grains were photographed under LM and SEM using standard procedures. | Zenodo (https://doi.org/10.5281/zenodo.6900308; Lu et al., 2022) |
| 9360 pollen morphological trait measurements of 36 representative species. | .xlsx | Statistical data of pollen morphological traits were measured by standard methods. | |
| 1800 pollen morphological trait measurements for testing the pollen intraspecific variability within *Artemisia*. | .xlsx | Statistical data of pollen morphological traits were measured by standard methods. | |
| 30858 source plant occurrence information, and corresponding environmental factors including altitude and 19 climate parameters. | .xlsx | Their source plant distribution coordinates were obtained from GBIF (https://doi.org/10.15468/dl.596xd9). The corresponding environmental factors of these coordinates were obtained from WorldClim (https://www.worldclim.org/) with a spatial resolution of 30 seconds between 1970-2000. | |

4. These additional measurements of intraspecific variability have been added to the "Statistical pollen morphological traits" Excel file as a new sheet in datasets available at Zenodo (https://doi.org/10.5281/zenodo.6900308).